# Martingale Posterior Neural Networks for Fast Sequential Decision Making

**Gerardo Duran-Martin**[1]    **Leandro Sánchez-Betancourt**[1,2]
**Álvaro Cartea**[1,2]    **Kevin Murphy**[3]
[1] Oxford-Man Institute of Quantitative Finance
[2] Mathematical Institute, University of Oxford
[3]Google Deepmind

gerardo.duran-martin@eng.ox.ac.uk
{leandro.sanchezbetancourt,alvaro.cartea}@maths.ox.ac.uk
kpmurphy@google.com

## Abstract

We introduce scalable algorithms for online learning of neural network parameters and Bayesian sequential decision making. Unlike classical Bayesian neural networks, which induce predictive uncertainty through a posterior over model parameters, our methods adopt a predictive-first perspective based on martingale posteriors. In particular, we work directly with the one-step-ahead posterior predictive, which we parameterize with a neural network and update sequentially with incoming observations. This decouples Bayesian decision-making from parameter-space inference: we sample from the posterior predictive for decision making, and update the parameters of the posterior predictive via fast, frequentist Kalman-filter-like recursions. Our algorithms operate in a fully online, replay-free setting, providing principled uncertainty quantification without costly posterior sampling. Empirically, they achieve competitive performance–speed trade-offs in non-stationary contextual bandits and Bayesian optimization, offering 10–100 times faster inference than classical Thompson sampling while maintaining comparable or superior decision performance.

## 1   Introduction

In various sequential decision-making problems, such as Bayesian optimization and contextual bandits, uncertainty quantification and efficient belief updates are essential for balancing exploration and exploitation. A prominent Bayesian approach is Thompson sampling (TS) [52], which samples from the posterior over model parameters and evaluates the corresponding predictive function (e.g., reward or surrogate) for decision-making.

Various works combine Bayesian uncertainty with the expressiveness of neural networks for sequential decision making [5, 51]. These methods, often called Bayesian neural networks (BNNs), typically approximate the posterior over model parameters (e.g., via variational inference) and then use this posterior for downstream tasks. However, BNNs often rely on misspecified priors and likelihoods [33], which often degrade predictive performance [60] and result in slower inference than non-Bayesian alternatives [35].

These limitations are especially problematic in online settings where updates and decisions must be made quickly, such as in recommender systems, adaptive control, financial forecasting, and large-scale Bayesian optimization [55, 43, 45, 6, 59].

39th Conference on Neural Information Processing Systems (NeurIPS 2025).

To address these problems, we present a class of algorithms for Bayesian sequential decision making inspired by the predictive-first philosophy of the martingale posterior framework [20, 26]. Our approach operates in a fully online setting: the one-step-ahead posterior predictive is parameterized by a neural network, updated directly from real observations via frequentist Kalman filter-style recursions, and sampled once per step for decision making. This avoids the need for costly posterior inference, while still supporting uncertainty and sampling-based sequential decision making.

The parameters defining the predictive density are updated with a single-pass frequentist Kalman filter-style methods, which resemble online natural gradient steps [48] and require no replay buffer (copies of past observations, which one typically uses for multiple inner-iterations). We explore three structured covariance strategies: LRKF applies low-rank updates to all layers; HiLoFi applies a full-rank update to the last layer and a low-rank update to the hidden layers; LoLoFi applies low-rank updates to both. HiLoFi and LoLoFi are inspired by last-layer Bayesian methods [25], and offer different tradeoffs between computational efficiency and expressiveness.

Decision making is performed by sampling directly from the posterior predictive, resulting in 10 to 100 times faster inference than that of classical TS, even when compared with structured posteriors that use diagonal plus low-rank covariances [8]. Our algorithms scale to million-parameter networks, maintain constant-time updates, and operate in fully online, streaming-compatible settings. In addition, our methodology supports any acquisition strategy, such as classical TS or expected improvement.

In summary, our probabilistic method for training neural networks enjoys the following properties: (i) efficient closed-form updates (no Monte Carlo sampling required); (ii) sample efficiency through low-rank Kalman filter-like updates which do not require the specification of a posterior density; (iii) closed-form uncertainty-aware predictions via the posterior predictive.

We evaluate the performance of our methods across a range of sequential decision-making tasks. In non-stationary neural contextual bandits, our methods achieve the highest performance at the lowest computational cost compared to that of baselines. In stationary settings such as Bayesian optimization, our approach matches the performance of replay-buffer methods while achieving Pareto-efficient tradeoffs between runtime and performance. Our code is available at https://github.com/gerdm/martingale-posterior-neural-networks.

A complete summary of the notation used throughout the paper is in Appendix A.

## 2  Problem statement

We consider a sequential-decision-making agent that at time $t$ observes a context-action pair $\boldsymbol{x}_t = (\boldsymbol{c}_t, \boldsymbol{a}_t)$ with context $\boldsymbol{c}_t \in \mathbb{R}^{D_c}$ (possibly empty) and action $\boldsymbol{a}_t \in \mathcal{A} \subseteq \mathbb{R}^{D_a}$. The environment returns a reward (or observation) according to

$$\boldsymbol{y}_t \sim p_{\text{env}}(\cdot \,|\, \boldsymbol{x}_t), \qquad \boldsymbol{y}_t \in \mathbb{R}^{D_{\boldsymbol{y}}}, \tag{1}$$

which depends only on the context-action pair $\boldsymbol{x}_t$. We use the notation $y_t$ when the outcome is scalar ($D_{\boldsymbol{y}} = 1$) and $\boldsymbol{y}_t$ when it is vector-valued ($D_{\boldsymbol{y}} > 1$).

This formulation covers, e.g., Bayesian optimization ($\boldsymbol{c}_t$ empty and $\boldsymbol{a}_t$ the query location), multi-armed bandits ($\boldsymbol{c}_t$ empty and $\boldsymbol{a}_t$ the chosen arm), and contextual bandits ($\boldsymbol{c}_t$ non-empty and $\boldsymbol{a}_t$ the chosen arm). It also serves as a proxy representation for multi-armed bandits by letting the components of $\boldsymbol{y}_t$ encode per-arm signals or auxiliary outcomes.

We write $\mathcal{D}_t = (\boldsymbol{x}_t, \boldsymbol{y}_t)$ for a datapoint and $\mathcal{D}_{1:t} = \{\mathcal{D}_1, \ldots, \mathcal{D}_t\}$ for the dataset. Here, we assume that $p_{\text{env}}$ is unknown, but we approximate it through the observation model

$$\boldsymbol{y}_t = f(\boldsymbol{\theta}_t, \boldsymbol{x}_t) + \boldsymbol{e}_t, \tag{2}$$

with $\boldsymbol{e}_t$ a zero-mean random vector with covariance matrix $\mathbf{R}_t \in \mathcal{M}_{D_{\boldsymbol{y}}}^{++}(\mathbb{R})$, $f$ a neural network, and $\boldsymbol{\theta}_t \in \mathbb{R}^{D_{\boldsymbol{\theta}}}$ unknown time-varying model parameters.

The agent maintains a belief state $\boldsymbol{b}_t$ that summarizes its knowledge of the environment after observing $\mathcal{D}_{1:t}$. This belief defines a parametric approximation to the one-step-ahead posterior predictive

$$p_{\boldsymbol{b}_t}(\boldsymbol{y}_{t+1} \,|\, \boldsymbol{x}_{t+1}) \approx p(\boldsymbol{y}_{t+1} \,|\, \boldsymbol{x}_{t+1}, \mathcal{D}_{1:t}). \tag{3}$$

Our objective is to maintain $b_t$ via efficient, single-pass frequentist updates and to use the predictive $p_{b_t}(y_{t+1} \mid x_{t+1})$ for probabilistic, sample-based sequential decision making. In this sense, we combine frequentist recursive parameter estimation with Bayesian decision making via the posterior predictive.

# 3 Background

## 3.1 Tools for sequential decision making

A central challenge in sequential decision making is the exploration-exploitation tradeoff, where an agent balances whether to use its current knowledge of the environment (exploitation) or to acquire new information to improve future decisions (exploration) [3].

For example, when solving Bayesian neural contextual bandit problems [51] with the classical Thompson sampling (TS) algorithm [52], an agent samples a parameter vector from the posterior distribution (or an approximation [50]), evaluates the reward function under this sampled parameter for each action, and selects the action with the highest sampled reward.

Here, we propose a novel approach that avoids sampling from (high-dimensional) parameter posteriors often found in Bayesian neural networks. Instead, we work directly with the posterior predictive distribution defined by the current belief state $b_t$. For each candidate action, we sample a possible outcome from (3) and then choose the action with the highest sampled reward. The belief $b_t$ is subsequently updated via frequentist recursions (Section 3.2).

Table 1 highlights the key differences between our predictive sampling approach and classical TS. The essential distinction is whether uncertainty is represented in parameter space (TS) or directly in outcome space (our approach). Algorithm 1 shows our predictive sampling procedure for contextual

| Step | Predictive sampling (Our approach) | Classical TS |
|---|---|---|
| Sample | **Posterior predictive at each action** | **Posterior over model parameters** |
| Evaluate | **N/A** | **Sampled parameters on reward function** |
| Select action | Argmax over sampled rewards | Argmax over sampled rewards |
| Get reward | $y_{t+1} \sim p_{\text{env}}(\cdot \mid x_{t+1})$ | $y_{t+1} \sim p_{\text{env}}(\cdot \mid x_{t+1})$ |
| Update belief | **Frequentist update** (e.g., Algorithm 2) | **Bayesian posterior update over parameters** |

Table 1: Comparison between Bayesian predictive sampling (our approach) and classical Thompson sampling. Differences highlighted in bold.

bandits. At each decision-making step, the agent samples rewards from the predictive distribution for each action, selects the action with the highest sampled reward, observes the reward from the environment, and updates its belief state.

---

**Algorithm 1** Predictive sampling for sequential decision making in contextual bandits.

---

**Require:** $b_t$ // belief state obtained at time $t$
**Require:** $c_{t+1}$ // context
**Require:** $p_b(\cdot \mid x)$ // posterior predictive density parameterized by $b$
 1: **for all** $a \in \mathcal{A}$ **do**
 2:    $\hat{x} \leftarrow (c_{t+1}, a)$
 3:    $\hat{y}_{t,a} \sim p_{b_t}(\cdot \mid \hat{x})$
 4: **end for**
 5: $a_{t+1} \leftarrow \arg\max_{a \in \mathcal{A}} \hat{y}_{t+1,a}$
 6: $x_{t+1} \leftarrow (c_{t+1}, a_{t+1})$
 7: $y_{t+1} \sim p_{\text{env}}(\cdot \mid x_{t+1})$ // observe outcome
 8: $b_{t+1} \leftarrow \texttt{Update}(b_t, y_{t+1}, x_{t+1})$ // e.g, Algorithm 2
 9: **Return** $(b_{t+1}, y_{t+1})$ // belief and reward

---

## 3.2 Extended Kalman filtering for online learning and sequential decision making

To maintain and update the belief state $b_t$, we adapt ideas from the frequentist perspective of the extended Kalman filter (EKF). Here, $b_t = (\theta_{t|t}, \Sigma_t)$, $\theta_{t|t} = A_t \, y_{1:t}$, $A_t = \arg\min_A \mathbb{E}[\|\theta_t -$

$\mathbf{A}\,\boldsymbol{y}_{1:t}\|_2^2]$ is the best linear unbiased predictor (BLUP) of $\boldsymbol{\theta}_t$, and $\boldsymbol{\Sigma}_t = \mathrm{Var}(\boldsymbol{\theta}_t - \boldsymbol{\theta}_{t|t})$ is the error-variance-covariance (EVC) matrix of the estimate. The initial belief state $\boldsymbol{b}_0 = (\boldsymbol{\theta}_{0|0}, \boldsymbol{\Sigma}_0)$ is given.

Next, we partition the neural network parameters into last-layer weights and hidden-layer weights: $\boldsymbol{\theta}_t = (\boldsymbol{\ell}_t, \boldsymbol{h}_t)$, with $\boldsymbol{\ell}_t \in \mathbb{R}^{D_\ell}$ (last layer) and $\boldsymbol{h}_t \in \mathbb{R}^{D_h}$ (hidden layers) so that $\boldsymbol{\theta}_{t|t} = (\boldsymbol{\ell}_{t|t}, \boldsymbol{h}_{t|t})$, and $f(\boldsymbol{\ell}_t, \boldsymbol{h}_t, \boldsymbol{x}_t) = \boldsymbol{\ell}_t^\mathsf{T}\,\phi(\boldsymbol{x}_t, \boldsymbol{h}_t) \in \mathbb{R}$, where $\phi(\boldsymbol{x}_t, \boldsymbol{h}_t)$ is the hidden-layer feature map of the neural network.

To obtain closed-form updates, we linearize the neural network around the previous parameter estimate $\boldsymbol{\theta}_{t-1|t-1} = (\boldsymbol{\ell}_{t-1|t-1}, \boldsymbol{h}_{t-1|t-1})$, and make a random-walk assumption for the model parameters $\boldsymbol{\theta}_t = \boldsymbol{\theta}_{t-1} + \boldsymbol{u}_t$, with $\mathrm{Var}(\boldsymbol{u}_t) = \mathbf{Q}_t \in \mathcal{M}_{D_\theta}^{++}(\mathbb{R})$. This assumption prevents uncertainty collapse, adaptation in non-stationary environments, and numerical stability (see e.g., [46] or Section 2.6.4 in [13]).

With the above assumptions, we obtain the linearized state-space model

$$\boldsymbol{\theta}_t = \boldsymbol{\theta}_{t-1} + \boldsymbol{u}_t, \tag{4}$$

$$\boldsymbol{y}_t = f(\boldsymbol{\ell}_{t-1|t-1}, \boldsymbol{h}_{t-1|t-1}, \boldsymbol{x}_t) + \tilde{\mathbf{L}}_t(\boldsymbol{\ell}_t - \boldsymbol{\ell}_{t-1|t-1}) + \tilde{\mathbf{H}}_t(\boldsymbol{h}_t - \boldsymbol{h}_{t-1|t-1}) + \boldsymbol{e}_t, \tag{5}$$

where $\boldsymbol{e}_t$ is a zero-mean random variable with $\mathrm{Var}[\boldsymbol{e}_t] = \mathbf{R}_t \in \mathcal{M}_{D_y}^{++}(\mathbb{R})$, $\tilde{\mathbf{L}}_t = \nabla_\ell f(\boldsymbol{\ell}_{t-1|t-1}, \boldsymbol{h}_{t-1|t-1}, \boldsymbol{x}_t) \in \mathcal{M}_{1\times 2}(\mathbb{R})$, and $\tilde{\mathbf{H}}_t = \nabla_h f(\boldsymbol{\ell}_{t-1|t-1}, \boldsymbol{h}_{t-1|t-1}, \boldsymbol{x}_t)$ are the Jacobians of the neural network w.r.t. the last and hidden layers respectively.

**Updates.** The state-space model defined by (4)-(5) allows for recursive estimation of $\boldsymbol{b}_t$ via Kalman-like updates. See Appendix B for details.

**Posterior predictive model.** At each timestep the one-step-ahead posterior predictive induced by the state-space model (4)-(5), given the belief state $\boldsymbol{b}_t$ and under the assumption $p(\boldsymbol{e}_t) = \mathcal{N}(\boldsymbol{e}_t \mid \mathbf{0}, \mathbf{R}_t)$ is

$$p_{\boldsymbol{b}_t}(\boldsymbol{y}_{t+1} \mid \boldsymbol{x}_{t+1}) = \mathcal{N}\left(\boldsymbol{y}_{t+1} \,\middle|\, f(\boldsymbol{\theta}_{t|t}, \boldsymbol{x}_{t+1}),\, \underbrace{\tilde{\mathbf{J}}_{t+1}\,\boldsymbol{\Sigma}_t\,\tilde{\mathbf{J}}_{t+1}^\mathsf{T}}_{\text{epistemic}} + \underbrace{\mathbf{R}_t}_{\text{aleatoric}}\right), \tag{6}$$

where $\tilde{\mathbf{J}}_t = \begin{bmatrix} \tilde{\mathbf{L}}_t & \tilde{\mathbf{H}}_t \end{bmatrix}$. Here, the covariance of the posterior predictive decomposes into epistemic uncertainty (from parameter uncertainty under the linearization) and aleatoric uncertainty (from observation noise $\mathbf{R}_t$). The condition $\mathbf{R}_t \in \mathcal{M}_{D_y}^{++}(\mathbb{R})$ ensures that (6) defines a proper Gaussian density.

### 3.3 Martingale posteriors

Our notion of Bayesian uncertainty is inspired by the martingale posterior framework [20], which interprets Bayesian uncertainty through missing data [26]. In the original setting, one imagines imputing an infinite sequence of future observations; statistics computed on these extended datasets form a martingale, and under mild conditions they converge.

However, in sequential decision making, however, given the context-action pair $\boldsymbol{x}_{t+1}$, the only missing data, is the next observation $\boldsymbol{y}_{t+1}$ from the environment. In our case, uncertainty is quantified to guide decisions: $\boldsymbol{y}_{t+1}$ (or a partial observation of it, in the bandit setting) is always revealed before the next step. Thus, rather than simulating an infinite sequence of future steps, we focus entirely on the one-step-ahead posterior predictive $p_{\boldsymbol{b}_t}(\boldsymbol{y}_{t+1} \mid \boldsymbol{x}_{t+1})$.

## 4 Method

The standard EKF strategy for online learning presented in Section 3.2 does not scale to large neural networks because the memory cost is $O(D_{\boldsymbol{\theta}}^2)$ and the compute cost is $O(D_{\boldsymbol{\theta}}^3)$. A number of approaches have been proposed to tackle this issue (see Section 6 for details). Our method leverages the frequentist perspective of filtering to derive low-rank updates and the Bayesian interpretation of filtering to make sequential decisions through the posterior predictive.

## 4.1 Algorithm

Consider the linearized model (5), with initial conditions

$$\mathbb{E}[\boldsymbol{\ell}_0] = \boldsymbol{\ell}_{0|0}, \quad \mathbb{E}[\boldsymbol{h}_0] = \boldsymbol{h}_{0|0}, \quad \mathrm{Var}(\boldsymbol{\ell}_0) = \hat{\boldsymbol{\Sigma}}_{\boldsymbol{\ell},0}, \quad \mathrm{Var}(\boldsymbol{h}_0) = \mathbf{C}_0^\mathsf{T} \mathbf{C}_0, \tag{7}$$

for known $\mathbf{C}_0 \in \mathcal{M}_{d_{\boldsymbol{h}} \times D_{\boldsymbol{h}}}(\mathbb{R})$, $\boldsymbol{h}_{0|0} \in \mathbb{R}^{D_{\boldsymbol{h}}}$, $\hat{\boldsymbol{\Sigma}}_{\boldsymbol{\ell},0} \in \mathcal{M}_{D_{\boldsymbol{\ell}}}^{++}(\mathbb{R})$, and $\boldsymbol{\ell}_{0|0} \in \mathbb{R}^{D_{\boldsymbol{\ell}}}$. Next, assume that for $t \geq 1$, the last layer parameters $\boldsymbol{\ell}_{1:t}$ and the hidden layer parameters $\boldsymbol{h}_{1:t}$ follow the dynamics

$$\boldsymbol{\ell}_t = \boldsymbol{\ell}_{t-1} + \boldsymbol{u}_{\boldsymbol{\ell},t}, \qquad \boldsymbol{h}_t = \boldsymbol{h}_{t-1} + \boldsymbol{u}_{\boldsymbol{h},t}, \tag{8}$$

where $\boldsymbol{u}_{\boldsymbol{\ell},1:t}$ and $\boldsymbol{u}_{\boldsymbol{h},1:t}$ are zero-mean independent noise variables with covariance matrices $\mathbf{Q}_{\boldsymbol{\ell},t} = q_{\boldsymbol{\ell},t} \mathbf{I}_{D_{\boldsymbol{\ell}}}$ and $\mathbf{Q}_{\boldsymbol{h},t} = q_{\boldsymbol{h},t} \mathbf{I}_{D_{\boldsymbol{h}}}$, with $q_{\boldsymbol{\ell},t} \geq 0$ and $q_{\boldsymbol{h},t} \geq 0$. Our method maintains a belief state for the state-space model represented by $\boldsymbol{b}_t = (\boldsymbol{\ell}_{t|t}, \boldsymbol{h}_{t|t}, \hat{\boldsymbol{\Sigma}}_{\boldsymbol{\ell},t}^{1/2}, \mathbf{C}_t)$, where $\boldsymbol{\ell}_{t|t} \in \mathbb{R}^{D_{\boldsymbol{\ell}}}$ is the estimate for model parameters in the last layer, $\boldsymbol{h}_{t|t} \in \mathbb{R}^{D_{\boldsymbol{h}}}$ is the estimate for model parameters in the hidden layers, $\hat{\boldsymbol{\Sigma}}_{t-1}^{1/2} \in \mathcal{M}_{D_{\boldsymbol{\ell}}}^{++}(\mathbb{R})$ is the Cholesky EVC factor for the covariance of the last layer, and $\mathbf{C}_t \in \mathcal{M}_{d_{\boldsymbol{h}} \times D_{\boldsymbol{h}}}(\mathbb{R})$ is the low-rank EVC factor for the hidden layers.

**Updates.** Given $\boldsymbol{b}_{t-1}$, the updated $\boldsymbol{\theta}_{t|t} = (\boldsymbol{\ell}_{t|t}, \boldsymbol{h}_{t|t})$ follows directly from the EKF equations which takes the form $\boldsymbol{\theta}_{t|t} = \boldsymbol{\theta}_{t-1|t-1} + \mathbf{K}_t \boldsymbol{\epsilon}_t$, where $\boldsymbol{\epsilon}_t = \boldsymbol{y}_t - \hat{\boldsymbol{y}}_t$ is the error or innovation term, and $\mathbf{K}_t$ is the gain matrix. Because of the block-diagonal assumption, it can be shown that $\mathbf{K}_t$ is decoupled into last-layer-only and a hidden-layers-only submatrices.

Next, applying the EKF EVC update to $\boldsymbol{b}_{t-1}$ yields the dense matrix $\boldsymbol{\Sigma}_t \in \mathcal{M}_{D_{\boldsymbol{\theta}}}^{+}(\mathbb{R})$ which couples the effects from the last and hidden layers. To obtain a block-diagonal matrix with components $(\hat{\boldsymbol{\Sigma}}_{\boldsymbol{\ell},t}^{1/2}, \mathbf{C}_t)$, we consider a two-stage approximation. The first stage replaces $\boldsymbol{\Sigma}_t$ with a surrogate covariance matrix $\tilde{\boldsymbol{\Sigma}}_t$ that avoids the interaction between the hidden layer parameters and the synthetic dynamics $q_{\boldsymbol{h},t}$; the second stage approximates the surrogate matrix with the best block-diagonal approximation whose first block (for last-layer parameters) are of rank $D_{\boldsymbol{\ell}}$ and the second block (for hidden-layer parameters) are rank-$d_{\boldsymbol{h}}$. Finally, to maintain numerical stability and low-memory updates, we track the Cholesky factor for the last layer and a low-rank factor for the hidden layers, i.e., $\hat{\boldsymbol{\Sigma}}_t^{1/2} = \mathrm{diag}(\mathbf{C}_t, \hat{\boldsymbol{\Sigma}}_{\boldsymbol{\ell},t}^{1/2})$. In this sense, estimation of the factors for the last-layer and the hidden layer follows the sequence

$$\boldsymbol{\Sigma}_t \to \tilde{\boldsymbol{\Sigma}}_t \to \hat{\boldsymbol{\Sigma}}_t \to \hat{\boldsymbol{\Sigma}}_t^{1/2} = \mathrm{diag}(\mathbf{C}_t, \hat{\boldsymbol{\Sigma}}_{\boldsymbol{\ell},t}^{1/2}).$$

Algorithm 2 shows a single step of `HiLoFi`. The derivation is in Appendix D.1.

The following proposition shows the upper bound of the the per-step error incurred by `HiLoFi`.

**Proposition 4.1** (Covariance approximation error). *At time t, the per-step approximation error of* `HiLoFi` *under the linearized SSM* (5) *is bounded by*

$$\|\boldsymbol{\Sigma}_{t|t} - \hat{\boldsymbol{\Sigma}}_{t|t}\|_\mathrm{F} \leq q_{\boldsymbol{h},t}\, \mathbf{E}_{\boldsymbol{h},\mathrm{surr}} + q_{\boldsymbol{\ell},t}\, \mathbf{E}_{\boldsymbol{\ell},\mathrm{surr}} + \sqrt{\sum_{k=d_{\boldsymbol{h}}+1}^{D_{\boldsymbol{h}}} \lambda_k^2 + 2\|\mathbf{K}_{\boldsymbol{\ell},t}\mathbf{R}_t\mathbf{K}_{\boldsymbol{h},t}^\mathsf{T}\|_\mathrm{F}^2}, \tag{9}$$

*where* $\mathbf{E}_{\boldsymbol{h},\mathrm{surr}} = (2\|\mathbf{K}_{\boldsymbol{h},t}\tilde{\mathbf{H}}_t\|_\mathrm{F} + \|\mathbf{K}_{\boldsymbol{h},t}\tilde{\mathbf{H}}_t\|_\mathrm{F}^2)$, $\mathbf{E}_{\boldsymbol{\ell},\mathrm{surr}} = \|(\mathbf{I} - \mathbf{K}_{\boldsymbol{\ell},t}\tilde{\mathbf{L}}_t)(\mathbf{I} - \mathbf{K}_{\boldsymbol{\ell},t}\tilde{\mathbf{L}}_t)^\mathsf{T}\|_\mathrm{F}$, *and* $\{\lambda_k\}_{k=d_{\boldsymbol{h}}+1}^{D_{\boldsymbol{h}}}$ *are smallest* $(D_{\boldsymbol{h}} - d_{\boldsymbol{h}})$ *eigenvalues of* $\tilde{\boldsymbol{\Sigma}}_t$.

*Remark* 4.2. Proposition 4.1 shows that the per-step error in the `HiLoFi` approximation is bounded by: (i) the neglected terms by each surrogate matrix, (ii) the low-rank approximation for the covariance of the hidden layers, and (iii) the cross-terms from the blocks in the off-diagonal Note, however, that the bound does not account for error due to linearization of the neural network. One could minimize the above lower bound by setting $q_{\boldsymbol{h},t} = q_{\boldsymbol{\ell},t} = 0$. However, this may result in lower per-step performance. See Appendix G.1.2 for an example of the main sources of error. See Appendix E.4 for a proof.

**Posterior predictive model.** After every update, the posterior predictive model used for sequential decision making is

$$p_{\boldsymbol{b}_t}(\boldsymbol{y}_{t+1} \mid \boldsymbol{x}_{t+1}) = \mathcal{N}\left(\boldsymbol{y}_t \mid f(\boldsymbol{\ell}_{t|t}, \boldsymbol{h}_{t|t}, \boldsymbol{x}_{t+1}),\ \tilde{\mathbf{L}}_{t+1}\boldsymbol{\Sigma}_{\boldsymbol{\ell},t}\tilde{\mathbf{L}}_{t+1}^\mathsf{T} + \tilde{\mathbf{H}}_{t+1}\boldsymbol{\Sigma}_{\boldsymbol{h},t}\tilde{\mathbf{H}}_{t+1}^\mathsf{T} + \mathbf{R}_{t+1}\right). \tag{10}$$

**Algorithm 2** Single update step of `HiLoFi`. The notation $\mathcal{Q}_R(\mathbf{B}_1, \ldots, \mathbf{B}_k)$ denotes the Cholesky factorization of the matrix $\sum_{k=1}^{K} \mathbf{B}_k^\mathsf{T} \mathbf{B}_k$, where $\mathbf{B}_k$ is an upper-triangular Cholesky factor; see Appendix C.1 for details. The function $\mathcal{P}_d(\mathbf{A}_1, \ldots, \mathbf{A}_K)$ returns a $d \times D$ matrix which is the best rank-$d$ approximation of the matrix $\sum_{k=1}^{K} \mathbf{A}_k^\mathsf{T} \mathbf{A}_k$; see Appendix C.2 for details.

---

**Require:** $\mathbf{R}_t$ `// measurement variance`
**Require:** $(q_{\ell,t}, q_{h,t})$ `// dynamics covariance for last layer and hidden layers`
**Require:** $(\boldsymbol{y}_t, \boldsymbol{x}_t)$ `// observation and context-action pair`
**Require:** $b_{t-1} = \left( \boldsymbol{\ell}_{t-1|t-1}, \boldsymbol{h}_{t-1|t-1}, \hat{\boldsymbol{\Sigma}}_{\ell,t-1}^{1/2}, \mathbf{C}_{t-1} \right)$ `// previous belief`
    `// predict step`
1: $\hat{\boldsymbol{y}}_t \leftarrow f(\boldsymbol{\ell}_{t-1|t-1}, \boldsymbol{h}_{t-1|t-1}, \boldsymbol{x}_t)$
2: $\tilde{\mathbf{L}}_t = \nabla_\ell f(\boldsymbol{\ell}_{t-1|t-1}, \boldsymbol{h}_{t-1|t-1}, \boldsymbol{x}_t)$
3: $\tilde{\mathbf{H}}_t = \nabla_h f(\boldsymbol{\ell}_{t-1|t-1}, \boldsymbol{h}_{t-1|t-1}, \boldsymbol{x}_t)$
    `// innovation (one-step-ahead error) and Cholesky innovation variance`
4: $\boldsymbol{\epsilon}_t \leftarrow \boldsymbol{y}_t - \hat{\boldsymbol{y}}_t$
5: $\mathbf{S}_t^{1/2} = \mathcal{Q}_R \left( \hat{\boldsymbol{\Sigma}}_{\ell,t-1}^{1/2} \tilde{\mathbf{L}}_t^\mathsf{T}, \sqrt{q_{\ell,t}} \tilde{\mathbf{L}}_t^\mathsf{T}, \mathbf{C}_{t-1} \tilde{\mathbf{H}}_t^\mathsf{T}, \sqrt{q_{h,t}} \tilde{\mathbf{H}}_t^\mathsf{T}, \mathbf{R}_t^{1/2} \right)$,
    `// gain for hidden layers`
6: $\mathbf{V}_{h,t} \leftarrow \mathbf{S}_t^{-1/2} \mathbf{S}_t^{-\mathsf{T}/2} \tilde{\mathbf{H}}_t$
7: $\mathbf{K}_{h,t}^\mathsf{T} \leftarrow \mathbf{V}_{h,t} \mathbf{C}_{t-1} \mathbf{C}_{t-1}^\mathsf{T} + q_{h,t} \mathbf{V}_{h,t}$
    `// gain for last layer`
8: $\mathbf{V}_{\ell,t} \leftarrow \mathbf{S}_t^{-1/2} \mathbf{S}_t^{-\mathsf{T}/2} \tilde{\mathbf{L}}_t$
9: $\mathbf{K}_{\ell,t}^\mathsf{T} \leftarrow \mathbf{V}_{\ell,t} \boldsymbol{\Sigma}_{\ell,t|t-1} + q_{\ell,t} \mathbf{V}_{\ell,t}$
    `// mean update step`
10: $\boldsymbol{h}_{t|t} \leftarrow \boldsymbol{h}_{t-1|t-1} + \mathbf{K}_{h,t} \boldsymbol{\epsilon}_t$
11: $\boldsymbol{\ell}_{t|t} \leftarrow \boldsymbol{\ell}_{t-1|t-1} + \mathbf{K}_{\ell,t} \boldsymbol{\epsilon}_t$
    `// EVC low-rank and Cholesky update step`
12: $\mathbf{C}_t \leftarrow \mathcal{P}_{d_h, +q_{h,t}} \left( \mathbf{C}_{t-1} \left( \mathbf{I} - \mathbf{K}_{h,t} \tilde{\mathbf{H}}_t \right)^\mathsf{T}, \mathbf{R}_t^{1/2} \mathbf{K}_{h,t}^\mathsf{T} \right)$
13: $\hat{\boldsymbol{\Sigma}}_{\ell,t}^{1/2} \leftarrow \mathcal{Q}_R \left( \hat{\boldsymbol{\Sigma}}_{\ell,t-1}^{1/2}, \left( \mathbf{I} - \mathbf{K}_{\ell,t} \tilde{\mathbf{L}}_t \right)^\mathsf{T}, \mathbf{R}_t^{1/2} \mathbf{K}_{\ell,t}^\mathsf{T} \right)$
14: **Return** $b_t = \left( \boldsymbol{\ell}_{t|t}, \boldsymbol{h}_{t|t}, \hat{\boldsymbol{\Sigma}}_{\ell,t}^{1/2}, \mathbf{C}_t \right)$ `// updated belief`

---

The computational complexity of `HiLoFi` is $O(D_\ell (D_\ell + D_y)^2 + D_h (d_h + D_y)^2)$. A full breakdown of the computational complexity and its relationship to other methods is in Appendix D.2.

**Variants.** Whenever the output layer or the input to the last layer is high dimensional, we perform a low-rank approximation to the last-layer to reduce computational costs. We call this variant `LoLoFi`. Its computational complexity is $O(D_\ell{}^3 + D_h d_h{}^2)$. See Appendix F.1 for details.

A purely low-rank version of our method, which we call `LRKF`, is discussed in Appendix F.2. This method uses a single low rank covariance matrix approximation for all parameters in the neural network. An error analysis of `LRKF` in the linear setting is presented in Appendix F.2.1. In the experiments, we find that this approach performs worse than that of `HiLoFi`, which is not surprising. It also has lower performance than `LoLoFi` (at least on the contextual bandit problem in section 5.2), presumably because modeling the last layer separately allows us to use a different subspace for the low rank approximation of the output layer weights.

## 5 Experiments

In this section, we evaluate our method. First, we consider a one-dimensional example "in-between" uncertainty, to illustrate the behavior of the method. We then compare its performance to that of other methods across the following sequential decision-making tasks:

(I) **MNIST for classification as a bandit problem**: This is an online high-dimensional classification problem, which we study as a contextual bandit problem.

(II) **Recommender systems**: This is a challenging real-world problem with non-stationary data. Here, tackling exploration-exploitation tradeoff is key.

(III) **Bayesian optimization**: The goal of this task is to find the maximum of an unknown blackbox function $g : [0,1]^{D_x} \to \mathbb{R}$. This is a static inference problem where sample efficiency and uncertainty quantification are crucial.

For each of the above tasks, we assess the various methods on predictive performance and wall-clock time. All experiments were run on a TPU v4-8. An additional experiment demonstrating the scalability of LRKF to multi-million-parameter neural networks for online classification is provided in Appendix G.2.

We consider the following variants of our framework. **High-rank low-rank filter** (HiLoFi): Our main method that uses full rank for last layer and low rank for hidden layers. **Low-rank low-rank filter** (LoLoFi): A version of our method that uses low-rank matrices for last and hidden layers. We only consider LoLoFi in task (II), due to the relatively large dimension of the last layer. **Low-rank square-root Kalman Filter** (LRKF): This is a special case of our method, described in Appendix F.2, that uses a single low-rank covariance for all the parameters. This illustrates the gains from modeling the final layer separately.

As existing baselines, we consider the following. **Diagonal plus low-rank precision** (LoFi) [8]: a fully online method that models the precision matrix using a low-rank plus diagonal matrix, and uses the same linearization scheme as us. **Gaussian processes** (GP) [62]: a standard approach to modeling predictive uncertainty. We only include GPs in the low-dimensional tasks in (III) because of scalability limitations. **Variational Bayesian last layer** (VBLL) [5]: a partially Bayesian method that requires access to the full dataset (or a replay buffer), and performs multiple optimization steps per update. See Appendix F.3 for details. **Online variational Bayesian last-layer** (OnVBLL): a straightforward modification of the VBLL method with a first-in-first-out (FIFO) buffer and no regularization. **Last-layer Laplace approximation with FIFO buffer** (LLL) [11]: a method that uses a Laplace approximation on the last layer, with a FIFO replay buffer and multiple inner updates per time step. Unless otherwise stated, we train the parameters of the neural network with the AdamW optimization algorithm [44].

## 5.1 In-between uncertainty

**Problem description.** In this experiment, we test the ability of HiLoFi to capture the *in-between uncertainty* [21] after a single pass of the data. This concept refers to how well a model captures uncertainty in regions of input space that lie between, or away from, observed data. Intuitively, we expect uncertainty to be higher in such unexplored areas. This behavior is important for Bayesian decision-making problems for effective balancing of exploration and exploitation. We consider the one-dimensional dataset introduced in [58].

**Model.** We use an MLP with 4 hidden layers and 128 units per layer, and we employ an ELU activation function.

**Results.** Figure 1 shows the evolution of the posterior predictive mean and the posterior predictive variance as a function of the number of seen observations. We see that HiLoFi behaves in an intuitively sensible way, showing more uncertainty away from the data, just like a Gaussian process. See Appendix G.5 for the performance of other methods (specifically LRKF and VBLL) on this problem, as well an ablation study that illustrates the effect of changing rank.

## 5.2 MNIST

**Problem description.** We consider the MNIST contextual bandit task introduced by [49], where the agent is presented with an image and must choose one of ten possible classes as an action, and then gets a binary feedback, based on whether its prediction was correct or not; we refer to this as the incomplete information case. An additional experiment on online classification is in Appendix G.1.1.

**Model.** The predictive model (for the 10 label logits or the per-action rewards) is represented using a modified LeNet5 CNN architecture [39] with ELU activation function.

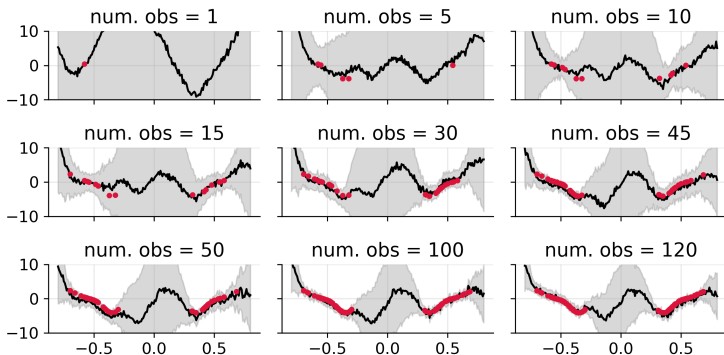

Figure 1: In-between uncertainty induced by `HiLoFi` as a function of the processed observations.

**Exploration Methods.** We consider two exploration strategies: Classical TS and our proposed predictive-sampling (PS) as outlined in Table 1. Additional results comparison TS, PS, and $\epsilon$-greedy are in Appendix G.1.3.

**Inference methods.** We compare `LLL`, `LRKF`, `LoFi`, and `HiLoFi`. For `LLL`, we use the online last-layer variant from [11], without hyperparameter re-optimization at each timestep. Choice for hyperparameters are detailed in Appendix G.1.3.

**Bandit results.** Figure 2 shows the average cumulative reward over 10 runs of the contextual bandit version of MNIST. We observe that the top three performing methods are `HiLoFi`, `LoLoFi`, and `LRKF`. This outcome is expected because `HiLoFi` employs a full-rank covariance matrix in the last layer, which is better than a low-rank approximation of the covariance matrix associated with the last layer (`LoLoFi`), or a low-rank covariance to model all dependencies across layers (`LRKF`). For further results showing regret and comparison using $\varepsilon$-greedy, see Appendix G.1.3.

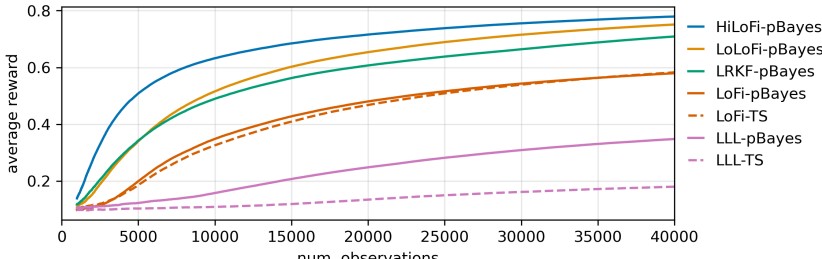

Figure 2: Cumulative average reward for the bandit MNIST problem.

Next, Table 5.2 reports the average cumulative reward and total runtime (over 10 trials) for each method, comparing posterior predictive sampling (pBayes) with Thompson sampling (TS). Overall,

| Agent | Reward (pBayes) | Reward (TS) | Time (pBayes) | Time (TS) |
|---|---|---|---|---|
| HiLoFi | **31,167.7** | – | 8.44 | – |
| LoLoFi | 30,038.5 | – | 3.67 | – |
| LRKF | 28,354.9 | – | 2.51 | – |
| LoFi | 23,184.5 | 23,304.3 | 4.76 | 38.90 |
| LLL | 13,922.3 | 7,201.9 | 1.36 | 191.55 |

Table 2: Performance comparison of Bayesian decision-making strategies: posterior predictive sampling (pBayes) vs. Thompson sampling (TS). Rewards are cumulative and averaged over 10 runs; time is in minutes.

pBayes is consistently faster across all methods. For the two agents where both approaches are applicable (`LoFi` and `LLL`), the average cumulative reward is nearly identical for `LoFi`, while pBayes clearly outperforms TS for `LLL`. However, TS incurs a much higher runtime due to the need to sample repeatedly from the high-dimensional posterior over model parameters, whereas pBayes only requires sampling in the comparatively low-dimensional outcome space (one dimension per

arm/class). Finally, as expected, `HiLoFi` achieves the best performance, followed by `LoLoFi` and then `LRKF`. This performance hierarchy comes at a cost in runtime: `HiLoFi` is slower than `LoLoFi`, which in turn is slower than `LRKF`.

## 5.3 Recommender systems

**Problem description.** We study the performance of the methods on the Kuairec dataset of [22], which is derived from a real-world, non-stationary, video recommender system. This dataset is used to study non-stationary contextual bandits in [65].

**Dataset.** We group rows in the dataset (for a given user) in blocks of the 10 next videos that the user saw. For the features, we consider the $\log(1 + x)$ transform of like count, share count, play count, comment count, complete play count, follow count, reply comment count, and download count. For the target variable, we use the $\log(1 + x)$ transform of the watch ratio $[0, \infty)$.

**Model.** The reward model is a neural network with an embedding layer (one per video/arm), and a dense layer for all features. We join the embedding layer and the dense layer, and we then consider a three-hidden-layer neural network (50 units per layer, ELU activations), and linear output unit.

**Algorithm.** We perform sequential Bayesian inference using the relevant method, and then use TS to choose the action at each step (see Section 3.1). The choice of hyperparameters is in G.3.

**Results.** Figure 3 shows the average daily reward (left panel) and the running time of each method (right panel). We observe that `HiLoFi` has the highest average daily reward with second lowest running time. On average, `LRKF` and `LoFi` have similar performance, however, `LoFi` has a higher running time. Next, `VBLL` has higher average daily reward than `OnVBLL`, but it is slower. For an in-depth analysis of the results see Appendix G.3.

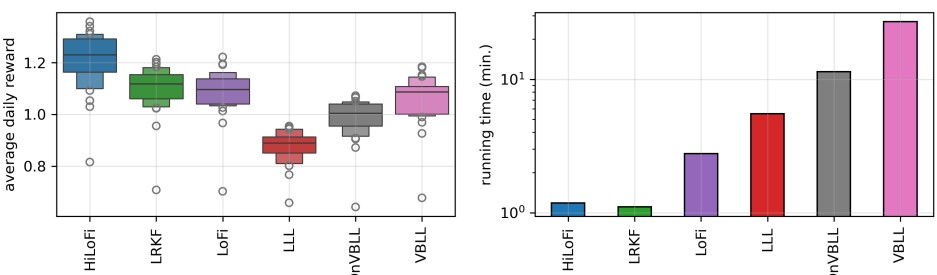

Figure 3: Recommender system results. Left: average daily reward. Right: running time.

## 5.4 Bayesian optimization

**Problem description.** Following [5], we evaluate the competing methods on classical Bayesian optimization (BO) tasks, where the goal is to maximize an unknown function $g : [0, 1]^{D_x} \to \mathbb{R}$ with a surrogate model trained sequentially on past observations.

**Model.** We employ a three-layer MLP (180 units per layer, ELU activations) as the surrogate for all neural-based methods. For the GP baseline, we use a Matérn-5/2 kernel with lengthscale 0.1 and a 20-point buffer to reduce computational cost. While GP performs better when its hyperparameters are tuned during training [28], this incurs much higher computational costs, so we do not pursue it here. The choice of hyperparameters is detailed in Appendix G.4.

**Datasets.** We consider seven benchmark functions commonly used in BO [5]: Ackley (2D, 5D, 10D, 50D), Branin (2D), Hartmann (6D), and DrawNN (50D and 200D). These span a range of dimensionalities and multi-modalities, and are standard in evaluating global optimal performance. For all methods, except for DrawNN, we sample a function from the posterior predictive and evaluate it on a fixed set of candidate points generated using a Sobol sequence [54] to find its maximum. Instead, for Drawnn functions, we use projected gradient descent, implemented with the Jaxopt library [4], to optimize the sampled function directly over the continuous domain $[0, 1]$.

**Algorithm.** We perform sequential Bayesian inference using the relevant method, and then choose the next query point at each step using TS (see Section 3.1). Figure 12 in the Appendix shows the results using the expected improvement algorithm [29].

**Results.** Figure 4 (left) shows the final best value on the vertical axis (higher is better), and the amount of run time on the horizontal axis (lower is better). The number of steps for each dataset are in the x-axis in Figure 11 in Appendix G.4. Figure 4 (right) shows the tradeoff between performance (measured by rank) and compute time. Our method is on the Pareto frontier for a range of tradeoffs between time and performance; furthermore, across tasks, the performance and compute time of `HiLoFi` dominates (better performance and less compute time) those from competing methods except from VBLL (higher performance) and LRKF (lower compute time). Thus, `HiLoFi` offers strong tradeoffs between compute time and performance. For further analyses and results see Appendix G.4.

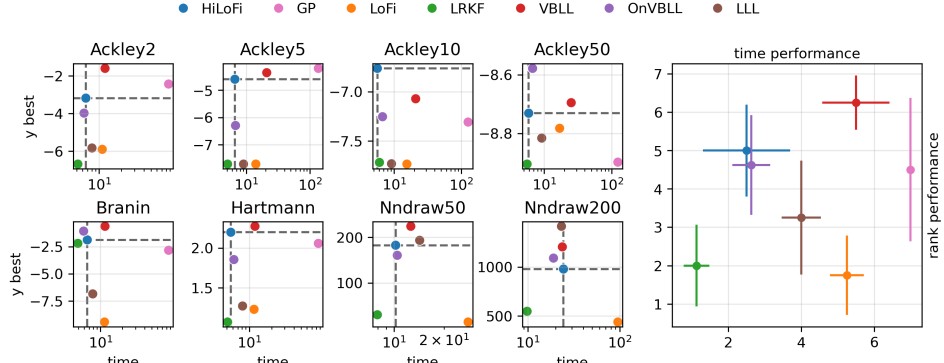

Figure 4: Left panel: Performance across all BO benchmark functions. For time, lower is better. For performance, higher is better. The dashed lines correspond to the results for our method, so methods that are above and to the left are better. Right panel: Time versus performance tradeoff plots.

# 6 Related work

We briefly review deterministic approaches for approximate online parameter inference. Several methods maintain diagonal plus low-rank (DLR) structures for efficient covariance updates, including L-RVGA [36], SLANG [47], and LoFi [8]. Low-rank Kalman filtering has also been explored in state-space models [53], though with stronger restrictions on the rank relative to measurement dimension. Beyond low-rank, FOO-VB [64] exploits Kronecker-structured approximations, while offline methods explore similar ideas for scaling BNNs [41, 56, 34]. Another direction builds on the lottery ticket hypothesis [42, 38], restricting learning to sparse or low-dimensional subspaces. Examples include the subspace neural bandit [15] and PULSE [7], which pre-train projection matrices offline before online adaptation.

# 7 Conclusion and future work

We introduced a predictive-first approach for efficient online training of neural networks, based on linearization and structured low-rank approximations of error covariances. Our three variants achieve strong performance-runtime trade-offs, with `HiLoFi` particularly effective in high-dimensional, non-stationary settings. Limitations include the lack of fixed-lag smoothing, sensitivity to linearization and outliers [14, 37], and reliance on hyperparameters. Future work includes extending our approach to fully-online reinforcement learning [18].

We view our contributions as primarily methodological, with no particular ethical concerns.

## Acknowledgements

We thank the TPU Research Cloud program for providing the compute resources used in our experiments, and James Harrison for helpful comments.

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

# Appendix

# A Notation

| Symbol | Description |
|---|---|
| $\|\cdot\|_2$ | $\ell_2$ norm. |
| $\|\cdot\|_F$ | Frobenius norm. |
| $D_{\boldsymbol{y}}, D_{\boldsymbol{x}} \in \mathbb{N}$ | Dimension of observations / inputs. |
| $D_{\boldsymbol{h}}, D_{\boldsymbol{\ell}} \in \mathbb{N}$ | Number of parameters for the hidden layer / last layer. |
| $D_{\boldsymbol{\theta}} = D_{\boldsymbol{h}} + D_{\boldsymbol{\ell}}$ | Total number of neural network parameters. |
| $\mathcal{N}(\boldsymbol{x} \,\|\, \boldsymbol{m}, \mathbf{S})$ | Multivariate Gaussian density evaluated at $\boldsymbol{x}$ with mean $\boldsymbol{m}$ and covariance matrix $\mathbf{S}$. |
| $p_{\text{env}}(\boldsymbol{y} \,\|\, \boldsymbol{x})$ | Unknown data-generating process for $\boldsymbol{y}$ conditioned on $\boldsymbol{x}$ |
| $p_{\boldsymbol{b}}(\boldsymbol{y} \,\|\, \boldsymbol{x})$ | Posterior predictive model for $\boldsymbol{y}$, conditioned on $\boldsymbol{x}$, and parameterized by $\boldsymbol{b}$. |
| $\mathcal{M}_{m \times n}(\mathbb{R})$ | Space of $m$-by-$n$ real-valued matrices. |
| $\mathcal{M}_m^{++}(\mathbb{R})$ | Space of $m$-dimensional positive definite (pd) matrices. |
| $\mathcal{M}_m^{+}(\mathbb{R})$ | Space of $m$-dimensional positive semidefinite (psd) matrices. |
| $\mathbf{I}_D$ | $D$-dimensional identity matrix. |
| $\mathbf{S}^{1/2}$ | Upper-triangular Cholesky decomposition of the pd matrix $\mathbf{S}$. |
| $\boldsymbol{x}_t = (\boldsymbol{c}_t, \boldsymbol{a}_t)$ | Inputs with context $\boldsymbol{c}_t \in \mathbb{R}^{D_c}$ (possibly empty) and action $\boldsymbol{a}_t \in \mathcal{A} \subseteq \mathbb{R}^{D_a}$. |
| $y_t \in \mathbb{R}$ | Scalar measurements, rewards, or observations obtained at timestep $t$. |
| $\boldsymbol{y}_t \in \mathbb{R}^{D_{\boldsymbol{y}}}$ | Measurements, rewards, or observations obtained at timestep $t$. |
| $\mathcal{D}_t = (\boldsymbol{x}_t, \boldsymbol{y}_t)$ | Datapoint at time $t$. |
| $\boldsymbol{u}_{1:t} = (\boldsymbol{u}_1, \ldots, \boldsymbol{u}_t)$ | Time-ordered collection of vectors $\boldsymbol{u}_k \in \mathbb{R}^{D_{\boldsymbol{u}}}$. |
| $\mathcal{D}_{1:t} = (\mathcal{D}_1, \ldots, \mathcal{D}_t)$ | Dataset at time $t$. |
| $\boldsymbol{\ell} \in \mathbb{R}^{D_{\boldsymbol{\ell}}}$ | Last layer parameters. |
| $\boldsymbol{h} \in \mathbb{R}^{D_{\boldsymbol{h}}}$ | Hidden layer parameters. |
| $\boldsymbol{\theta} = (\boldsymbol{\ell}, \boldsymbol{h}) \in \mathbb{R}^{D_{\boldsymbol{\theta}}}$ | Collection of all neural network parameters. |
| $f(\boldsymbol{\theta}, \boldsymbol{x}) = f(\boldsymbol{\ell}, \boldsymbol{h}, \boldsymbol{x}) \in \mathbb{R}^{D_{\boldsymbol{y}}}$ | Neural network output with parameters $\boldsymbol{\theta}$ and inputs $\boldsymbol{x}$. |
| $\boldsymbol{e}_t \in \mathbb{R}^{D_{\boldsymbol{y}}}$ | Zero-mean noise with covariance $\mathbf{R}_t \in \mathcal{M}_{D_{\boldsymbol{y}}}^{++}(\mathbb{R})$. |
| $\boldsymbol{u}_t \in \mathbb{R}^{D_{\boldsymbol{\theta}}}$ | Zero-mean dynamics noise with variance $\mathbf{Q}_t \in \mathcal{M}_{D_{\boldsymbol{\theta}}}^{+}(\mathbb{R})$. |
| $q_{\boldsymbol{\ell}, t} \geq 0$ | Last-layer dynamics. |
| $q_{\boldsymbol{h}, t} \geq 0$ | Hidden-layer dynamics. |
| $\boldsymbol{u}_{t, \boldsymbol{\ell}} \in \mathbb{R}^{D_{\boldsymbol{\ell}}}$ | Zero-mean dynamics noise of last layer parameters with variance $q_{\boldsymbol{\ell}, t}\, \mathbf{I}_{D_{\boldsymbol{\ell}}}$. |
| $\boldsymbol{u}_{t, \boldsymbol{h}} \in \mathbb{R}^{D_{\boldsymbol{h}}}$ | Zero-mean dynamics noise of hidden-layer parameters with variance $q_{\boldsymbol{h}, t}\, \mathbf{I}_{D_{\boldsymbol{h}}}$. |
| $\boldsymbol{\ell}_{t\|t} \in \mathbb{R}^{D_{\boldsymbol{\ell}}}$ | Frequentist estimate for $\boldsymbol{\ell}_t$ given $\mathcal{D}_{1:t}$. |
| $\boldsymbol{h}_{t\|t} \in \mathbb{R}^{D_{\boldsymbol{h}}}$ | Frequentist estimate for $\boldsymbol{h}_t$ given $\mathcal{D}_{1:t}$. |
| $\boldsymbol{\theta}_{t\|t} = (\boldsymbol{\ell}_{t\|t}, \boldsymbol{h}_{t\|t}) \in \mathbb{R}^{D_{\boldsymbol{\theta}}}$ | Frequentist estimate for $\boldsymbol{\theta}_t$ given $\mathcal{D}_{1:t}$. |
| $\boldsymbol{\Sigma}_t = \text{Var}(\boldsymbol{\theta}_t - \boldsymbol{\theta}_{t\|t})$ | Error variance-covariance matrix. |
| $\tilde{\boldsymbol{\Sigma}}_t$ | Algorithm-dependent surrogate covariance matrix to $\boldsymbol{\Sigma}_t$. |
| $\hat{\boldsymbol{\Sigma}}_t = \arg\min_{\boldsymbol{\Sigma}:\text{rank}(\boldsymbol{\Sigma})=d} \left\| \tilde{\boldsymbol{\Sigma}}_{t\|t} - \boldsymbol{\Sigma} \right\|_F^2$ | Best rank-$d$ approximation to the surrogate covariance matrix $\tilde{\boldsymbol{\Sigma}}_t$. |
| $\mathbf{C}_t \in \mathcal{M}_{d \times D}(\mathbb{R})$ | Rectangular matrix $(d \times D)$ matrix such that $\mathbf{C}_t^{\mathsf{T}}\, \mathbf{C}_t = \hat{\boldsymbol{\Sigma}}_t$ |
| $\tilde{\mathbf{L}}_{t+1} = \nabla_{\boldsymbol{\ell}} f(\boldsymbol{\ell}_{t\|t}, \boldsymbol{h}_{t\|t}, \boldsymbol{x}_{t+1})$ | Jacobian of neural network w.r.t parameters in last layer. |
| $\tilde{\mathbf{H}}_{t+1} = \nabla_{\boldsymbol{h}} f(\boldsymbol{\ell}_{t\|t}, \boldsymbol{h}_{t\|t}, \boldsymbol{x}_{t+1})$ | Jacobian of neural network w.r.t. parameter in hidden layers. |
| $\mathcal{Q}_R(\mathbf{B}_1, \ldots, \mathbf{B}_k)$ | Cholesky factorization of $\sum_{k=1}^{K} \mathbf{B}_k^{\mathsf{T}}\, \mathbf{B}_k$, where each $\mathbf{B}_k$ is an upper-triangular Cholesky factor (see Appendix C.1). |
| $\mathcal{P}_d(\mathbf{A}_1, \ldots, \mathbf{A}_K) \in \mathcal{M}_{d \times D}(\mathbb{R})$ | Best rank-$d$ approximation of $\sum_{k=1}^{K} \mathbf{A}_k^{\mathsf{T}}\, \mathbf{A}_k$, with $\mathbf{A}_k \in \mathcal{M}_{d_k \times D}(\mathbb{R})$, $d_k \leq D$. |
| $\mathcal{P}_{d,+a}(\mathbf{A}_1, \ldots, \mathbf{A}_K) \in \mathcal{M}_{d \times D}(\mathbb{R})$ | Best rank-$d$ approximation of $\sum_{k=1}^{K} \mathbf{A}_k^{\mathsf{T}}\, \mathbf{A}_k + a\, \mathbf{I}_D$, with $a > 0$ (see Appendix C.2). |

# B  Extended Kalman filtering for online learning

Here, we explain in more detail the background of the EKF for online learning presented in Section 3.2. To make this section self-contained, we repeat parts of the material introduced in that section. Let us assume that $y_t = \boldsymbol{\ell}_t^\intercal \phi(\boldsymbol{x}_t, \boldsymbol{h}_t) + \boldsymbol{e}_t$, where $\boldsymbol{e}_t$ is a zero-mean random variable with observation covariance $\mathbf{R}_t \in \mathcal{M}_{D_{\boldsymbol{y}}}^{++}(\mathbb{R})$.  Given a starting vector $\boldsymbol{\theta}_0$, we assume that model parameters $\boldsymbol{\theta}_t = (\boldsymbol{\ell}_t, \boldsymbol{h}_t) \in \mathbb{R}^{D_\theta}$ and observations $\boldsymbol{y}_t \in \mathbb{R}^{D_y}$ evolve according to the state-space model

$$
\begin{aligned}
\boldsymbol{\theta}_t &= \boldsymbol{\theta}_{t-1} + \boldsymbol{u}_t, \\
\boldsymbol{y}_t &= f(\boldsymbol{\theta}_t, \boldsymbol{x}_t) + \boldsymbol{e}_t,
\end{aligned}
\tag{11}
$$

where $\boldsymbol{u}_t \in \mathbb{R}^{D_\theta}$ is a zero-mean random vector with known dynamics covariance $\mathbf{Q}_t \in \mathcal{M}_{D_\theta}^+(\mathbb{R})$, and $\boldsymbol{e}_t \in \mathbb{R}^{D_y}$ is a zero-mean random vector with known observation covariance $\mathbf{R}_t \in \mathcal{M}_{D_{\boldsymbol{y}}}^{++}(\mathbb{R})$. As before, we assume $\mathrm{Cov}(\boldsymbol{\theta}_0, \boldsymbol{e}_t) = \mathbf{0}$ for all $t \in \mathbb{N}$ and $\mathrm{Cov}(\boldsymbol{u}_t, \boldsymbol{e}_k) = \mathbf{0}$ for all $t, k \in \mathbb{N}$.

We now consider a first-order approximation of $f$ around $\boldsymbol{\theta}_{t-1|t-1}$, that is,

$$
\boldsymbol{y}_t \approx f(\boldsymbol{\theta}_{t-1|t-1}, \boldsymbol{x}_t) + \tilde{\mathbf{J}}_t (\boldsymbol{\theta}_t - \boldsymbol{\theta}_{t-1|t-1}) + \boldsymbol{e}_t,
\tag{12}
$$

where $\boldsymbol{\theta}_{t-1|t-1} = \begin{bmatrix} \boldsymbol{\ell}_{t-1|t-1}^\intercal & \boldsymbol{h}_{t-1|t-1}^\intercal \end{bmatrix}^\intercal$ is given, and $\tilde{\mathbf{J}}_t = \begin{bmatrix} \tilde{\mathbf{L}}_t & \tilde{\mathbf{H}}_t \end{bmatrix}$ with $\tilde{\mathbf{L}}_t = \nabla_{\boldsymbol{\ell}} f(\boldsymbol{\ell}_{t-1|t-1}, \boldsymbol{h}_{t-1|t-1}, \boldsymbol{x}_t)$ and $\tilde{\mathbf{H}}_t = \nabla_{\boldsymbol{h}} f(\boldsymbol{\ell}_{t-1|t-1}, \boldsymbol{h}_{t-1|t-1}, \boldsymbol{x}_t)$.

Following a frequentist approach to the Kalman filter [27] and given the starting latent random vector $\boldsymbol{\theta}_{0|0} := \boldsymbol{\theta}_0$, we seek to obtain sequential updates for the *best* ($L^2$) linear estimate of the expected value of $\boldsymbol{\theta}_t$ given data $\boldsymbol{y}_{1:t}$. We formalize this in the following proposition.

**Proposition B.1.** *Let $k, t \in \mathbb{N}$ and let $\boldsymbol{y}_t$, $\boldsymbol{\theta}_t$ follow the SSM* (12). *The solution to the optimization problem*

$$
\underset{\mathbf{A} \in \mathcal{M}_{(D_{\boldsymbol{\ell}} + D_{\boldsymbol{h}}) \times (k\, D_{\boldsymbol{y}})}(\mathbb{R})}{\arg\max} \mathbb{E}\left[ ||\boldsymbol{\theta}_t - \mathbf{A}\, \boldsymbol{y}_{1:k}||_2^2 \right],
\tag{13}
$$

*is the matrix $\mathbf{A}_{t|k}^\star$ given by*

$$
\mathbf{A}_{t|k}^\star = \mathrm{Cov}(\boldsymbol{\theta}_t, \boldsymbol{y}_{1:k})\, \mathrm{Var}(\boldsymbol{y}_{1:k})^{-1}.
\tag{14}
$$

*The best linear unbiased predictor (BLUP) for model parameters $\boldsymbol{\theta}_t$, given observations $\boldsymbol{y}_{1:k}$ is*

$$
\boldsymbol{\theta}_{t|k} = \mathbf{A}_{t|k}^\star\, \boldsymbol{y}_{1:k},
\tag{15}
$$

*and the error variance covariance (EVC) matrix is defined as*

$$
\boldsymbol{\Sigma}_{t|k} := \mathrm{Var}(\boldsymbol{\theta}_t) - \mathbf{A}_{t|k}^\star\, \mathrm{Var}(\boldsymbol{y}_{1:k})\, \mathbf{A}_{t|k}^{\star\intercal}.
\tag{16}
$$

**Proposition B.2.** *Under the SSM* (12), *the BLUP and the EVC can be written in the form of Kalman filtering predict and update equations. Here, the predict equations are*

$$
\begin{aligned}
\boldsymbol{\theta}_{t|t-1} &= \boldsymbol{\theta}_{t-1|t-1}, \\
\boldsymbol{\Sigma}_{t|t-1} &= \boldsymbol{\Sigma}_{t-1|t-1} + \mathbf{Q}_t,
\end{aligned}
\tag{17}
$$

*and the update equations are*

$$
\begin{aligned}
\boldsymbol{\theta}_{t|t} &= \boldsymbol{\theta}_{t|t-1} + \mathbf{K}_t\, \boldsymbol{\epsilon}_t, \\
\boldsymbol{\Sigma}_{t|t} &= (\mathbf{I} - \mathbf{K}_t\, \tilde{\mathbf{J}}_t)\, \boldsymbol{\Sigma}_{t|t-1}(\mathbf{I} - \mathbf{K}_t\, \tilde{\mathbf{J}}_t)^\intercal + \mathbf{K}_t\, \mathbf{R}_t \mathbf{K}_t^\intercal,
\end{aligned}
\tag{18}
$$

*with*

$$
\begin{aligned}
\boldsymbol{\epsilon}_t &= \boldsymbol{y}_t - f(\boldsymbol{\theta}_{t-1|t-1}, \boldsymbol{x}_t), \\
\mathbf{K}_t &= \boldsymbol{\Sigma}_{t|t-1} \tilde{\mathbf{J}}_t^\intercal\, \mathbf{S}_t^{-1}, \\
\mathbf{S}_t &= \tilde{\mathbf{J}}_t \boldsymbol{\Sigma}_{t|t-1} \tilde{\mathbf{J}}_t^\intercal + \mathbf{R}_t.
\end{aligned}
\tag{19}
$$

*Proof.* The proof follows from the linear form of (12), Lemmas 2.1 to 2.3, and Theorem 4.2 in [19]. $\square$

*Remark* B.3. The update equations in (18) are recursive and characterize estimate of the unknown latent state and the estimated error estimation through the first two moments. Furthermore, the update for the covariance in (18) is in Joseph form [63] and is known to be numerically stable. In our method, we leverage these two facts to target a low-rank EVC through a second-step optimization that has numerically-stable updates.

**Corollary B.4** (Kalman filter as a Bayesian posterior). *Assume a Gaussian density for the initial parameters $p(\boldsymbol{\theta}_0) = \mathcal{N}(\boldsymbol{\theta}_0 \,|\, \boldsymbol{\theta}_{0|0}, \boldsymbol{\Sigma}_{0|0})$ and for the noise term $p(\boldsymbol{e}_t) = \mathcal{N}(\boldsymbol{e}_t \,|\, \boldsymbol{0}, \, \mathbf{R}_t)$. The BLUP $\boldsymbol{\theta}_{t|t}$ and the EVC $\boldsymbol{\Sigma}_{t|t}$ are the parameters of the Gaussian that characterize the posterior of model parameters $\boldsymbol{\theta}_t$ given $\boldsymbol{y}_{1:t}$, more precisely,*

$$p(\boldsymbol{\theta}_t \,|\, \boldsymbol{y}_{1:t}) = \mathcal{N}(\boldsymbol{\theta}_t \,|\, \boldsymbol{\theta}_{t|t}, \boldsymbol{\Sigma}_{t|t}). \tag{20}$$

*As a consequence, the posterior predictive for the next observation in the sequence is*

$$\begin{aligned} p(\boldsymbol{y}_t \,|\, \boldsymbol{y}_{1:t-1}) &= \int \mathcal{N}\left(\boldsymbol{y}_t \,|\, f(\boldsymbol{\theta}_{t-1|t-1}, \boldsymbol{x}_t) + \tilde{\mathbf{J}}_t(\boldsymbol{\theta}_t - \boldsymbol{\theta}_{t|t-1}), \mathbf{R}_t\right) p(\boldsymbol{\theta}_t \,|\, \mathcal{D}_{1:t-1}) \mathrm{d}\boldsymbol{\theta}_t \\ &= \mathcal{N}(\boldsymbol{y}_t \,|\, f(\boldsymbol{\theta}_{t-1|t-1}, \boldsymbol{x}_t), \ \tilde{\mathbf{J}}_t \boldsymbol{\Sigma}_{t|t-1} \tilde{\mathbf{J}}_t^{\mathsf{T}} + \mathbf{R}_t). \end{aligned} \tag{21}$$

# C   Linear algebra results

## C.1   Sum of Cholesky matrices

Here, we generalize the result found in [57] to any set of $K > 2$ matrices, which we use to derive the method `LRKF`, `HiLoFi`, and `LoLoFi`.

**Proposition C.1** (QR of sum of Cholesky matrices). *Let $\mathbf{A}_i$ for $i \in \{1, 2, \ldots, I\}$ be a collection of $(D \times D)$ positive definite matrices and let $\mathbf{A}_i^{1/2}$ be the upper-triangular Cholesky decomposition of $\mathbf{A}_i$. Let $\mathbf{M}$ be the stacked $(D\,I \times D)$ matrix given by*

$$\mathbf{M} = \begin{bmatrix} \mathbf{A}_1^{1/2} \\ \mathbf{A}_2^{1/2} \\ \vdots \\ \mathbf{A}_I^{1/2} \end{bmatrix}. \tag{22}$$

*It follows that*

$$\mathbf{R}^{\mathsf{T}} \mathbf{R} = \sum_{i=1}^{I} \mathbf{A}_i. \tag{23}$$

*where $\mathbf{R}$ is R component in the QR decomposition of $\mathbf{M}$.*

*Proof.* We see that

$$\mathbf{M}^{\mathsf{T}} \mathbf{M} = \sum_{i=1}^{I} \mathbf{A}_i. \tag{24}$$

Consider the QR decomposition of $\mathbf{M}$

$$\mathbf{M} = \mathbf{Q}\mathbf{R}, \tag{25}$$

with $\mathbf{Q}$ an orthogonal matrix and $\mathbf{R}$ and upper-triangular matrix.

Then,

$$\sum_{i=1}^{I} \mathbf{A}_i = \mathbf{M}^{\mathsf{T}} \mathbf{M} = \mathbf{R}^{\mathsf{T}} \mathbf{Q}^{\mathsf{T}} \mathbf{Q} \mathbf{R} = \mathbf{R}^{\mathsf{T}} \mathbf{R}. \tag{26}$$

As a consequence, we note that the *square root* of the matrix $\mathbf{A}_1 + \cdots + \mathbf{A}_I$ is the upper-triangular matrix in the QR decomposition of $\mathbf{M}$. $\qquad\square$

**Remark**   In what follows, we denote the result of obtaining the $\mathbf{R}$ matrix of the QR decomposition of the row-stacked matrices $\mathbf{A}_i^{1/2}$ for $i \in \{1, \ldots, I\}$ by

$$\mathcal{Q}_R(\mathbf{A}_1^{1/2}, \ldots, \mathbf{A}_I^{1/2}). \tag{27}$$

## C.2 Singular value decomposition given sum of low-rank-matrices

**Proposition C.2** (SVD of sum of low-rank matrices)*. Let $\mathbf{A}_i$ for $i \in \{1, 2, \dots, N\}$ be $\mathcal{M}_D^{++}(\mathbb{R})$ positive semi-definite matrices such that $\mathbf{A}_i = \mathbf{W}_i^\intercal \mathbf{W}_i$, where $\mathbf{W}_i \in \mathbb{R}^{d \times D}$ and $d \ll D$. The best rank-$d$ approximation of the sum is*

$$\sum_{i=N}^{I} \mathbf{A}_i \approx \mathbf{J}^\intercal \mathbf{J}, \tag{28}$$

*where $\mathbf{J} = \mathbf{S}_{:d} \mathbf{V}^\intercal \in \mathbb{R}^{d \times D}$, $\mathbf{S}_{:d}$ are the top $d$ singular values and $\mathbf{V}$ contains the right singular vectors of the stacked $(d\,N \times D)$ matrix given by*

$$\mathbf{N} = \begin{bmatrix} \mathbf{W}_1 \\ \mathbf{W}_2 \\ \vdots \\ \mathbf{W}_N \end{bmatrix}. \tag{29}$$

*Proof.* Let $\mathbf{A}_i$ for $i \in \{1, 2, \dots, N\}$ be $\mathcal{M}_D^{++}(\mathbb{R})$ matrices with $\mathbf{A}_i = \mathbf{W}_i^\intercal \mathbf{W}_i$, $\mathbf{W}_i \in \mathbb{R}^{d \times D}$, and $d \ll D$. Form the matrix $\mathbf{N}$ given by

$$\mathbf{N} = \begin{bmatrix} \mathbf{W}_1 \\ \mathbf{W}_2 \\ \vdots \\ \mathbf{W}_N \end{bmatrix} = \mathbf{U}\,\mathbf{S}\,\mathbf{V}^\intercal, \tag{30}$$

where $\mathbf{U}\,\mathbf{S}\,\mathbf{V}^\intercal$ is its full singular value decomposition, with $\mathbf{U} \in \mathbb{R}^{dN \times dN}$, $\mathbf{S} \in \mathbb{R}^{dN \times D}$, and $\mathbf{V} \in \mathbb{R}^{D \times D}$. Then

$$\begin{aligned} \sum_{i=1}^{N} \mathbf{A}_i = \sum_{i=1}^{N} \mathbf{W}_i^\intercal \mathbf{W}_i &= \mathbf{N}^\intercal \mathbf{N} \\ &= (\mathbf{U}\,\mathbf{S}\,\mathbf{V}^\intercal)^\intercal (\mathbf{U}\,\mathbf{S}\,\mathbf{V}^\intercal) \\ &= \mathbf{V}\,\mathbf{S}^\intercal\,\mathbf{U}^\intercal\,\mathbf{U}\,\mathbf{S}\,\mathbf{V}^\intercal \\ &= \mathbf{V}\,\mathbf{S}^2\,\mathbf{V}^\intercal. \end{aligned} \tag{31}$$

Hence $\sum_i \mathbf{A}_i$ has eigenvectors $\mathbf{V}$ and eigenvalues given by the diagonal entries of $\mathbf{S}^2$. By the symmetric-matrix form of the Eckart–Young theorem [17, 10], its best rank-$d$ approximation in Frobenius norm is

$$\mathbf{V} \operatorname{diag}(\sigma_1^2, \dots, \sigma_d^2, 0, \dots) \mathbf{V}^\intercal. \tag{32}$$

Writing $\mathbf{S}_{:d} = \operatorname{diag}(\sigma_1, \dots, \sigma_d) \in \mathbb{R}^{d \times D}$, one checks

$$\mathbf{V} \operatorname{diag}(\sigma_1^2, \dots, \sigma_d^2, 0, \dots) \mathbf{V}^\intercal = \left(\mathbf{S}_{:d}\,\mathbf{V}^\intercal\right)^T \left(\mathbf{S}_{:d}\,\mathbf{V}^\intercal\right) = \mathbf{J}^\intercal \mathbf{J}, \tag{33}$$

where

$$\begin{aligned} \mathbf{J} &= \mathbf{S}_{:d}\,\mathbf{V}^\intercal \in \mathbb{R}^{d \times D}, \\ \mathbf{S}_{:d} &:= \operatorname{diag}\left(\sigma_1, \dots, \sigma_d\right), \end{aligned} \tag{34}$$

and $\{\sigma_k\}$ are the singular vectors of $\sum_{n=1}^{N} \mathbf{A}_n$. $\qquad\square$

**Takeaway.** Proposition C.2 shows that to form the best rank-$d$ approximation of $\sum_{i=1}^{N} \mathbf{A}_i$ one does not need to assemble or carry out the SVD of the full $D \times D$ sum. Instead, one stacks the low-rank factors into

$$\mathbf{N} = \begin{bmatrix} \mathbf{W}_1 \\ \vdots \\ \mathbf{W}_N \end{bmatrix} \in \mathbb{R}^{(dN) \times D}, \tag{35}$$

and compute a reduced SVD (or symmetric eigendecomposition) of the small $(dN) \times (dN)$ Gram matrix $\mathbf{N}\mathbf{N}^\top$. This costs

$$\mathcal{O}((dN)^2 D + (dN)^3) \approx \mathcal{O}(d^2 N^2 D) \text{ when } dN \ll D, \tag{36}$$

versus the $\mathcal{O}(D^3)$ required for a full SVD on the $D \times D$ sum. Moreover, memory and computation scale with $d$ and $dN$ rather than with the large dimension $D$ because all operations involve only the $(dN) \times D$ matrix $\mathbf{N}$ and the $d \times D$ factor $\mathbf{J}$.

In what follows, we denote the result of obtaining the best rank-$d$ approximation matrix of the SVD decomposition of the row-stacked matrices $\mathbf{W}_i^{1/2}$ for $i \in \{1, \dots, N\}$ by

$$\mathcal{P}_d(\mathbf{W}_1, \dots, \mathbf{W}_N). \tag{37}$$

**Corollary C.3.** *If the sum of matrices is of the form*

$$\sum_{n=1}^{N} \mathbf{A}_n + a \, \mathbf{I}_D \tag{38}$$

*with $a > 0$, then the best rank-d approximation is of the form*

$$\sum_{n=1}^{N} \mathbf{A}_n + a \, \mathbf{I}_D \approx \mathbf{J}_{+q}^{\mathsf{T}} \mathbf{J}_{+q} \tag{39}$$

*where*

$$\mathbf{J}_{+q} = \mathbf{S}_{:d,+q} \, \mathbf{V}^{\mathsf{T}} \in \mathbb{R}^{d \times D},$$
$$\mathbf{S}_{:d,+q} := \mathrm{diag} \left( \sqrt{\sigma_1^2 + q}, \dots, \sqrt{\sigma_d^2 + q} \right), \tag{40}$$

*and $\{\sigma_k\}_{k=1}^{d}$ are the top-d singular values of $\sum_{n=1}^{N} \mathbf{A}_n$.*

*Proof.* From (31) in Proposition C.2 we obtain

$$\sum_{i=1}^{N} \mathbf{A}_i + a \, \mathbf{I} = \mathbf{V} \, \mathbf{S}^2 \, \mathbf{V}^{\mathsf{T}} + a\mathbf{I} = \mathbf{V} \, (\mathbf{S}^2 + a\mathbf{I}_D) \, \mathbf{V}^{\mathsf{T}} = \mathbf{V} \, \mathbf{S}_{+a}^2 \, \mathbf{V}^{\mathsf{T}}, \tag{41}$$

where

$$\mathbf{S}_{+a}^2 = \mathrm{diag}(\sigma_1^2 + a, \dots, \sigma_D^2 + a). \tag{42}$$

Let

$$\mathbf{S}_{+a} = \mathrm{diag} \left( \sqrt{\sigma_1^2 + a}, \dots, \sqrt{\sigma_D^2 + a} \right), \tag{43}$$

then

$$\sum_{n=1}^{N} \mathbf{A}_n = \mathbf{V} \, \mathbf{S}_{+a}^2 \, \mathbf{V}^{\mathsf{T}}$$
$$= (\mathbf{V} \, \mathbf{S}_{+a} \mathbf{V}^{\mathsf{T}})^{\mathsf{T}} \, (\mathbf{V} \, \mathbf{S}_{+a} \mathbf{V}^{\mathsf{T}})$$
$$= \mathbf{J}_{+a}^{\mathsf{T}} \, \mathbf{J}_{+a}. \tag{44}$$

$\square$

**Takeaway.** If the sum of matrices can be written as the sum of multiplied low-rank factors plus an additional identity matrix times a constant, then, computation of the best rank $d$-matrix in low-rank factor can be obtained by performing SVD over the low-rank factors and modifying the singular values to include the term $a$.

In what follows, we denote the result of obtaining the best rank-$d$ approximation matrix of the SVD decomposition of the row-stacked matrices $\mathbf{W}_i^{1/2}$ for $i \in \{1, \dots, N\}$ plus an identity times a real-valued number $a > 0$ as

$$\mathcal{P}_{d,+a}(\mathbf{W}_1, \dots, \mathbf{W}_N). \tag{45}$$

# D HiLoFi— further details

## D.1 Derivation of HiLoFi

Consider the linearized model

$$\boldsymbol{y}_t = f(\boldsymbol{\ell}_{t-1|t-1}, \boldsymbol{h}_{t-1|t-1}, \boldsymbol{x}_t) + \tilde{\mathbf{L}}_t(\boldsymbol{\ell}_t - \boldsymbol{\ell}_{t-1|t-1}) + \tilde{\mathbf{H}}_t(\boldsymbol{h}_t - \boldsymbol{h}_{t-1|t-1}) + \boldsymbol{e}_t, \qquad (46)$$

where $\boldsymbol{\ell}_{t-1|t-1}$ and $\boldsymbol{h}_{t-1|t-1}$ are given.

We start the algorithm by initialising the beliefs about the last layer parameters $\boldsymbol{\ell}$ and the hidden layer parameters $\boldsymbol{h}$. We set

$$\mathbb{E}[\boldsymbol{\ell}_0] = \boldsymbol{\ell}_{0|0}, \quad \mathbb{E}[\boldsymbol{h}_0] = \boldsymbol{h}_{0|0}, \quad \mathrm{Var}(\boldsymbol{\ell}_0) = \hat{\boldsymbol{\Sigma}}_{\boldsymbol{\ell},0}, \quad \mathrm{Var}(\boldsymbol{h}_0) = \mathbf{C}_0^\intercal \, \mathbf{C}_0, \qquad (47)$$

for known $\mathbf{C}_0 \in \mathcal{M}_{d_{\boldsymbol{h}} \times D_{\boldsymbol{h}}}(\mathbb{R})$, $\boldsymbol{h}_{0|0} \in \mathbb{R}^{D_{\boldsymbol{h}}}$, $\hat{\boldsymbol{\Sigma}}_{\boldsymbol{\ell},0} \in \mathcal{M}_{D_{\boldsymbol{\ell}}}^{++}(\mathbb{R})$, and $\boldsymbol{\ell}_{0|0} \in \mathbb{R}^{D_{\boldsymbol{\ell}}}$. Next, we assume that for $t \geq 1$, the latent last layer parameters $\boldsymbol{\ell}_{1:t}$ and the hidden layer parameters $\boldsymbol{h}_{1:t}$ follow the dynamics

$$\boldsymbol{\ell}_t = \boldsymbol{\ell}_{t-1} + \boldsymbol{u}_{\boldsymbol{\ell},t}, \qquad \boldsymbol{h}_t = \boldsymbol{h}_{t-1} + \boldsymbol{u}_{\boldsymbol{h},t}, \qquad (48)$$

where $\boldsymbol{u}_{\boldsymbol{\ell},1:t}$ and $\boldsymbol{u}_{\boldsymbol{h},1:t}$ are zero-mean independent noise variables[1] with dynamics covariance matrices $\mathrm{Var}(\boldsymbol{u}_{\boldsymbol{\ell},t}) = \mathbf{Q}_{\boldsymbol{\ell},t} \in \mathcal{M}_{D_{\boldsymbol{\ell}}}^+(\mathbb{R})$ and $\mathrm{Var}(\boldsymbol{u}_{\boldsymbol{h},t}) = \mathbf{Q}_{\boldsymbol{h},t} \in \mathcal{M}_{D_{\boldsymbol{h}}}^+(\mathbb{R})$. For simplicity and to exploit efficient linear algebra techniques for belief updates, we assume $\mathbf{Q}_{\boldsymbol{\ell},t} = q_{\boldsymbol{\ell},t}\mathbf{I}_{D_{\boldsymbol{\ell}}}$ and $\mathbf{Q}_{\boldsymbol{h},t} = q_{\boldsymbol{h},t}\mathbf{I}_{D_{\boldsymbol{h}}}$. A simple choice is to set $\mathbf{Q}_{\boldsymbol{h},t} = 0\mathbf{I}$ to avoid forgetting past data; see e.g., [16].

**Proposition D.1** (Predict step). *With the SSM assumption* (48)*, the approximate covariance* $\mathrm{Var}(\boldsymbol{\ell}_{t-1} - \boldsymbol{\ell}_{t-1|t-1}) = \hat{\boldsymbol{\Sigma}}_{\boldsymbol{\ell},t-1}$ *and the approximate covariance* $\mathrm{Var}(\boldsymbol{h}_{t-1} - \boldsymbol{h}_{t-1|t-1}) = \mathbf{C}_{t-1}^\intercal \, \mathbf{C}_{t-1}$*, the predict step becomes*

$$\begin{aligned} \boldsymbol{h}_{t|t-1} &= \boldsymbol{h}_{t-1|t-1}, & \boldsymbol{\ell}_{t|t-1} &= \boldsymbol{\ell}_{t-1|t-1}, \\ \boldsymbol{\Sigma}_{\boldsymbol{\ell},t|t-1} &= \hat{\boldsymbol{\Sigma}}_{\boldsymbol{\ell},t-1} + q_{\boldsymbol{\ell},t} \, \mathbf{I}_{D_{\boldsymbol{\ell}}}, & \boldsymbol{\Sigma}_{\boldsymbol{h},t|t-1} &= \mathbf{C}_{t-1}^\intercal \, \mathbf{C}_{t-1} + q_{\boldsymbol{h},t} \, \mathbf{I}_{D_{\boldsymbol{h}}}. \end{aligned} \qquad (49)$$

The proof of Proposition D.1 is in Appendix E.1.

**Proposition D.2** (Variance of the innovation). *The upper Cholesky decomposition of innovation variance takes the form*

$$\mathbf{S}_t^{1/2} = \mathcal{Q}_R\left(\hat{\boldsymbol{\Sigma}}_{\boldsymbol{\ell},t-1}^{1/2} \tilde{\mathbf{L}}_t^\intercal, \sqrt{q_{\boldsymbol{\ell},t}} \tilde{\mathbf{L}}_t^\intercal, \mathbf{C}_{t-1} \tilde{\mathbf{H}}_t^\intercal, \sqrt{q_{\boldsymbol{h},t}} \tilde{\mathbf{H}}_t^\intercal, \mathbf{R}_t^{1/2}\right), \qquad (50)$$

*where $\tilde{\mathbf{L}}_t$ and $\tilde{\mathbf{H}}_t$ are defined in Section 3.2.*

The proof of Proposition D.2 is in Appendix E.2.

Next, our update step for the BLUP resembles an EKF update, while the EVC update approximates a block-diagonal covariance: the last layer block is full-rank, and the hidden layer blocks are low-rank. To maintain numerical stability and low-memory updates, we track the Cholesky factor for the last layer and a low-rank factor for the hidden layers. The Cholesky factor is taken from a surrogate matrix that avoids computing a full $\mathcal{M}_{D_{\boldsymbol{\ell}} \times D_{\boldsymbol{\ell}}}(\mathbb{R})$ matrix by neglecting the artificial dynamics covariance $q_{\boldsymbol{\ell},t}$. The low-rank component for hidden layers is approximated via a two-step process: (i) a fast *surrogate* predicted covariance, and (ii) a rank $D_{\boldsymbol{h}}$ projection error. Like the Cholesky factor, the surrogate covariance in (i) reduces the computational cost of dynamics noise, which does not reflect the system's true dynamics. However, we retain some information from the dynamics noise to update all hidden layer parameters. Step (ii) enables a fast update rule, crucial for overparameterized neural networks [38]. We detail this step in the following proposition.

**Proposition D.3** (Kalman gain, update BLUP, and update EVC). *Define the gain matrices*

$$\mathbf{K}_{\boldsymbol{h},t}^\intercal = \mathbf{V}_{\boldsymbol{h},t}\mathbf{C}_{t-1}\mathbf{C}_{t-1}^\intercal + q_{\boldsymbol{h},t}\mathbf{V}_{\boldsymbol{h},t}, \quad \mathbf{K}_{\boldsymbol{\ell},t}^\intercal = \mathbf{V}_{\boldsymbol{\ell},t}\boldsymbol{\Sigma}_{\boldsymbol{\ell},t|t-1} + q_{\boldsymbol{\ell},t}\mathbf{V}_{\boldsymbol{\ell},t}, \qquad (51)$$

*where $\mathbf{V}_{\boldsymbol{h},t} = \mathbf{S}_t^{-1/2} \mathbf{S}_t^{-\intercal/2} \tilde{\mathbf{H}}_t$, and $\mathbf{V}_{\boldsymbol{\ell},t} = \mathbf{S}_t^{-1/2} \mathbf{S}_t^{-\intercal/2} \tilde{\mathbf{L}}_t$. The updated BLUP for the last layer parameters and hidden layer parameters are*

$$\boldsymbol{h}_{t|t} = \boldsymbol{h}_{t-1|t-1} + \mathbf{K}_{\boldsymbol{h},t} \, \boldsymbol{\epsilon}_t, \quad \boldsymbol{\ell}_{t|t} = \boldsymbol{\ell}_{t-1|t-1} + \mathbf{K}_{\boldsymbol{\ell},t} \, \boldsymbol{\epsilon}_t. \qquad (52)$$

---

[1]i.e., $\mathrm{Cov}(\boldsymbol{u}_{\boldsymbol{\ell},i}, \boldsymbol{u}_{\boldsymbol{h},j}) = \mathbf{0}$ for all $i, j \in \{1, \ldots, T\}$.

The approximate posterior covariance $\hat{\mathbf{\Sigma}}_{t|t}$ is the best block-diagonal approximation (in Frobenius norm) of the surrogate matrix $\tilde{\mathbf{\Sigma}}_{t|t}$ given by

$$\begin{bmatrix} (\mathbf{I}_{D_{\boldsymbol{\ell}}} - \mathbf{K}_{\boldsymbol{\ell},t}\tilde{\mathbf{L}}_t)\hat{\mathbf{\Sigma}}_{t-1}(\mathbf{I}_{D_{\boldsymbol{\ell}}} - \mathbf{K}_{\boldsymbol{\ell},t}\tilde{\mathbf{L}}_t)^{\mathsf{T}} - \mathbf{K}_{\boldsymbol{\ell},t}\mathbf{R}_t\mathbf{K}_{\boldsymbol{\ell},t}^{\mathsf{T}} & \mathbf{K}_{\boldsymbol{\ell},t}\mathbf{R}_t\mathbf{K}_{\boldsymbol{h},t}^{\mathsf{T}} \\ \mathbf{K}_{\boldsymbol{h},t}\mathbf{R}_t\mathbf{K}_{\boldsymbol{\ell},t}^{\mathsf{T}} & (\mathbf{I}_{D_{\boldsymbol{h}}} - \mathbf{K}_{\boldsymbol{h},t}\tilde{\mathbf{H}}_t)\mathbf{C}_{t-1}^{\mathsf{T}}\mathbf{C}_{t-1}(\mathbf{I}_{D_{\boldsymbol{h}}} - \mathbf{K}_{\boldsymbol{h},t}\tilde{\mathbf{H}}_t)^{\mathsf{T}} - \mathbf{K}_{\boldsymbol{h},t}\mathbf{R}_t\mathbf{K}_{\boldsymbol{h},t}^{\mathsf{T}} + q_{\boldsymbol{h},t}\mathbf{I}_{D_{\boldsymbol{h}}} \end{bmatrix},$$

that has full rank for the last layer and rank $d_{\boldsymbol{h}}$ for the hidden layer. This results in a Cholesky factor for the last-layer covariance given by

$$\hat{\mathbf{\Sigma}}_{\boldsymbol{\ell},t}^{1/2} = \mathcal{Q}_R\left(\hat{\mathbf{\Sigma}}_{\boldsymbol{\ell},t-1}^{1/2}(\mathbf{I} - \mathbf{K}_{\boldsymbol{\ell},t}\tilde{\mathbf{L}}_t)^{\mathsf{T}}, \ \mathbf{R}_t^{1/2}\mathbf{K}_{\boldsymbol{\ell},t}^{\mathsf{T}}\right), \tag{53}$$

and a low-rank factor for the hidden layers given by

$$\mathbf{C}_t = \mathcal{P}_{d_{\boldsymbol{h}},+q_{\boldsymbol{h},t}}\left(\mathbf{C}_{t-1}(\mathbf{I} - \mathbf{K}_{\boldsymbol{h},t}\tilde{\mathbf{H}}_t)^{\mathsf{T}}, \ \mathbf{R}_t^{1/2}\mathbf{K}_{\boldsymbol{h},t}^{\mathsf{T}}\right), \tag{54}$$

The proof is in Appendix E.3.

### D.2 Computational complexity

We now compare the computational complexity of the algorithms we employ. Recall that $D_{\boldsymbol{\ell}}, D_{\boldsymbol{h}}$, and $D_{\boldsymbol{y}}$ are the number of parameters in the last layer, hidden layers, and the size of the output layer, respectively. As before, $D_{\boldsymbol{\theta}} = D_{\boldsymbol{\ell}} + D_{\boldsymbol{h}}$. The dimensions $d_{\boldsymbol{\ell}}, d_{\boldsymbol{h}}$, with $d_{\boldsymbol{\ell}} \ll D_{\boldsymbol{\ell}}$ and $d_{\boldsymbol{h}} \ll D_{\boldsymbol{h}}$, are the low-dimensional subspaces for the last and hidden layers, respectively, and $d = d_{\boldsymbol{\ell}} + d_{\boldsymbol{h}}$.

The computational complexity of our algorithm when using full-rank in the last layer is $O(D_{\boldsymbol{\ell}}(D_{\boldsymbol{\ell}} + D_{\boldsymbol{y}})^2 + D_{\boldsymbol{h}}(d_{\boldsymbol{h}} + D_{\boldsymbol{y}})^2)$. For low-dimensional outputs, this is $O(D_{\boldsymbol{\ell}}^3 + D_{\boldsymbol{h}}d_{\boldsymbol{h}}^2)$. For high-dimensional outputs, we may choose to use a low-rank approximation for the last layer (which we call LoLoFi) this reduces the cost to $O(D_{\boldsymbol{\ell}}(d_{\boldsymbol{\ell}} + D_{\boldsymbol{y}})^2 + D_{\boldsymbol{h}}(d_{\boldsymbol{h}} + D_{\boldsymbol{y}})^2)$, which is linear in $D_{\boldsymbol{\theta}}$ The primary computational bottlenecks in our method are the calculations for the Kalman gain of the hidden and last layers, with costs $O(2\,D_{\boldsymbol{h}}\,D_{\boldsymbol{y}}^2)$ and $O(2\,D_{\boldsymbol{\ell}}\,D_{\boldsymbol{y}}^2)$ respectively. This efficiency arises because $\mathbf{S}_t^{1/2}$ is upper triangular, allowing the system $\mathbf{S}_t^{\mathsf{T}/2}\mathbf{S}_t^{1/2}\mathbf{V} = \tilde{\mathbf{J}}$ for $\mathbf{V}$ to be solved in $O(2\,D_{\boldsymbol{h}}\,D_{\boldsymbol{y}}^2)$ as opposed to $O(D_{\boldsymbol{h}}\,D_{\boldsymbol{y}}^3)$. Additionally, approximating the covariance matrix via truncated SVD incurs a cost of $O(D_{\boldsymbol{h}}(d_{\boldsymbol{h}} + D_{\boldsymbol{y}})^2 + (d_{\boldsymbol{h}} + D_{\boldsymbol{y}})^3 + d_{\boldsymbol{h}}(d_{\boldsymbol{h}} + D_{\boldsymbol{y}})D_{\boldsymbol{h}})$, as detailed in Figure 8.6.1 in [23]. Finally, the corresponding cost for the last layer is $O(D_{\boldsymbol{\ell}}(D_{\boldsymbol{\ell}} + D_{\boldsymbol{y}})^2)$.

Among related methods, the closest to ours in terms of computational costs are variational Bayes approaches such as the Slang method [47], the L-RVGA method [36], and the LoFi method [8]. In particular, LoFi uses the linearized Gaussian updates (similar to the ones in this paper), but approximates the precision matrix using a diagonal-plus-low rank (DLR) form. The appeal for DLR precision matrices is twofold. First, they enable the application of the Woodbury identity, leading to a predict step of cost $O(D_{\boldsymbol{\theta}}\,d + d^3)$ and an update step of $O(D_{\boldsymbol{\theta}}(d + D_{\boldsymbol{y}})^2) = O(D_{\boldsymbol{\ell}}(d + D_{\boldsymbol{y}})^2 + D_{\boldsymbol{h}}(d + D_{\boldsymbol{y}})^2)$. Second, incorporating positive diagonal terms ensures the matrix is positive definite, so that a valid posterior Gaussian density is defined.

Although HiLoFi and LoFi have the same asymptotic complexity, LoFi incurs additional practical overhead due to three key operations absent in HiLoFi: (1) the inversion of a $(d + D_{\boldsymbol{y}})$ rectangular matrix, (2) the inversion of a $d$ rectangular matrix, and (3) the Cholesky decomposition of a $d$ rectangular matrix. These operations increase the actual per-step computational time, which in our example, scales at about one second per additional rank. This behaviour is shown in the empirical comparison across HiLoFi, LRKF, and LoFi in the online classification setting (Figure 6, Appendix G.1).

## E  Proofs

### E.1  Proof of Proposition D.1

*Proof.* Following Proposition B.2, the predicted mean takes the form

$$\boldsymbol{\theta}_{t|t-1} = \boldsymbol{\theta}_{t-1|t-1} = \begin{bmatrix} \boldsymbol{\ell}_{t-1|t-1} \\ \boldsymbol{h}_{t-1|t-1} \end{bmatrix}. \tag{55}$$

Next, the predicted posterior covariance takes the form

$$\mathbf{\Sigma}_{t|t-1} = \mathbf{\Sigma}_{t-1|t-1} + \mathbf{Q}_t, \tag{56}$$

where

$$
\begin{aligned}
&\mathbf{\Sigma}_{t-1|t-1} \\
&= \operatorname{Var}(\boldsymbol{\theta}_{t-1} - \boldsymbol{\theta}_{t-1|t-1}) \\
&= \begin{bmatrix} \operatorname{Var}(\boldsymbol{\ell}_{t-1} - \boldsymbol{\ell}_{t-1|t-1}) & \operatorname{Cov}(\boldsymbol{\ell}_{t-1} - \boldsymbol{\ell}_{t-1|t-1}, \boldsymbol{h}_{t-1} - \boldsymbol{h}_{t-1|t-1}) \\ \operatorname{Cov}(\boldsymbol{h}_{t-1} - \boldsymbol{h}_{t-1|t-1}, \boldsymbol{\ell}_{t-1} - \boldsymbol{\ell}_{t-1|t-1}) & \operatorname{Var}(\boldsymbol{h}_{t-1} - \boldsymbol{h}_{t-1|t-1}). \end{bmatrix} \\
&= \begin{bmatrix} \mathbf{\Sigma}_{\boldsymbol{\ell},t-1} & \operatorname{Cov}(\boldsymbol{\ell}_{t-1} - \boldsymbol{\ell}_{t-1|t-1}, \boldsymbol{h}_{t-1} - \boldsymbol{h}_{t-1|t-1}) \\ \operatorname{Cov}(\boldsymbol{h}_{t-1} - \boldsymbol{h}_{t-1|t-1}, \boldsymbol{\ell}_{t-1} - \boldsymbol{\ell}_{t-1|t-1}) & \mathbf{W}_{t-1}^{\mathsf{T}} \mathbf{W}_{t-1} \end{bmatrix}.
\end{aligned}
\tag{57}
$$

Next,

$$
\begin{aligned}
&\operatorname{Cov}(\boldsymbol{\ell}_{t-1} - \boldsymbol{\ell}_{t-1|t-1}, \boldsymbol{h}_{t-1} - \boldsymbol{h}_{t-1|t-1}) \\
&= \operatorname{Cov}(\boldsymbol{\ell}_{t-1}, \boldsymbol{h}_{t-1}) \\
&= \operatorname{Cov}\left(\boldsymbol{\ell}_0 + \sum_{\tau=1}^{t-1} \boldsymbol{u}_{\boldsymbol{\ell},\tau}, \; \boldsymbol{h}_0 + \sum_{\tau=1}^{t-1} \boldsymbol{u}_{\boldsymbol{h},\tau}\right) \\
&= \operatorname{Cov}(\boldsymbol{\ell}_0, \boldsymbol{h}_0) + \sum_{\tau=1}^{t-1} \operatorname{Cov}(\boldsymbol{\ell}_0, \boldsymbol{u}_{\boldsymbol{h},\tau}) + \sum_{\tau=1}^{t-1} \operatorname{Cov}(\boldsymbol{u}_{\boldsymbol{\ell},\tau}, \boldsymbol{h}_0) + \sum_{\tau=1}^{t-1} \sum_{\tau'=1}^{-1} \operatorname{Cov}(\boldsymbol{u}_{\boldsymbol{\ell},\tau}, \boldsymbol{u}_{\boldsymbol{h},\tau'}) \\
&= \mathbf{0},
\end{aligned}
\tag{58}
$$

which follows from the assumptions about the SSM. Then

$$\mathbf{\Sigma}_{t-1|t-1} = \operatorname{diag}(\mathbf{\Sigma}_{\boldsymbol{\ell},t-1|t-1}, \; \mathbf{C}_{t-1}^{\mathsf{T}} \mathbf{C}_{t-1}). \tag{59}$$

and

$$
\begin{aligned}
\mathbf{\Sigma}_{t|t-1} &= \mathbf{\Sigma}_{t-1|t-1} + \mathbf{Q}_t \\
&= \operatorname{diag}(\mathbf{\Sigma}_{\boldsymbol{\ell},t-1|t-1} + q_{\boldsymbol{\ell}} \mathbf{I}_{D_{\boldsymbol{\ell}}}, \; \mathbf{C}_{t-1}^{\mathsf{T}} \mathbf{C}_{t-1} + q_{\boldsymbol{h}} \mathbf{I}_{D_{\boldsymbol{h}}}).
\end{aligned}
\tag{60}
$$

Given that $\mathbf{\Sigma}_{\boldsymbol{\ell},t-1|t-1}$ is approximated at $t-1$ through $\hat{\mathbf{\Sigma}}_{\boldsymbol{\ell},t-1}$ and $\mathbf{\Sigma}_{\boldsymbol{h},t-1|t-1}$ is approximated at $t-1$ through $\mathbf{C}_t^{\mathsf{T}} \mathbf{C}_t$, we obtain

$$
\begin{aligned}
\mathbf{\Sigma}_{t|t-1,\boldsymbol{\ell}} &\approx \hat{\mathbf{\Sigma}}_{\boldsymbol{\ell},t-1} + q_{\boldsymbol{\ell}} \mathbf{I}_{D_{\boldsymbol{\ell}}}, \\
\mathbf{\Sigma}_{t|t-1,\boldsymbol{h}} &\approx \mathbf{C}_{t-1}^{\mathsf{T}} \mathbf{C}_{t-1} + q_{\boldsymbol{h}} \mathbf{I}_{D_{\boldsymbol{\ell}}}.
\end{aligned}
\tag{61}
$$

$\square$

## E.2   Proof of Proposition D.2

*Proof.* By definition of $\boldsymbol{y}_t$ and Proposition B.2, the innovation takes the form

$$
\begin{aligned}
\boldsymbol{\epsilon}_t &= \boldsymbol{y}_t - f(\boldsymbol{\ell}_{t-1|t-1}, \boldsymbol{h}_{t-1|t-1}, \boldsymbol{x}_t) \\
&= \tilde{\mathbf{J}}_t(\boldsymbol{\theta}_t - \boldsymbol{\theta}_{t|t-1}) + \boldsymbol{e}_t \\
&= \tilde{\mathbf{L}}_t(\boldsymbol{\ell}_t - \boldsymbol{\ell}_{t|t-1}) + \tilde{\mathbf{H}}_t(\boldsymbol{h}_t - \boldsymbol{h}_{t|t-1}) + \boldsymbol{e}_t.
\end{aligned}
\tag{62}
$$

Next, the variance of the innovation is

$$
\begin{aligned}
\mathbf{S}_t &= \mathrm{Var}(\boldsymbol{\epsilon}_t) \\
&= \mathrm{Var}\left(\tilde{\mathbf{L}}_t\left(\boldsymbol{\ell}_t - \boldsymbol{\ell}_{t|t-1}\right) + \tilde{\mathbf{H}}_t\left(\boldsymbol{h}_t - \boldsymbol{h}_{t|t-1}\right) + \boldsymbol{e}_t\right) \\
&= \tilde{\mathbf{L}}_t\,\mathrm{Var}(\boldsymbol{\ell}_t - \boldsymbol{\ell}_{t|t-1})\tilde{\mathbf{L}}_t^{\mathsf{T}} + \tilde{\mathbf{H}}_t\,\mathrm{Var}(\boldsymbol{h}_t - \boldsymbol{h}_{t|t-1})\tilde{\mathbf{H}}_t^{\mathsf{T}} + \mathrm{Var}(\boldsymbol{e}_t) \\
&= \tilde{\mathbf{L}}_t\,\boldsymbol{\Sigma}_{\boldsymbol{\ell},t|t-1}\,\tilde{\mathbf{L}}_t^{\mathsf{T}} + \tilde{\mathbf{H}}_t\,\boldsymbol{\Sigma}_{\boldsymbol{h},t|t-1}\,\tilde{\mathbf{H}}_t^{\mathsf{T}} + \mathbf{R}_t \\
&= \tilde{\mathbf{L}}_t\,\hat{\boldsymbol{\Sigma}}_{\boldsymbol{\ell},t-1}\,\tilde{\mathbf{L}}_t^{\mathsf{T}} + q_{\boldsymbol{\ell},t}\,\tilde{\mathbf{L}}_t\,\tilde{\mathbf{L}}_t^{\mathsf{T}} + \tilde{\mathbf{H}}_t\,\mathbf{C}_t^{\mathsf{T}}\,\mathbf{C}_t\tilde{\mathbf{H}}_t^{\mathsf{T}} + q_{\boldsymbol{h},t}\tilde{\mathbf{H}}_t\,\tilde{\mathbf{H}}_t^{\mathsf{T}} + \mathbf{R}_t \\
&= \begin{bmatrix} \tilde{\mathbf{L}}\,\hat{\boldsymbol{\Sigma}}_{\boldsymbol{\ell},t-1}^{\mathsf{T}/2} & \sqrt{q_{\boldsymbol{\ell},t}}\,\tilde{\mathbf{L}}_t & \tilde{\mathbf{H}}\,\mathbf{C}_{t-1}^{\mathsf{T}} & \sqrt{q_{\boldsymbol{h},t}}\,\tilde{\mathbf{H}}_t & \mathbf{R}^{\mathsf{T}/2} \end{bmatrix} \begin{bmatrix} \tilde{\mathbf{L}}\,\hat{\boldsymbol{\Sigma}}_{\boldsymbol{\ell},t-1}^{1/2} \\ \sqrt{q_{\boldsymbol{\ell},t}}\,\tilde{\mathbf{L}}_t^{\mathsf{T}} \\ \mathbf{C}_{t-1}\,\tilde{\mathbf{H}}^{\mathsf{T}} \\ \sqrt{q_{\boldsymbol{\ell},t}}\,\tilde{\mathbf{L}}_t^{\mathsf{T}} \\ \mathbf{R}_t^{1/2} \end{bmatrix} \\
&= \mathbf{S}_t^{\mathsf{T}/2}\,\mathbf{S}_t^{1/2},
\end{aligned}
\tag{63}
$$

with

$$
\mathbf{S}_t^{1/2} = \mathcal{Q}_R\left(\boldsymbol{\Sigma}_{\boldsymbol{\ell},t|t}^{1/2}\,\tilde{\mathbf{L}}_t^{\mathsf{T}},\ \sqrt{q_{\boldsymbol{\ell},t}}\,\tilde{\mathbf{L}}_t^{\mathsf{T}},\ \mathbf{C}_{t-1}\,\tilde{\mathbf{H}}_t^{\mathsf{T}},\ \sqrt{q_{\boldsymbol{h},t}}\,\tilde{\mathbf{H}}_t^{\mathsf{T}},\ \mathbf{R}_t^{1/2}\right).
\tag{64}
$$

$\square$

### E.3   Proof of Proposition D.3

*Proof.* First, by Proposition B.2, the Kalman gain matrix is given by

$$
\mathbf{K}_t = \boldsymbol{\Sigma}_{t|t-1}\,\tilde{\mathbf{J}}_t^{\mathsf{T}}\,\mathbf{S}_t^{-1}.
\tag{65}
$$

Next, following Proposition D.1 and Proposition D.2, we rewrite the Kalman gain as

$$
\begin{aligned}
\mathbf{K}_t &= \mathrm{diag}(\boldsymbol{\Sigma}_{\boldsymbol{\ell},t|t-1}\,\mathbf{C}_{t-1}^{\mathsf{T}}\,\mathbf{C}_{t-1})\begin{bmatrix} \tilde{\mathbf{L}}_t \\ \tilde{\mathbf{H}}_t \end{bmatrix}\mathbf{S}_t^{-1} \\
&= \mathrm{diag}(\boldsymbol{\Sigma}_{\boldsymbol{\ell},t|t-1}\,\mathbf{C}_{t-1}^{\mathsf{T}}\,\mathbf{C}_{t-1})\begin{bmatrix} \tilde{\mathbf{L}}_t\,\mathbf{S}_t^{-1} \\ \tilde{\mathbf{H}}_t\,\mathbf{S}_t^{-1} \end{bmatrix} \\
&= \begin{bmatrix} \boldsymbol{\Sigma}_{\boldsymbol{\ell},t|t-1}\,\tilde{\mathbf{L}}_t\,\mathbf{S}_t^{-1} \\ \mathbf{C}_{t-1}^{\mathsf{T}}\,\mathbf{C}_{t-1}\,\tilde{\mathbf{L}}_t\,\mathbf{S}_t^{-1} \end{bmatrix} \\
&= \begin{bmatrix} \mathbf{K}_{\boldsymbol{\ell},t} \\ \mathbf{K}_{\boldsymbol{h}_t} \end{bmatrix},
\end{aligned}
\tag{66}
$$

where

$$
\begin{aligned}
\mathbf{K}_{\boldsymbol{\ell},t} &:= \left(\mathbf{L}_t^{-1}\,\mathbf{L}_t^{-\mathsf{T}}\tilde{\mathbf{L}}_t\,\boldsymbol{\Sigma}_{\boldsymbol{\ell},t|t-1}\right)^{\mathsf{T}}, \\
\mathbf{K}_{\boldsymbol{h},t} &= \left(\mathbf{L}_t^{-1}\,\mathbf{L}_t^{-\mathsf{T}}\tilde{\mathbf{H}}_t\,\mathbf{W}_{t-1}\,\mathbf{W}_{t-1}^{\mathsf{T}}\right)^{\mathsf{T}}.
\end{aligned}
\tag{67}
$$

From the above and Proposition D.1 the updated covariance matrix takes the form

$$
\boldsymbol{\Sigma}_{t|t} = \left(\mathbf{I} - \mathbf{K}_t\,\tilde{\mathbf{J}}_t\right)\boldsymbol{\Sigma}_{t|t-1}\left(\mathbf{I} - \mathbf{K}_t\tilde{\mathbf{J}}_t\right)^{\mathsf{T}} + \mathbf{K}_t\,\mathbf{R}_t\,\mathbf{K}_t^{\mathsf{T}},
\tag{68}
$$

where

$$
\begin{aligned}
\mathbf{I} &- \mathbf{K}_t\,\tilde{\mathbf{J}}_t \\
&= \begin{bmatrix} \mathbf{I}_{D_\ell} & \mathbf{0} \\ \mathbf{0} & \mathbf{I}_{D_h} \end{bmatrix} - \begin{bmatrix} \mathbf{K}_{\boldsymbol{\ell},t} \\ \mathbf{K}_{\boldsymbol{h},t} \end{bmatrix}\begin{bmatrix} \tilde{\mathbf{L}}_t & \tilde{\mathbf{H}}_t \end{bmatrix} \\
&= \begin{bmatrix} \mathbf{I}_{D_\ell} & \mathbf{0} \\ \mathbf{0} & \mathbf{I}_{D_h} \end{bmatrix} - \begin{bmatrix} \mathbf{K}_{\boldsymbol{\ell},t}\,\tilde{\mathbf{L}}_t & \mathbf{K}_{\boldsymbol{\ell},t}\,\tilde{\mathbf{H}}_t \\ \mathbf{K}_{\boldsymbol{h},t}\,\tilde{\mathbf{L}}_t & \mathbf{K}_{\boldsymbol{h},t}\,\tilde{\mathbf{H}}_t \end{bmatrix} \\
&= \begin{bmatrix} \mathbf{I}_{D_\ell} - \mathbf{K}_{\boldsymbol{\ell},t}\,\tilde{\mathbf{L}}_t & \mathbf{K}_{\boldsymbol{\ell},t}\,\tilde{\mathbf{H}}_t \\ \mathbf{K}_{\boldsymbol{h},t}\,\tilde{\mathbf{L}}_t & \mathbf{I}_{D_h} - \mathbf{K}_{\boldsymbol{h},t}\,\tilde{\mathbf{H}}_t \end{bmatrix},
\end{aligned}
\tag{69}
$$

the predicted error covariance $\boldsymbol{\Sigma}_{t|t-1}$ is

$$\boldsymbol{\Sigma}_{t|t-1} = \begin{bmatrix} \boldsymbol{\Sigma}_{\boldsymbol{\ell},t|t-1} & \mathbf{0} \\ \mathbf{0} & \boldsymbol{\Sigma}_{\boldsymbol{h},t|t-1} \end{bmatrix}, \tag{70}$$

and

$$\begin{aligned} \mathbf{K}_t\,\mathbf{R}_t\,\mathbf{K}_t^{\mathsf{T}} &= \begin{bmatrix} \mathbf{K}_{\boldsymbol{\ell},t} \\ \mathbf{K}_{\boldsymbol{h},t} \end{bmatrix} \mathbf{R}_t \begin{bmatrix} \mathbf{K}_{\boldsymbol{\ell},t}^{\mathsf{T}} & \mathbf{K}_{\boldsymbol{h},t}^{\mathsf{T}} \end{bmatrix} \\ &= \begin{bmatrix} \mathbf{K}_{\boldsymbol{\ell},t}\,\mathbf{R}_t\,\mathbf{K}_{\boldsymbol{\ell},t}^{\mathsf{T}} & \mathbf{K}_{\boldsymbol{\ell},t}\,\mathbf{R}_t\,\mathbf{K}_{\boldsymbol{h},t}^{\mathsf{T}} \\ \mathbf{K}_{\boldsymbol{h},t}\,\mathbf{R}_t\,\mathbf{K}_{\boldsymbol{\ell},t}^{\mathsf{T}} & \mathbf{K}_{\boldsymbol{h},t}\,\mathbf{R}_t\,\mathbf{K}_{\boldsymbol{h},t}^{\mathsf{T}} \end{bmatrix}. \end{aligned} \tag{71}$$

Thus,

$$\boldsymbol{\Sigma}_{t|t}$$
$$= \begin{bmatrix} (\mathbf{I}_{D_{\boldsymbol{\ell}}} - \mathbf{K}_{\boldsymbol{\ell},t}\,\tilde{\mathbf{L}}_t)\boldsymbol{\Sigma}_{\boldsymbol{\ell},t|t-1}(\mathbf{I}_{D_{\boldsymbol{\ell}}} - \mathbf{K}_{\boldsymbol{\ell},t}\,\tilde{\mathbf{L}}_t)^{\mathsf{T}} + \mathbf{K}_{\boldsymbol{\ell},t}\,\mathbf{R}_t\,\mathbf{K}_{\boldsymbol{\ell},t}^{\mathsf{T}} & \mathbf{K}_{\boldsymbol{\ell},t}\,\mathbf{R}_t\,\mathbf{K}_{\boldsymbol{h},t}^{\mathsf{T}} \\ \mathbf{K}_{\boldsymbol{h},t}\,\mathbf{R}_t\,\mathbf{K}_{\boldsymbol{\ell},t}^{\mathsf{T}} & (\mathbf{I}_{D_{\boldsymbol{h}}} - \mathbf{K}_{\boldsymbol{h},t}\,\tilde{\mathbf{H}}_t)\boldsymbol{\Sigma}_{\boldsymbol{h},t|t-1}(\mathbf{I}_{D_{\boldsymbol{h}}} - \mathbf{K}_{\boldsymbol{h},t}\,\tilde{\mathbf{H}}_t)^{\mathsf{T}} + \mathbf{K}_{\boldsymbol{h},t}\,\mathbf{R}_t\,\mathbf{K}_{\boldsymbol{h},t}^{\mathsf{T}} \end{bmatrix}. \tag{72}$$

Next, we consider the following surrogate covariance matrix which modifies the second block-diagonal entry

$$\tilde{\boldsymbol{\Sigma}}_{t|t} =$$
$$\begin{bmatrix} (\mathbf{I}_{D_{\boldsymbol{\ell}}} - \mathbf{K}_{\boldsymbol{\ell},t}\tilde{\mathbf{L}}_t)\boldsymbol{\Sigma}_{\boldsymbol{\ell},t-1|t-1}(\mathbf{I}_{D_{\boldsymbol{\ell}}} - \mathbf{K}_{\boldsymbol{\ell},t}\tilde{\mathbf{L}}_t)^{\mathsf{T}} + \mathbf{K}_{\boldsymbol{\ell},t}\mathbf{R}_t\mathbf{K}_{\boldsymbol{\ell},t}^{\mathsf{T}} & \mathbf{K}_{\boldsymbol{\ell},t}\mathbf{R}_t\mathbf{K}_{\boldsymbol{h},t}^{\mathsf{T}} \\ \mathbf{K}_{\boldsymbol{h},t}\mathbf{R}_t\mathbf{K}_{\boldsymbol{\ell},t}^{\mathsf{T}} & (\mathbf{I}_{D_{\boldsymbol{h}}} - \mathbf{K}_{\boldsymbol{h},t}\tilde{\mathbf{H}}_t)\mathbf{C}_{t-1}^{\mathsf{T}}\mathbf{C}_{t-1}(\mathbf{I}_{D_{\boldsymbol{h}}} - \mathbf{K}_{\boldsymbol{h},t}\tilde{\mathbf{H}}_t)^{\mathsf{T}} + \mathbf{K}_{\boldsymbol{h},t}\mathbf{R}_t\mathbf{K}_{\boldsymbol{h},t}^{\mathsf{T}} + q_{\boldsymbol{h},t}\mathbf{I}_{D_{\boldsymbol{h}}} \end{bmatrix}. \tag{73}$$

It then follows that

$$\begin{aligned} \hat{\boldsymbol{\Sigma}}_{\boldsymbol{\ell},t}, \hat{\boldsymbol{\Sigma}}_{\boldsymbol{h},t} &= \underset{\boldsymbol{\Sigma}_{\boldsymbol{\ell}}, \boldsymbol{\Sigma}_{\boldsymbol{h}}: \mathrm{rank}(\boldsymbol{\Sigma}_{\boldsymbol{h}})=d}{\arg\min} \left\| \tilde{\boldsymbol{\Sigma}}_{t|t} - \begin{bmatrix} \boldsymbol{\Sigma}_{\boldsymbol{\ell}} & \mathbf{0} \\ \mathbf{0} & \boldsymbol{\Sigma}_{\boldsymbol{h}} \end{bmatrix} \right\|_{\mathrm{F}}^2 \\ &= \underset{\boldsymbol{\Sigma}_{\boldsymbol{\ell}}, \boldsymbol{\Sigma}_{\boldsymbol{h}}: \mathrm{rank}(\boldsymbol{\Sigma}_{\boldsymbol{h}})=d}{\arg\min} \|\tilde{\boldsymbol{\Sigma}}_{\boldsymbol{\ell},t} - \boldsymbol{\Sigma}_{\boldsymbol{\ell}}\|_{\mathrm{F}}^2 + \|\tilde{\boldsymbol{\Sigma}}_{\boldsymbol{h},t} - \boldsymbol{\Sigma}_{\boldsymbol{h}}\|_{\mathrm{F}}^2 + 2\,\|\mathbf{K}_{\boldsymbol{\ell},t}\,\mathbf{R}_t\,\mathbf{K}_{\boldsymbol{h},t}^{\mathsf{T}}\|_{\mathrm{F}}^2, \end{aligned} \tag{74}$$

with

$$\begin{aligned} \tilde{\boldsymbol{\Sigma}}_{\boldsymbol{\ell},t} &= (\mathbf{I}_{D_{\boldsymbol{\ell}}} - \mathbf{K}_{\boldsymbol{\ell},t}\,\tilde{\mathbf{L}}_t)\boldsymbol{\Sigma}_{\boldsymbol{\ell},t-1|t-1}(\mathbf{I}_{D_{\boldsymbol{\ell}}} - \mathbf{K}_{\boldsymbol{\ell},t}\,\tilde{\mathbf{L}}_t)^{\mathsf{T}} + \mathbf{K}_{\boldsymbol{\ell},t}\,\mathbf{R}_t\,\mathbf{K}_{\boldsymbol{\ell},t}^{\mathsf{T}}, \\ \tilde{\boldsymbol{\Sigma}}_{\boldsymbol{h},t} &= (\mathbf{I}_{D_{\boldsymbol{h}}} - \mathbf{K}_{\boldsymbol{h},t}\,\tilde{\mathbf{H}}_t)\mathbf{C}_{t-1}^{\mathsf{T}}\,\mathbf{C}_{t-1}(\mathbf{I}_{D_{\boldsymbol{h}}} - \mathbf{K}_{\boldsymbol{h},t}\,\tilde{\mathbf{H}}_t)^{\mathsf{T}} + \mathbf{K}_{\boldsymbol{h},t}\,\mathbf{R}_t\,\mathbf{K}_{\boldsymbol{h},t}^{\mathsf{T}} + q_{\boldsymbol{h},t}\mathbf{I}_{D_{\boldsymbol{h}}}. \end{aligned} \tag{75}$$

Then, $\hat{\boldsymbol{\Sigma}}_{\boldsymbol{\ell},t} = \tilde{\boldsymbol{\Sigma}}_{\boldsymbol{\ell},t}$ and $\hat{\boldsymbol{\Sigma}}_{\boldsymbol{h},t}$ is given by the first $d_{\boldsymbol{h}}$ principal components of $\tilde{\boldsymbol{\Sigma}}_{\boldsymbol{h},t}$.

Next, we find the update rule for the Cholesky and low-rank factors. First, the EVC for the last layer takes the form

$$\begin{aligned} \hat{\boldsymbol{\Sigma}}_{\boldsymbol{\ell},t} &= \left(\mathbf{I} - \mathbf{K}_{\boldsymbol{\ell},t}\,\tilde{\mathbf{L}}_t\right)\boldsymbol{\Sigma}_{\boldsymbol{\ell},t|t-1}\left(\mathbf{I} - \mathbf{K}_{\boldsymbol{\ell},t}\,\tilde{\mathbf{L}}_t\right)^{\mathsf{T}} + \mathbf{K}_{\boldsymbol{\ell},t}\,\mathbf{R}_t\mathbf{K}_{\boldsymbol{\ell},t}^{\mathsf{T}} \\ &= \left(\mathbf{I} - \mathbf{K}_{\boldsymbol{\ell},t}\,\tilde{\mathbf{L}}_t\right)\boldsymbol{\Sigma}_{\boldsymbol{\ell},t|t-1}\left(\mathbf{I} - \mathbf{K}_{\boldsymbol{\ell},t}\,\tilde{\mathbf{L}}_t\right)^{\mathsf{T}} + \mathbf{K}_{\boldsymbol{\ell},t}\,\mathbf{R}_t^{\mathsf{T}/2}\,\mathbf{R}_t^{1/2}\mathbf{K}_{\boldsymbol{\ell},t}^{\mathsf{T}} \\ &= \left[\left(\mathbf{I} - \mathbf{K}_{\boldsymbol{\ell},t}\,\tilde{\mathbf{L}}_t\right)\boldsymbol{\Sigma}_{\boldsymbol{\ell},t-1|t-1}^{\mathsf{T}/2} \quad \mathbf{K}_{\boldsymbol{\ell},t}\,\mathbf{R}_t^{\mathsf{T}/2}\right] \begin{bmatrix} \boldsymbol{\Sigma}_{\boldsymbol{\ell},t|t-1}^{1/2}\left(\mathbf{I} - \mathbf{K}_{\boldsymbol{\ell},t}\,\tilde{\mathbf{L}}_t\right)^{\mathsf{T}} \\ \mathbf{R}_t^{1/2}\mathbf{K}_{\boldsymbol{\ell},t}^{\mathsf{T}} \end{bmatrix}, \end{aligned} \tag{76}$$

so that

$$\hat{\boldsymbol{\Sigma}}_{\boldsymbol{\ell},t}^{1/2} = \mathcal{Q}_R\left(\boldsymbol{\Sigma}_{\boldsymbol{\ell},t-1|t-1}^{1/2}\left(\mathbf{I} - \mathbf{K}_{\boldsymbol{\ell},t}\,\tilde{\mathbf{L}}_t\right)^{\mathsf{T}}, \mathbf{R}_t^{1/2}\mathbf{K}_{\boldsymbol{\ell},t}^{\mathsf{T}}\right). \tag{77}$$

Then, the EVC covariance for the hidden layers (lower right block-diagonal) is

$$\begin{aligned} \tilde{\boldsymbol{\Sigma}}_{\boldsymbol{h},t} &= \left(\mathbf{I} - \mathbf{K}_{\boldsymbol{h},t}\,\tilde{\mathbf{H}}_t\right)\mathbf{C}_{t-1}^{\mathsf{T}}\,\mathbf{C}_{t-1}\left(\mathbf{I} - \mathbf{K}_{\boldsymbol{h},t}\,\tilde{\mathbf{H}}_t\right)^{\mathsf{T}} + \mathbf{K}_{\boldsymbol{h},t}\,\mathbf{R}_t^{\mathsf{T}/2}\,\mathbf{R}_t^{1/2}\mathbf{K}_{\boldsymbol{h},t}^{\mathsf{T}} + q_{\boldsymbol{h},t}\,\mathbf{I}_{D_{\boldsymbol{h}}} \\ &= \left[\left(\mathbf{I} - \mathbf{K}_{\boldsymbol{h},t}\,\tilde{\mathbf{H}}_t\right)\mathbf{C}_{t-1}^{\mathsf{T}} \quad \mathbf{K}_{\boldsymbol{h},t}\,\mathbf{R}_t^{\mathsf{T}/2}\right] \begin{bmatrix} \mathbf{C}_{t-1}\left(\mathbf{I} - \mathbf{K}_{\boldsymbol{h},t}\,\tilde{\mathbf{H}}_t\right)^{\mathsf{T}} \\ \mathbf{R}_t^{1/2}\mathbf{K}_{\boldsymbol{h},t}^{\mathsf{T}} \end{bmatrix} + q_{\boldsymbol{h},t}\,\mathbf{I}_{D_{\boldsymbol{h}}} \\ &= \tilde{\mathbf{C}}_t^{\mathsf{T}}\,\tilde{\mathbf{C}}_t + q_{\boldsymbol{h},t}\,\mathbf{I}_{D_{\boldsymbol{h}}}. \end{aligned} \tag{78}$$

Finally, by Corollary C.3, the best rank-$d$ low-rank factor is

$$\mathbf{C}_t = \mathcal{P}_{d_{\boldsymbol{h}},+q_{\boldsymbol{h},t}} \left( \mathbf{C}_{t-1} \left( \mathbf{I} - \mathbf{K}_{\boldsymbol{h},t}\, \tilde{\mathbf{H}}_t \right)^{\mathsf{T}}, \mathbf{R}_t^{1/2} \mathbf{K}_{\boldsymbol{h},t}^{\mathsf{T}} \right) \in \mathcal{M}_{(d_{\boldsymbol{h}}+D_{\boldsymbol{y}}) \times D_{\boldsymbol{h}}}(\mathbb{R}). \tag{79}$$

So that

$$\hat{\boldsymbol{\Sigma}}_{\boldsymbol{h},t} = \mathbf{C}_t^{\mathsf{T}} \mathbf{C}_t. \tag{80}$$

$\square$

### E.4   Proof of Proposition 4.1

*Proof.*  We seek to bound

$$\|\boldsymbol{\Sigma}_{t|t} - \hat{\boldsymbol{\Sigma}}_{t|t}\|_{\mathrm{F}}. \tag{81}$$

Following the arguments from Proposition F.3, we obtain

$$\|\boldsymbol{\Sigma}_{t|t} - \hat{\boldsymbol{\Sigma}}_{t|t}\|_{\mathrm{F}} \leq \| \underbrace{\boldsymbol{\Sigma} - \tilde{\boldsymbol{\Sigma}}_{t|t}}_{=\mathbf{E}_{\mathrm{surr}}} \|_{\mathrm{F}} + \| \underbrace{\tilde{\boldsymbol{\Sigma}}_{t|t} - \hat{\boldsymbol{\Sigma}}_{t|t}}_{=\mathbf{E}_{\mathrm{proj}}} \|_{\mathrm{F}}, \tag{82}$$

where

$$\mathbf{E}_{\mathrm{surr}} = q_{\boldsymbol{h},t} \left( 2\|\mathbf{K}_{\boldsymbol{h},t}\tilde{\mathbf{H}}_t\|_{\mathrm{F}} + \|\mathbf{K}_{\boldsymbol{h},t}\tilde{\mathbf{H}}_t\|_{\mathrm{F}}^2 \right) + q_{\boldsymbol{\ell},t} \left\| \left( \mathbf{I} - \mathbf{K}_{\boldsymbol{\ell},t}\, \tilde{\mathbf{L}}_t \right) \left( \mathbf{I} - \mathbf{K}_{\boldsymbol{\ell},t}\, \tilde{\mathbf{L}}_t \right)^{\mathsf{T}} \right\|_{\mathrm{F}} \tag{83}$$

and

$$\mathbf{E}_{\mathrm{proj}} = \|\tilde{\boldsymbol{\Sigma}}_{\boldsymbol{h},t} - \mathbf{C}_t^{\mathsf{T}} \mathbf{C}_t\|_{\mathrm{F}} + 2\,\|\mathbf{K}_{\boldsymbol{\ell},t}\, \mathbf{R}_t\, \mathbf{K}_{\boldsymbol{h},t}^{\mathsf{T}}\|_{\mathrm{F}} \tag{84}$$

Given that $\mathbf{C}_t^{\mathsf{T}} \mathbf{C}_t$ is the best rank-$d$ approximation of $\tilde{\boldsymbol{\Sigma}}_{\boldsymbol{h},t}$, it follows from a similar derivation to (107) that

$$\|\tilde{\boldsymbol{\Sigma}}_{\boldsymbol{h},t} - \mathbf{C}_t^{\mathsf{T}} \mathbf{C}_t\|_{\mathrm{F}} = \sqrt{\sum_{k=d_{\boldsymbol{h}}+1}^{D_{\boldsymbol{h}}} \lambda_k^2}, \tag{85}$$

where $\{\lambda_k\}_{k=d_{\boldsymbol{h}}+1}^{D_{\boldsymbol{h}}}$ are bottom $(D_{\boldsymbol{h}} - d_{\boldsymbol{h}})$ eigenvalues of $\tilde{\boldsymbol{\Sigma}}_t$.   $\square$

## F   Further algorithms

### F.1   `LoLoFi`

Algorithm 3 shows a single step of `LoLoFi`. The low-rank factor for last layers $\mathbf{C}_{\boldsymbol{\ell},0} \in \mathcal{M}_{d_{\boldsymbol{\ell}} \times D_{\boldsymbol{\ell}}}(\mathbb{R})$ and the low-rank factor for the hidden layers $\mathbf{C}_{\boldsymbol{h},0} \in \mathcal{M}_{d_{\boldsymbol{h}} \times D_{\boldsymbol{h}}}(\mathbb{R})$ are chosen at initialization.

### F.2   Low-rank Kalman filter (`LRKF`)

Inspired by [57], we derive a novel version of the low-rank Kalman filter which is (i) more computationally efficient, (ii) simple to implement, and (iii) provably optimal (in terms of Frobenius norm). We call this the `LRKF` method. Instead of keeping track of $\boldsymbol{\Sigma}_{t|t}$ or $\mathbf{S}_t$, we keep track of a low-rank factor for the covariance, and compute the Cholesky factor for the innovation variance $\mathbf{S}_t^{1/2}$. Algorithm 4 summarizes the predict and update steps for `LRKF`. Next we turn to the derivation.

We assume the model parameters $\boldsymbol{\theta}$ and the observations $\boldsymbol{y}$ follow

$$\begin{aligned} \boldsymbol{\theta}_t &= \boldsymbol{\theta}_{t-1} + \boldsymbol{u}_t, \\ \boldsymbol{y}_t &= \mathbf{H}_t\, \boldsymbol{\theta}_t + \boldsymbol{e}_t, \end{aligned} \tag{86}$$

where $\mathrm{Var}(\boldsymbol{e}_t) = \mathbf{R}_t$ is known and $\mathbf{H}_t$ is the known projection matrix. Both, $\boldsymbol{e}_t$ and $\boldsymbol{u}_t$ are zero-mean random vectors.

In the Kalman filter update equations, the so-called posterior mean and covariances are given by

$$\begin{aligned} \boldsymbol{\mu}_t &= \boldsymbol{\mu}_{t-1} + \mathbf{K}_t \left( \boldsymbol{y}_t - \hat{\boldsymbol{y}}_t \right), \\ \boldsymbol{\Sigma}_t &= \mathrm{Var}(\boldsymbol{\theta}_t - \boldsymbol{\mu}_t), \end{aligned} \tag{87}$$

---

**Algorithm 3** Single step of `LoLoFi` for online learning for $t \geq 1$.

---

**Require:** $\mathbf{R}_t$ // measurement variance
**Require:** $(q_{\boldsymbol{\ell},t}, q_{\boldsymbol{h},t})$ dynamics covariance for last layer and hidden layers
**Require:** $(\boldsymbol{y}_t, \boldsymbol{x}_t)$ // observation and input
**Require:** $b_{t-1} = \left(\boldsymbol{\ell}_{t-1|t-1}, \boldsymbol{h}_{t-1|t-1}, \mathbf{C}_{\boldsymbol{\ell},t-1}, \mathbf{C}_{\boldsymbol{h},t-1}\right)$ // previous belief
    // predict step
1: $\hat{\boldsymbol{y}}_t \leftarrow f(\boldsymbol{\ell}_{t-1|t-1}, \boldsymbol{h}_{t-1|t-1}, \boldsymbol{x}_t)$
2: $\tilde{\mathbf{L}}_t = \nabla_{\boldsymbol{\ell}} f(\boldsymbol{\ell}_{t-1|t-1}, \boldsymbol{h}_{t-1|t-1}, \boldsymbol{x}_t)$
3: $\tilde{\mathbf{H}}_t = \nabla_{\boldsymbol{h}} f(\boldsymbol{\ell}_{t-1|t-1}, \boldsymbol{h}_{t-1|t-1}, \boldsymbol{x}_t)$
    // innovation(one-step-ahead error) and Cholesky innovation variance
4: $\boldsymbol{\epsilon}_t \leftarrow \boldsymbol{y}_t - \hat{\boldsymbol{y}}_t$
5: $\mathbf{S}_t^{1/2} \leftarrow \mathcal{Q}_R\left(\mathbf{C}_{\boldsymbol{\ell},t-1}\tilde{\mathbf{L}}_t^{\intercal}, \sqrt{q_{\boldsymbol{\ell},t}}\tilde{\mathbf{L}}_t^{\intercal}, \mathbf{C}_{\boldsymbol{h},t-1}\tilde{\mathbf{H}}_t^{\intercal}, \sqrt{q_{\boldsymbol{h},t}}\tilde{\mathbf{H}}_t^{\intercal}, \mathbf{R}_t^{1/2}\right)$,
    // gain for hidden layers
6: $\mathbf{V}_{\boldsymbol{h},t} \leftarrow \mathbf{S}_t^{-1/2}\mathbf{S}_t^{-\intercal/2}\tilde{\mathbf{H}}_t$
7: $\mathbf{K}_{\boldsymbol{h},t}^{\intercal} \leftarrow \mathbf{V}_{\boldsymbol{h},t}\mathbf{C}_{\boldsymbol{h},t-1}\mathbf{C}_{\boldsymbol{h},t-1}^{\intercal} + q_{\boldsymbol{h},t}\mathbf{V}_{\boldsymbol{h},t}$
    // gain for last layer
8: $\mathbf{V}_{\boldsymbol{\ell},t} \leftarrow \mathbf{S}_t^{-1/2}\mathbf{S}_t^{-\intercal/2}\tilde{\mathbf{L}}_t$
9: $\mathbf{K}_{\boldsymbol{\ell},t}^{\intercal} \leftarrow \mathbf{V}_{\boldsymbol{\ell},t}\boldsymbol{\Sigma}_{\boldsymbol{\ell},t|t-1} + q_{\boldsymbol{\ell},t}\mathbf{V}_{\boldsymbol{\ell},t}$
10: $\mathbf{K}_{\boldsymbol{\ell},t}^{\intercal} \leftarrow \mathbf{V}_{\boldsymbol{\ell},t}\mathbf{C}_{\boldsymbol{\ell},t-1}\mathbf{C}_{\boldsymbol{\ell},t-1}^{\intercal} + q_{\boldsymbol{\ell},t}\mathbf{V}_{\boldsymbol{\ell},t}$
    // mean update step
11: $\boldsymbol{h}_{t|t} \leftarrow \boldsymbol{h}_{t-1|t-1} + \mathbf{K}_{\boldsymbol{h},t}\boldsymbol{\epsilon}_t$
12: $\boldsymbol{\ell}_{t|t} \leftarrow \boldsymbol{\ell}_{t-1|t-1} + \mathbf{K}_{\boldsymbol{\ell},t}\boldsymbol{\epsilon}_t$
    // low-rank updates
13: $\mathbf{C}_{\boldsymbol{\ell},t} \leftarrow \mathcal{P}_{d_{\boldsymbol{\ell}},+q_{\boldsymbol{\ell},t}}\left(\mathbf{C}_{\boldsymbol{\ell},t-1}\left(\mathbf{I} - \mathbf{K}_{\boldsymbol{\ell},t}\tilde{\mathbf{L}}_t\right)^{\intercal}, \mathbf{R}_t^{1/2}\mathbf{K}_{\boldsymbol{\ell},t}^{\intercal}\right)$
14: $\mathbf{C}_{\boldsymbol{h},t} \leftarrow \mathcal{P}_{d_{\boldsymbol{h}},+q_{\boldsymbol{h},t}}\left(\mathbf{C}_{\boldsymbol{h},t-1}\left(\mathbf{I} - \mathbf{K}_{\boldsymbol{h},t}\tilde{\mathbf{H}}_t\right)^{\intercal}, \mathbf{R}_t^{1/2}\mathbf{K}_{\boldsymbol{h},t}^{\intercal}\right)$
15: **Return** $b_t = \left(\boldsymbol{\ell}_{t|t}, \boldsymbol{h}_{t|t}, \mathbf{C}_{\boldsymbol{\ell},t}, \mathbf{C}_{\boldsymbol{h},t}\right)$

---

where $\boldsymbol{\mu}_0$ is given,

$$
\begin{aligned}
\hat{\boldsymbol{y}}_t &= \mathbf{H}_t\,\boldsymbol{\mu}_{t|t-1} = \mathbf{H}_t\,\boldsymbol{\mu}_{t-1|t-1}, \\
\mathbf{K}_t &= \mathrm{Cov}(\boldsymbol{\theta}_t, \boldsymbol{\epsilon}_t)\mathrm{Var}(\boldsymbol{\epsilon}_t)^{-1} = \boldsymbol{\Sigma}_{t|t-1}\mathbf{H}_t^{\intercal}\mathbf{S}_t^{-1}, \\
\boldsymbol{\epsilon}_t &= \boldsymbol{y}_t - \mathbf{H}_t\,\boldsymbol{\mu}_{t|t-1}, \\
\mathbf{S}_t &= \mathbf{H}_t\,\boldsymbol{\Sigma}_{t|t-1}\mathbf{H}_t^{\intercal} + \mathbf{R}_t, \\
\mathbf{R}_t &= \mathrm{Var}(\boldsymbol{e}_t).
\end{aligned}
\tag{88}
$$

The memory requirements of the posterior covariance is $O(D^2)$, with $D$ number of parameters. This makes it unfeasible to store for moderately-sized neural networks.

Thus, to reduce the computational cost, we maintain the best rank $d$ approximation of the covariance matrix by

$$
\boldsymbol{\Sigma}_{t|t} \approx \mathbf{W}_t^{\intercal}\,\mathbf{W}_t,
\tag{89}
$$

where $\mathbf{W}_t \in \mathbb{R}^{d \times D}$ is the best rank-$d$ approximation to $\boldsymbol{\Sigma}_t$ (in a Frobenius norm sense). Furthermore, to maintain a numerically stable method, we work with the Cholesky of the innovation variance. In effect, this is a variant square-root low-rank Kalman filter method. We explain the details of this method below.

The predict step equations are given by

$$
\boldsymbol{\mu}_{t|t-1} = \boldsymbol{\mu}_{t-1},
\tag{90}
$$
$$
\boldsymbol{\Sigma}_{t|t-1} = \mathbf{C}_{t-1}^{\intercal}\,\mathbf{C}_{t-1} + q_t\mathbf{I}.
\tag{91}
$$

**Proposition F.1** (Kalman gain and innovations). *The variance $\mathbf{S}_t$ of the innovation is*

$$
\begin{aligned}
\mathbf{S}_t &= \mathbf{H}_t\,\mathbf{\Sigma}_{t|t-1}\,\mathbf{H}_t^\intercal + \mathbf{R}_t \\
&= \mathbf{H}_t\,(\mathbf{C}_{t-1}^\intercal\,\mathbf{C}_{t-1} + q_t\mathbf{I})\,\mathbf{H}_t^\intercal + \mathbf{R}_t \\
&= \mathbf{H}_t\,\mathbf{C}_t^\intercal\,\mathbf{C}_t\,\mathbf{H}_t^\intercal + q_t\,\mathbf{H}_t\,\mathbf{H}_t^\intercal + \mathbf{R}_t \\
&= \begin{bmatrix} \mathbf{H}_t\,\mathbf{C}_t^\intercal & \sqrt{q_t}\,\mathbf{H}_t & \mathbf{R}_t^{\intercal/2} \end{bmatrix} \begin{bmatrix} \mathbf{C}_t\,\mathbf{H}_t^\intercal \\ \sqrt{q_t}\mathbf{H}_t^\intercal \\ \mathbf{R}_t^{1/2} \end{bmatrix} \\
&= \mathbf{S}_t^{\intercal/2}\mathbf{S}_t^{1/2},
\end{aligned}
\tag{92}
$$

*with $\mathbf{S}_t^{1/2}$ given by*

$$
\mathbf{S}_t = \mathcal{Q}_R(\mathbf{C}_t\,\mathbf{H}_t^\intercal,\ \sqrt{q_t}\,\mathbf{H}_t^\intercal,\ \mathbf{R}_t^{1/2}),
\tag{93}
$$

*where $\mathcal{Q}_R$ returns the $R$ matrix from the QR decomposition row-stacked arguments and $\mathbf{R}_t^{1/2}$ is the upper-triangular Cholesky decomposition of $\mathbf{R}_t$. The Kalman gain is given by*

$$
\begin{aligned}
\mathbf{K}_t &= \mathbf{\Sigma}_{t|t-1}\,\mathbf{H}_t^\intercal\,\mathbf{S}_t^{-1} \\
&= \left(\mathbf{S}_t^{-1}\,\mathbf{H}_t\,\mathbf{\Sigma}_{t|t-1}\right)^\intercal \\
&= \left(\mathbf{S}_t^{-1}\,\mathbf{S}_t^{-\intercal}\,\mathbf{H}_t\,\left(\mathbf{C}_{t-1}^\intercal\,\mathbf{C}_{t-1} + q_t\,\mathbf{I}_{D_{\boldsymbol{\theta}}}\right)\right)^\intercal.
\end{aligned}
\tag{94}
$$

**Proposition F.2** (Update step). *The updated mean is*

$$
\boldsymbol{\mu}_t = \boldsymbol{\mu}_{t|t-1} + \mathbf{K}_t(\boldsymbol{y}_t - \hat{\boldsymbol{y}}_t).
\tag{95}
$$

*The update covariance $\mathbf{\Sigma}_{t|t}$, approximated through a two-step procedure: first, a surrogate covariance that ignores the cross-term effect of $\mathbf{K}_t\,\mathbf{H}_t$ on the artificial noise covariance $q_t$, and second, a best rank-d approximation (in Frobenious norm) to the surrogate matrix takes the low-rank form*

$$
\hat{\mathbf{\Sigma}}_{t|t} = \mathbf{C}_t^\intercal\,\mathbf{C}_t,
\tag{96}
$$

*with*

$$
\mathbf{C}_t = \mathcal{P}_d\left(\mathbf{C}_{t-1}\,(\mathbf{I} - \mathbf{K}_t\,\mathbf{H}_t)^\intercal,\ \mathbf{R}_t^{1/2}\mathbf{K}_t^\intercal\right) \in \mathcal{M}_{d\times D_{\boldsymbol{\theta}}}(\mathbb{R}),
\tag{97}
$$

*the low-rank (rectangular decomposition) matrix.*

*Proof.* The updated covariance mean follows directly from (18).

The updated covariance is

$$
\begin{aligned}
\mathbf{\Sigma}_t &= (\mathbf{I} - \mathbf{K}_t\,\mathbf{H}_t)\,\mathbf{\Sigma}_{t|t-1}\,(\mathbf{I} - \mathbf{K}_t\,\mathbf{H}_t)^\intercal + \mathbf{K}_t\,\mathbf{R}_t\mathbf{K}_t^\intercal \\
&= (\mathbf{I} - \mathbf{K}_t\,\mathbf{H}_t)\,\left(\mathbf{C}_{t-1}^\intercal\,\mathbf{C}_{t-1} + q_t\,\mathbf{I}_{D_{\boldsymbol{\theta}}}\right)\,(\mathbf{I} - \mathbf{K}_t\,\mathbf{H}_t)^\intercal + \mathbf{K}_t\,\mathbf{R}_t^{\intercal/2}\,\mathbf{R}_t^{1/2}\mathbf{K}_t^\intercal \\
&= (\mathbf{I} - \mathbf{K}_t\,\mathbf{H}_t)\,\left(\mathbf{C}_{t-1}^\intercal\,\mathbf{C}_{t-1} + q_t\,\mathbf{I}_{D_{\boldsymbol{\theta}}}\right)\,(\mathbf{I} - \mathbf{K}_t\,\mathbf{H}_t)^\intercal + \mathbf{K}_t\,\mathbf{R}_t^{\intercal/2}\,\mathbf{R}_t^{1/2}\mathbf{K}_t^\intercal \\
&= \begin{bmatrix} (\mathbf{I} - \mathbf{K}_t\,\mathbf{H}_t)\,\mathbf{C}_{t-1}^\intercal & \mathbf{K}_t\,\mathbf{R}_t^{\intercal/2} \end{bmatrix} \begin{bmatrix} \mathbf{C}_{t-1}\,(\mathbf{I} - \mathbf{K}_t\,\mathbf{H}_t)^\intercal \\ \mathbf{R}_t^{1/2}\mathbf{K}_t^\intercal. \end{bmatrix} + q_t\,(\mathbf{I} - \mathbf{K}_t\,\mathbf{H}_t)\,(\mathbf{I} - \mathbf{K}_t\,\mathbf{H}_t)^\intercal.
\end{aligned}
\tag{98}
$$

Next, we consider the *surrogate* EVC matrix

$$
\begin{aligned}
\tilde{\mathbf{\Sigma}}_t &= \begin{bmatrix} (\mathbf{I} - \mathbf{K}_t\,\mathbf{H}_t)\,\mathbf{C}_{t-1}^\intercal & \mathbf{K}_t\,\mathbf{R}_t^{\intercal/2} \end{bmatrix} \begin{bmatrix} \mathbf{C}_{t-1}\,(\mathbf{I} - \mathbf{K}_t\,\mathbf{H}_t)^\intercal \\ \mathbf{R}_t^{1/2}\mathbf{K}_t^\intercal. \end{bmatrix} + q_t\mathbf{I}_{D_{\boldsymbol{\theta}}} \\
&= \tilde{\mathbf{C}}_t^\intercal\,\tilde{\mathbf{C}}_t + q_t\mathbf{I}_{D_{\boldsymbol{\theta}}},
\end{aligned}
\tag{99}
$$

where

$$
\tilde{\mathbf{C}}_t = \begin{bmatrix} \mathbf{C}_{t-1}\,(\mathbf{I} - \mathbf{K}_t\,\mathbf{H}_t)^\intercal \\ \mathbf{R}_t^{1/2}\mathbf{K}_t^\intercal \end{bmatrix} \in \mathbb{R}^{(d+o)\times D}.
\tag{100}
$$

Next, the best rank-$d$ matrix is

$$\hat{\boldsymbol{\Sigma}}_t = \underset{\boldsymbol{\Sigma}:\mathrm{rank}(\boldsymbol{\Sigma})=d}{\arg\min} \left\| \tilde{\boldsymbol{\Sigma}}_t - \boldsymbol{\Sigma} \right\|_{\mathrm{F}}^2$$

$$= \underset{\boldsymbol{\Sigma}:\mathrm{rank}(\boldsymbol{\Sigma})=d}{\arg\min} \left\| \tilde{\mathbf{C}}_t^\intercal \tilde{\mathbf{C}}_t + q_t\,\mathbf{I}_{D_{\boldsymbol{\theta}}} - \boldsymbol{\Sigma} \right\|_{\mathrm{F}}^2 \tag{101}$$

$$= \mathbf{C}_t^\intercal \mathbf{C}_t,$$

where

$$\mathbf{C}_t = \mathcal{P}_{d,+q_t}\left( \mathbf{C}_{t-1}\,(\mathbf{I} - \mathbf{K}_t\,\mathbf{H}_t)^\intercal,\, \mathbf{R}_t^{1/2}\mathbf{K}_t^\intercal \right) \in \mathcal{M}_{d \times D_{\boldsymbol{\theta}}}(\mathbb{R}) \tag{102}$$

is the best $d$-dimensional low-rank matrix given the stacked matrices $\tilde{\mathbf{C}}_t$ and the dynamics covariance $q_t$.

$\square$

**Proposition F.3.** *The per-step error induced by the approximation of the covariance at time $t$ is bounded by*

$$\|\boldsymbol{\Sigma}_{t|t} - \hat{\boldsymbol{\Sigma}}_{t|t}\|_{\mathrm{F}} \leq q_t\left( 2\,\|\mathbf{K}_t\,\mathbf{H}_t\|_{\mathrm{F}} + \|\mathbf{K}_t\,\mathbf{H}_t\|_{\mathrm{F}}^2 \right) + \sqrt{\sum_{k=d+1}^{D_{\boldsymbol{\theta}}} \lambda_k^2}, \tag{103}$$

*where $\{\lambda_k\}_{k=d+1}^{D_{\boldsymbol{\theta}}}$ are the bottom $(D_{\boldsymbol{\theta}} - d)$ eigenvalues of $\tilde{\boldsymbol{\Sigma}}_t$.*

*Proof.* We seek to bound

$$\|\boldsymbol{\Sigma}_{t|t} - \hat{\boldsymbol{\Sigma}}_{t|t}\|_{\mathrm{F}}, \tag{104}$$

where $\boldsymbol{\Sigma}_{t|t}$ is given by (98) and $\hat{\boldsymbol{\Sigma}}_{t|t}$ is given by (101). We first note that

$$\boldsymbol{\Sigma}_{t|t} - \hat{\boldsymbol{\Sigma}}_{t|t} = \left( \boldsymbol{\Sigma} - \tilde{\boldsymbol{\Sigma}}_{t|t} \right) + \left( \tilde{\boldsymbol{\Sigma}}_{t|t} - \hat{\boldsymbol{\Sigma}}_{t|t} \right), \tag{105}$$

where $\tilde{\boldsymbol{\Sigma}}_{t|t}$ is the surrogate covariance matrix (99). Then,

$$\|\boldsymbol{\Sigma}_{t|t} - \hat{\boldsymbol{\Sigma}}_{t|t}\|_{\mathrm{F}} \leq \| \underbrace{\boldsymbol{\Sigma} - \tilde{\boldsymbol{\Sigma}}_{t|t}}_{=\mathbf{E}_{\mathrm{surr}}} \|_{\mathrm{F}} + \| \underbrace{\tilde{\boldsymbol{\Sigma}}_{t|t} - \hat{\boldsymbol{\Sigma}}_{t|t}}_{=\mathbf{E}_{\mathrm{proj}}} \|_{\mathrm{F}} \tag{106}$$

The norm for $\mathbf{E}_{\mathrm{proj}}$ follows directly from the definition of the low-rank approximation. Let $\mathbf{U}\,\mathbf{S}\,\mathbf{V}^\intercal$ be the SVD decomposition of $\tilde{\boldsymbol{\Sigma}}_{t|t}$ and let $\mathbf{U}\,\mathbf{S}_{:d}\,\mathbf{V}$ be the SVD decomposition of $\hat{\boldsymbol{\Sigma}}_{t|t}$. Here, $\mathbf{S} = \mathrm{diag}(\lambda_1, \ldots, \lambda_{D_{\boldsymbol{\theta}}})$ and $\mathbf{S}_{:d} = \mathrm{diag}(\lambda_1, \ldots, \lambda_d, 0, \ldots, 0)$ are the singular values (eigenvalues) of the matrices $\tilde{\boldsymbol{\Sigma}}_{t|t}$ and $\hat{\boldsymbol{\Sigma}}_{t|t}$ respectively, ordered in descending order. Then,

$$\begin{aligned}
\|\mathbf{E}_{\mathrm{proj}}\|_{\mathrm{F}} &= \|\mathbf{U}\,\mathbf{S}\,\mathbf{V}^\intercal - \mathbf{U}\,\mathbf{S}_{:d}\,\mathbf{V}^\intercal\|_{\mathrm{F}} \\
&= \|\mathbf{U}\,(\mathbf{S} - \mathbf{S}_{:d})\,\mathbf{V}^\intercal\|_{\mathrm{F}} \\
&= \sqrt{\mathrm{Tr}[(\mathbf{U}\,(\mathbf{S} - \mathbf{S}_{:d})\,\mathbf{V}^\intercal)\,(\mathbf{U}\,(\mathbf{S} - \mathbf{S}_{:d})\,\mathbf{V}^\intercal)^\intercal]} \\
&= \sqrt{\mathrm{Tr}[\mathbf{U}\,(\mathbf{S} - \mathbf{S}_{:d})\,\mathbf{V}^\intercal\mathbf{V}\,(\mathbf{S} - \mathbf{S}_{:d})\,\mathbf{U}^\intercal]} \\
&= \sqrt{\mathrm{Tr}[(\mathbf{S} - \mathbf{S}_{:d})\,\mathbf{V}^\intercal\mathbf{V}\,(\mathbf{S} - \mathbf{S}_{:d})\,\mathbf{U}^\intercal\mathbf{U}]} \\
&= \sqrt{\mathrm{Tr}[(\mathbf{S} - \mathbf{S}_{:d})^2]} \\
&= \sqrt{\sum_{k=d+1}^{D_{\boldsymbol{\theta}}} \lambda_k^2},
\end{aligned} \tag{107}$$

where $\{\lambda_k\}_{k=d+1}^{D_{\boldsymbol{\theta}}}$ are the bottom $(D_{\boldsymbol{\theta}} - d)$ eigenvalues of $\tilde{\boldsymbol{\Sigma}}_t$.

Next, an upper bound for the norm $\mathbf{E}_{\mathrm{surr}}$ is as follows

$$\begin{aligned}
\|\mathbf{E}_{\mathrm{surr}}\|_{\mathrm{F}} &= \|\boldsymbol{\Sigma}_{t|t} - \tilde{\boldsymbol{\Sigma}}_{t|t}\|_{\mathrm{F}} \\
&= \|q_t\,(\mathbf{I} - \mathbf{K}_t\,\mathbf{H}_t)\,(\mathbf{I} - \mathbf{K}_t\,\mathbf{H}_t) - q_t\,\mathbf{I}\|_{\mathrm{F}} \\
&= q_t\,\|(\mathbf{K}_t\,\mathbf{H}_t)\,(\mathbf{K}_t\,\mathbf{H}_t)^\intercal - (\mathbf{K}_t\,\mathbf{H}_t) - (\mathbf{K}_t\,\mathbf{H}_t)^\intercal\|_{\mathrm{F}} \\
&\leq q_t\left( \|(\mathbf{K}_t\,\mathbf{H}_t)\|_{\mathrm{F}}^2 + 2\,\|\mathbf{K}_t\,\mathbf{H}_t\|_{\mathrm{F}} \right).
\end{aligned} \tag{108}$$

$\square$

---

**Algorithm 4** Predict and update steps in the low-rank Kalman filter (LRKF)

---

**Require:** $\mathbf{R}_t$ // `measurement variance`
**Require:** $q_t$ // `dynamics covariance`
**Require:** $(\boldsymbol{y}_t, \boldsymbol{x}_t)$ // `observation and input`
**Require:** $\boldsymbol{b}_{t-1} = (\boldsymbol{\mu}_{t-1}, \mathbf{C}_{t-1})$ // `previous belief`
    // predict step
1: $\hat{\boldsymbol{y}}_t \leftarrow h(\boldsymbol{\theta}_t, \boldsymbol{x}_t)$
2: $\mathbf{H}_t \leftarrow \nabla_{\boldsymbol{\theta}} h(\boldsymbol{\mu}_{t-1}, \boldsymbol{x}_t)$
    // innovation and (Cholesky) innovation variance
3: $\boldsymbol{\epsilon}_t \leftarrow \boldsymbol{y}_t - \hat{\boldsymbol{y}}_t$
4: $\mathbf{S}_t^{1/2} \leftarrow \mathcal{Q}_R(\mathbf{C}_t \mathbf{H}_t^{\mathsf{T}}, \sqrt{q_t}\,\mathbf{H}_t, \mathbf{R}_t^{1/2})$
    // gain matrix
5: $\mathbf{V}_t \leftarrow \mathbf{S}_t^{-1/2}\,\mathbf{S}_t^{-\mathsf{T}/2}\,\mathbf{H}_t$
6: $\mathbf{K}_t^{\mathsf{T}} \leftarrow \mathbf{V}_t\,\mathbf{C}_{t-1}\,\mathbf{C}_{t-1}^{\mathsf{T}} + q_t\,\mathbf{V}_t$ // `low-rank update`
    // mean and low-rank factor updates
7: $\boldsymbol{\mu}_t \leftarrow \boldsymbol{\mu}_{t-1} + \mathbf{K}_t\,\boldsymbol{\epsilon}_t$
8: $\mathbf{C}_t \leftarrow \mathcal{P}_d\left(\mathbf{C}_{t-1}\,(\mathbf{I} - \mathbf{K}_t\,\mathbf{H}_t)^{\mathsf{T}}, \mathbf{R}_t^{1/2}\mathbf{K}_t^{\mathsf{T}}\right)$
9: **Return** $\boldsymbol{b}_t = (\boldsymbol{\mu}_t, \mathbf{C}_t)$ // `updated belief`

---

### F.2.1 Error analysis for LRKF

Below, we analyze the single-step error incurred by LRKF without dynamics in the model parameters and scalar linear model, i.e., $\boldsymbol{\theta}_t = \boldsymbol{\theta}_{t-1} = \ldots = \boldsymbol{\theta}$ and $y_t = \boldsymbol{\theta}^{\mathsf{T}}\boldsymbol{x}_t + e_t$ with $\mathrm{Var}[e_t] = r^2$.

**Proposition F.4.** *Consider the linear model*

$$y_t = \boldsymbol{\theta}^{\mathsf{T}}\boldsymbol{x}_t + e_t,$$

*where $e_t$ is zero-mean with $\mathrm{Var}[e_t] = r^2$, and let $\boldsymbol{\Sigma}_{t-1} = \mathrm{Var}(\boldsymbol{\theta} - \boldsymbol{\theta}_{t-1|t-1})$. Let $\widehat{\boldsymbol{\Sigma}}_{t-1}$ denote the rank-$d$ covariance used by LRKF, and define*

$$\gamma_t = \min\{\boldsymbol{x}_t^{\mathsf{T}}\boldsymbol{\Sigma}_{t-1}\boldsymbol{x}_t,\ \boldsymbol{x}_t^{\mathsf{T}}\widehat{\boldsymbol{\Sigma}}_{t-1}\boldsymbol{x}_t\} + r^2 \quad (>0), \qquad \epsilon_t = y_t - \boldsymbol{\theta}_{t-1|t-1}^{\mathsf{T}}\boldsymbol{x}_t.$$

*Then the one-step difference between the BLUP (full-rank update) and the rank-$d$ LRKF update satisfies*

$$\|\boldsymbol{\theta}_{t|t} - \hat{\boldsymbol{\theta}}_{t|t}\|_2 \leq |\epsilon_t|\,\sigma_{d+1}(\boldsymbol{\Sigma}_{t-1})\,\|\boldsymbol{x}_t\|_2 \left(\frac{1}{\gamma_t} + \frac{\sigma_1(\boldsymbol{\Sigma}_{t-1})\,\|\boldsymbol{x}_t\|_2^2}{\gamma_t^2}\right). \tag{109}$$

*Here $\sigma_1(\boldsymbol{\Sigma}_{t-1}) \geq \cdots \geq \sigma_D(\boldsymbol{\Sigma}_{t-1})$ are the singular values of $\boldsymbol{\Sigma}_{t-1}$, $\boldsymbol{\theta}_{t|t}$ is the BLUP, and $\hat{\boldsymbol{\theta}}_{t|t}$ is the rank-$d$ LRKF estimate.*

In this setting, the surrogate covariance introduced by LRKF is unnecessary, and $\hat{\boldsymbol{\Sigma}}_t = \mathbf{C}_t^{\mathsf{T}}\mathbf{C}_t$ reduces to the best rank-$d$ approximation of the EVC matrix $\boldsymbol{\Sigma}_{t-1} = \mathrm{Var}(\boldsymbol{\theta} - \boldsymbol{\theta}_{t-1|t-1})$. As Proposition F.4 shows, a single step of LRKF deviates from the dense (i.e., full-KF) update by a factor governed entirely by the rank truncation. At time $t$, the residual $\epsilon_t$, the maximum eigenvalue $\sigma_1(\boldsymbol{\Sigma}_{t-1})$, and the feature $\boldsymbol{x}_t$ are all fixed. Consequently, the only way to tighten the bound and reduce the gap to the full update is to increase the chosen rank $d$.

*Proof.* We first note that

$$\boldsymbol{\theta}_{t|t} = \boldsymbol{\theta}_{t-1|t-1} + \mathbf{K}_t\,\varepsilon_t,$$
$$\hat{\boldsymbol{\theta}}_{t|t} = \boldsymbol{\theta}_{t-1|t-1} + \hat{\mathbf{K}}_t\,\varepsilon_t, \tag{110}$$

with

$$\mathbf{K}_t = \frac{\boldsymbol{\Sigma}_{t-1}\,\boldsymbol{x}_t}{S_t}, \quad S_t = \boldsymbol{x}_t^{\mathsf{T}}\,\boldsymbol{\Sigma}_{t-1}\,\boldsymbol{x}_t + r^2,$$

$$\hat{\mathbf{K}}_t = \frac{\hat{\boldsymbol{\Sigma}}_{t-1}\,\boldsymbol{x}_t}{\hat{S}_t}, \quad \hat{S}_t = \boldsymbol{x}_t^{\mathsf{T}}\,\hat{\boldsymbol{\Sigma}}_{t-1}\,\boldsymbol{x}_t + r^2. \tag{111}$$

Then

$$\mathbf{K}_t - \hat{\mathbf{K}}_t = \frac{\boldsymbol{\Sigma}_{t-1}\, \boldsymbol{x}_t}{S_t} - \frac{\hat{\boldsymbol{\Sigma}}_{t-1}\, \boldsymbol{x}_t}{\hat{S}_t}$$

$$= \frac{(\boldsymbol{\Sigma}_{t-1} - \hat{\boldsymbol{\Sigma}}_{t-1})\, \boldsymbol{x}_t}{S_t} + \hat{\boldsymbol{\Sigma}}_{t-1}\, \boldsymbol{x}_t \left( \frac{1}{S_t} - \frac{1}{\hat{S}_t} \right). \tag{112}$$

Thus,

$$\|\boldsymbol{\theta}_{t|t} - \hat{\boldsymbol{\theta}}_{t|t}\|_2 = \|(\mathbf{K}_t - \hat{\mathbf{K}}_t)\, \varepsilon_t\|_2 \le |\varepsilon_t| \left( \underbrace{\left\| \frac{(\boldsymbol{\Sigma}_{t-1} - \hat{\boldsymbol{\Sigma}}_{t-1})\, \boldsymbol{x}_t}{S_t} \right\|_2}_{\text{(I)}} + \underbrace{\left\| \hat{\boldsymbol{\Sigma}}_{t-1}\, \boldsymbol{x}_t \left( \frac{1}{S_t} - \frac{1}{\hat{S}_t} \right) \right\|_2}_{\text{(II)}} \right).$$

$$\tag{113}$$

**Bound for** (I).

$$\left\| \frac{(\boldsymbol{\Sigma}_{t-1} - \hat{\boldsymbol{\Sigma}}_{t-1})\, \boldsymbol{x}_t}{S_t} \right\|_2 \le \frac{\|\boldsymbol{\Sigma}_{t-1} - \hat{\boldsymbol{\Sigma}}_{t-1}\|_2\, \|\boldsymbol{x}_t\|_2}{|S_t|} \le \frac{\sigma_{d+1}(\boldsymbol{\Sigma}_{t-1})\, \|\boldsymbol{x}_t\|_2}{\gamma_t},$$

where we used Eckart–Young–Mirsky [17] and $|S_t| \ge \gamma_t := \min\{\boldsymbol{x}_t^\top \boldsymbol{\Sigma}_{t-1} \boldsymbol{x}_t,\ \boldsymbol{x}_t^\top \hat{\boldsymbol{\Sigma}}_{t-1} \boldsymbol{x}_t\} + r^2 > 0$.

**Bound for** (II). Since

$$\left| \frac{1}{S_t} - \frac{1}{\hat{S}_t} \right| = \frac{|\hat{S}_t - S_t|}{|S_t\, \hat{S}_t|} = \frac{|\boldsymbol{x}_t^\top (\hat{\boldsymbol{\Sigma}}_{t-1} - \boldsymbol{\Sigma}_{t-1}) \boldsymbol{x}_t|}{|S_t\, \hat{S}_t|} \le \frac{\|\boldsymbol{x}_t\|_2^2\, \|\hat{\boldsymbol{\Sigma}}_{t-1} - \boldsymbol{\Sigma}_{t-1}\|_2}{\gamma_t^2},$$

we obtain

$$\text{(II)} \le \|\hat{\boldsymbol{\Sigma}}_{t-1}\|_2\, \|\boldsymbol{x}_t\|_2\, \frac{\|\boldsymbol{x}_t\|_2^2\, \|\hat{\boldsymbol{\Sigma}}_{t-1} - \boldsymbol{\Sigma}_{t-1}\|_2}{\gamma_t^2} = \frac{\|\hat{\boldsymbol{\Sigma}}_{t-1}\|_2\, \|\boldsymbol{x}_t\|_2^3\, \sigma_{d+1}(\boldsymbol{\Sigma}_{t-1})}{\gamma_t^2}.$$

Since $\|\hat{\boldsymbol{\Sigma}}_{t-1}\|_2 = \sigma_1(\boldsymbol{\Sigma}_{t-1})$, this becomes

$$\text{(II)} \le \frac{\sigma_1(\boldsymbol{\Sigma}_{t-1})\, \|\boldsymbol{x}_t\|_2^3\, \sigma_{d+1}(\boldsymbol{\Sigma}_{t-1})}{\gamma_t^2}.$$

Plugging the bounds for (I) and (II) into (113) yields

$$\|\boldsymbol{\theta}_{t|t} - \hat{\boldsymbol{\theta}}_{t|t}\|_2 \le |\varepsilon_t|\, \sigma_{d+1}(\boldsymbol{\Sigma}_{t-1})\, \|\boldsymbol{x}_t\|_2 \left( \frac{1}{\gamma_t} + \frac{\sigma_1(\boldsymbol{\Sigma}_{t-1})\, \|\boldsymbol{x}_t\|_2^2}{\gamma_t^2} \right),$$

as claimed. $\square$

### F.3 Replay-buffer variational Bayesian last layer (`OnVBLL`)

The `VBLL` method has explicit posterior predictive

$$p(\boldsymbol{y} \mid \boldsymbol{\ell}, \boldsymbol{h}, \boldsymbol{\Sigma}, \mathbf{R}) = \mathcal{N}\left(\boldsymbol{y} \mid \boldsymbol{\ell}^\mathsf{T}\, \psi(\boldsymbol{h}, \boldsymbol{x}+),\ \boldsymbol{\ell}^\mathsf{T}\, \psi(\boldsymbol{h}, \boldsymbol{x})\, \boldsymbol{\Sigma}\, \psi(\boldsymbol{h}, \boldsymbol{x})^\mathsf{T}\, \boldsymbol{\ell} + \mathbf{R}\right).$$

Building on [25, 5], we observe that for any given $q(\bar{\boldsymbol{\ell}}) = \mathcal{N}(\bar{\boldsymbol{\ell}} \mid \boldsymbol{\ell}, \boldsymbol{\Sigma})$, a lower-bound for the marginal log-likelihood of a single datapoint is given by

$$\log p(\boldsymbol{y}_t \mid \boldsymbol{x}_t, \boldsymbol{h}, \mathbf{R}) \ge \mathbb{E}_{q(\bar{\boldsymbol{\ell}})}[\log p(\boldsymbol{y}_t \mid \boldsymbol{x}_t, \bar{\boldsymbol{\ell}}, \boldsymbol{h}, \mathbf{R})]$$

$$= \log \mathcal{N}\left(\boldsymbol{y}_t \mid \boldsymbol{\ell}^\mathsf{T} \phi(\boldsymbol{h}, \boldsymbol{x}_t), \mathbf{R}\right) - \frac{1}{2} \phi(\boldsymbol{h}, \boldsymbol{x}_t) \boldsymbol{\Sigma}\, \phi(\boldsymbol{h}, \boldsymbol{x}_t)\, \mathrm{Tr}\left(\mathbf{R}^{-1}\right) \tag{114}$$

$$=: -\mathcal{L}_t(\boldsymbol{\ell}, \boldsymbol{h}, \mathbf{R}, \boldsymbol{\Sigma}).$$

Following [33, 32, 30] a *generalized* posterior with loss function $\mathcal{L}$ is given by

$$q_t(\boldsymbol{\ell}, \boldsymbol{h}, \mathbf{R}, \boldsymbol{\Sigma}) \propto q_{t-1}(\boldsymbol{\ell}, \boldsymbol{h}, \mathbf{R}, \boldsymbol{\Sigma})\, \exp(-\mathcal{L}_t(\boldsymbol{\ell}, \boldsymbol{h}, \mathbf{R}, \boldsymbol{\Sigma})). \tag{115}$$

Estimation of the posterior mean (MAP filtering) involves estimating

$$\boldsymbol{\ell}_t, \boldsymbol{h}_t, \mathbf{R}_t, \boldsymbol{\Sigma}_t = \underset{\boldsymbol{\ell}, \boldsymbol{h}, \mathbf{R}, \boldsymbol{\Sigma}}{\arg\max} \, q_t(\boldsymbol{\ell}, \boldsymbol{h}, \mathbf{R}, \boldsymbol{\Sigma}) \tag{116}$$

which can be done implicitly through adaptive optimization methods as shown in [2]. Next, to improve the performance, we consider a replay-buffer which has been shown to be much more efficient than doing fully-online SGD [1, 9, 40]. This assumption breaks with the fully-online assumption, but it is a good contender in sequential decision making problems in stationary environments.

## G  Additional experiments

### G.1  Online classification and sequential decision making on the MNIST dataset

The results in Section 5.2 consider the LeNet5 convolutional neural network (CNN) architecture [39]. In this architecture, the last layer is 80-dimensional and the output layer is 10-dimensional, which corresponds to each of the 10 possible classes. Applying `HiLoFi` in this setting would need the storage and update of an $800 \times 800$ covariance matrix and 800-dimensional mean vector. Sequential update of the Cholesky representation of this covariance matrix, although feasible, takes most of the computational cost in a single step of `HiLoFi`. Given this, here we consider `LoLoFi` to offset the computational cost of the layer layer. Another reason to prefer `LoLoFi` over `HiLoFi` in this setting is that the rank of the last layer is 800-dimensional, whereas the rank for the hidden layers is 50 dimensional. This corresponds to an overparametrized linear regression, which has been shown to have some pathologies in the offline setting [61].

As a simpler baseline, which does not model uncertainty, we consider **Muon** (`muon`) [31], which a special case of the Shampoo optimizer [24]. This is a quasi second-order optimization method that is scalable and shows strong empirical performance. We use the implementation in the Optax library [12]. Unlike the other methods, it only computes a point estimate, so we cannot use it for computing the posterior predictive. We use this method for the $\epsilon$-greedy bandit in Appendix G.1.3 and as an additional choice of optimizer in Section G.1.1.

### G.1.1  Online classification on the MNIST dataset

In this experiment, we consider the problem online learning and classification on the MNIST dataset using the LeNet5 CNN. We use the optimizers detailed in Section 5.2. We take the data $\mathcal{D}_{1:T}$ with $T = 60,000$ and $\mathcal{D}_t = (\boldsymbol{x}_t, \boldsymbol{y}_t)$, where $\boldsymbol{x}_t$ is a $(28 \times 28 \times 1)$ array and $\boldsymbol{y}_t \in \{0,1\}^{10}$ a one-hot-encoded vector such that $(\boldsymbol{y}_t)_i = 1$ if $\boldsymbol{x}_t$ represents the digit $i$ and 0 otherwise. At every timestep $t = 1, \ldots, T$ each agent is presented the image $\boldsymbol{x}_t$ which it has to classify. A predicted classification is made through the prediction $\boldsymbol{y}_{t|t-1} = f(\boldsymbol{\mu}_{t|t-1}, \boldsymbol{x}_t)$ with $\boldsymbol{\mu}_{t|t-1} = \mathbb{E}[\boldsymbol{\theta}_t \,|\, \mathcal{D}_{1:t-1}]$ and then updates its beliefs given the (true) reward $\boldsymbol{y}_t$.

Having the BLUP over parameters $\boldsymbol{\theta}_{t|t-1} = \mathbb{E}[\boldsymbol{\theta}_t \,|\, \mathcal{D}_{1:t-1}]$, the linearized model has mean and variance

$$\begin{aligned}
\boldsymbol{m}(\boldsymbol{\theta}_t, \boldsymbol{x}_t) &= \mathrm{softmax}(f(\boldsymbol{\theta}_t, \boldsymbol{x})), \\
\bar{\boldsymbol{m}}_t &= \boldsymbol{m}(\boldsymbol{\theta}_{t|t-1}, \boldsymbol{x}_t) + \nabla_{\boldsymbol{\theta}} \boldsymbol{m}(\boldsymbol{\theta}_{t|t-1}, \boldsymbol{x})(\boldsymbol{\theta} - \boldsymbol{m}_{t-1}), \\
\bar{\boldsymbol{s}}_t &= \mathrm{diag}(\boldsymbol{m}(\boldsymbol{\theta}_{t|t-1}, \boldsymbol{x}_t)) + \boldsymbol{m}(\boldsymbol{\theta}_{t|t-1}, \boldsymbol{x}_t)\, \boldsymbol{m}(\boldsymbol{\theta}_{t|t-1}, \boldsymbol{x}_t)^\mathsf{T} + \varepsilon \, \mathbf{I}_{D_{\boldsymbol{y}}}.
\end{aligned} \tag{117}$$

This moment-matched linearization corresponds, in the Bayesian setting, to a Gaussian likelihood whose first and second moments match that of a Multinomial distribution. This representation was introduced in the context of online learning in [48].

In (117), $\mathrm{diag}(\boldsymbol{v})$ is a function that takes as input the vector $\boldsymbol{v} \in \mathbb{R}^m$ and outputs a diagonal matrix with entries $\mathrm{diag}(\boldsymbol{v})_{i,j} = \boldsymbol{v}_i \, \delta(i - j)$, and the term $\varepsilon > 0$ is a small constant that ensures non-zero variance.

**Online learning results.**  Figure 5 shows the 5000-step rolling mean of the one-step-ahead classification outcome (one for correct, zero for incorrect) using the various learning algorithms. We observe that there is no clear difference between the methods. This shows that one can tackle this complete information problem without needing to model uncertainty. This is not the case in the incomplete information case that we study in any of the experiments presented in Section 5.

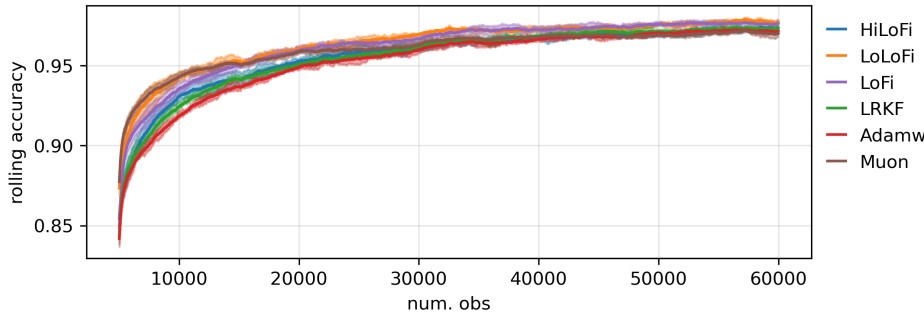

Figure 5: Rolling one-step-ahead accuracy for the MNIST dataset.

Figure 6 shows the one-step-ahead accuracy of the last $10,000$ images and the running time of processing all $60,000$ images for `HiLoFi`, `LoLoFi`, `LRKF`, and `LoFi` as a function of their rank. For `LoLoFi`, we fix the rank of the last layer to be $50$ and vary the rank of the hidden layers. We observe

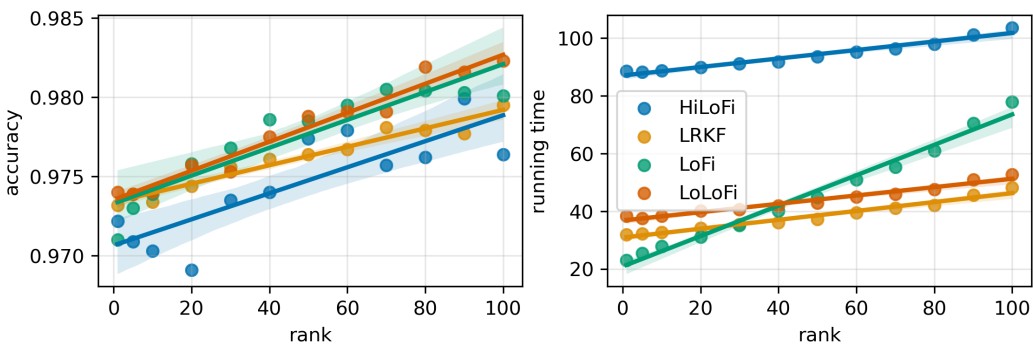

Figure 6: Results of MNIST for online classification. *Left panel*: accuracy as a function of rank. *Right panel:* running time as a function of rank.

that for low-ranks up to dimension 20, `LoFi` is faster than both `LoLoFi` and `LRKF`. However, the running time of `LoFi` increases much more rapidly (as a function of rank) than either `LoLoFi` and `LRKF`. This is a consequence of the computational costs associated with points (1,2,3) in Appendix D.2. Next, we observe that the running time of `LoLoFi` is sightly above `LRKF` for varying rank; this is because of the double approximation of the covariance matrix: one for the hidden layers and another one for the last layer. Finally, we observe that `HiLoFi` is the method with highest computational cost in this experiment. This is due the high-dimensionality of the last-layer, relative to the rest of the methods.

Lastly, we observe that all methods increase their performance as a function of rank, albeit marginally. The method that most benefit from an increase in performance in `HiLoFi`. We hypothesize that this is due to the total rank of the hidden layer, relative to the rank of the last layer, which in this experiment is $800$.

### G.1.2 Error analysis for `LRKF`

The top panel of Figure 7 shows the rolling mean for the two sources of approximation error in the covariance matrix for the `LRKF` method (detailed in Proposition F.3), as well as the rolling one-step-ahead accuracy. We observe that, on average, both sources of error decrease over time and the accuacy of `LRKF` improves over time. Here, the only sources of error are the surrogate error and the low-rank error. This is because `LRKF` does not distinguish between last-layer and hidden-layer parameters.

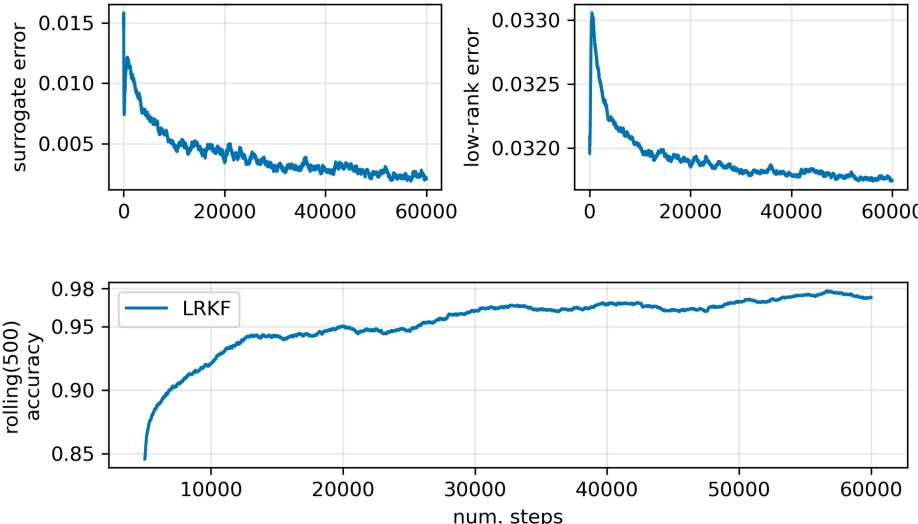

Figure 7: Top: upper bound error for the surrogate error and the low-rank approximation. Bottom: $5,000$-step rolling one-step-ahead accuracy for the MNIST dataset.

### G.1.3 MNIST as multi-armed bandit

Here, we present further results from Section 5.2 More precisely, we study the multi-armed bandit approach to solving the MNIST classification problem.

**Choice of hyperparameters.** The hyperparameters for this experiment were chosen as follows: For Adamw and muon (used in $\varepsilon$-greedy and in conjunction with LLL), we use a learning rate of $10^{-4}$, and $\varepsilon = 0.05$ For Adamw, we take 5 inner iterations and a buffer size of one and for muon, we take 1 inner iteration and a buffer size of one. Next, HiLoFi considers a rank of 50 for the hidden parameters, we take $q_{h,t} = 10^{-6}$ and $q_\ell = 10^{-6}$. The initial covariances $\Sigma_{h,0}$ and $\Sigma_{\ell,0}$ are both initialized as identity times a factor of $10^{-1}$. Similarly, LoLoFi is initialized as with HiLoFi, but with rank in the last-layer to be 100. For LRKF, we consider a rank of 50, $\Sigma_0$ is initialized as the identity matrix, and $q_t = 10^{-6}$. Finally, for LoFi, we also considered rank 50; then, following the experiments in [8], we considered the initial covariance to be $a\mathbf{I}$, with $a = \exp(-8)$ (this corresponds to low-rank comprised of a matrix of zeros and diagonal terms, all set to $a$); lastly, the dynamics covariance is $q_t = 0$.

All initial model parameters from the neural network are shared across methods and trials.

**Regret analysis.** Below, we present results considering regret. Here, the reward is either 1, if the classification is done correctly and 0 otherwise. Let $y_t \in \{0, 1\}$ be the reward obtained at time $t$, and $\boldsymbol{y}_{t,a}$ be the value of arm $a$. Then the regret obtain at time $t$ is given by

$$\sum_{\tau=1}^{t} (\max_a \boldsymbol{y}_{\tau,a} - y_\tau) = t - \sum_{\tau=1}^{t} y_\tau.$$

Figure 8 shows the average regret (across 10 trials) for HiLoFi, LoLoFi, LRKF, LoFi, and LLL. Here, we consider $\epsilon$-greedy variants for HiLoFi, LoLoFi, and LRKF. Next, TS with LLL and Adamw optimizer is denoted adamw-TS, TS with LLL and muon optimizer is denoted muon-TS. $\epsilon$-greedy with Adamw optimizer is the denoted adamw-eps, and $\epsilon$-greedy with muon optimizer is denoted muon-eps.

**Time-regret analysis.** Figure 9 shows the total running time to run the ten trials in the $x$-axis, the regret around one-standard deviation in the $y$-axis. We observe Adamw and muon are the methods with lowest performance and lowest relative cumulative reward for either $\epsilon$-greedy or TS using LLL. Next, we observe that LRKF is faster than LoLoFi, and LoLoFi is faster than HiLoFi. This result

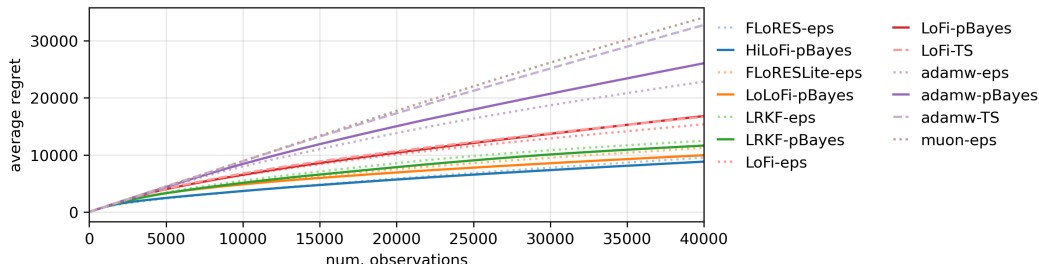

Figure 8: Total running time for experiments ($x$-axis) versus average regret across ten trials ($y$-axis).

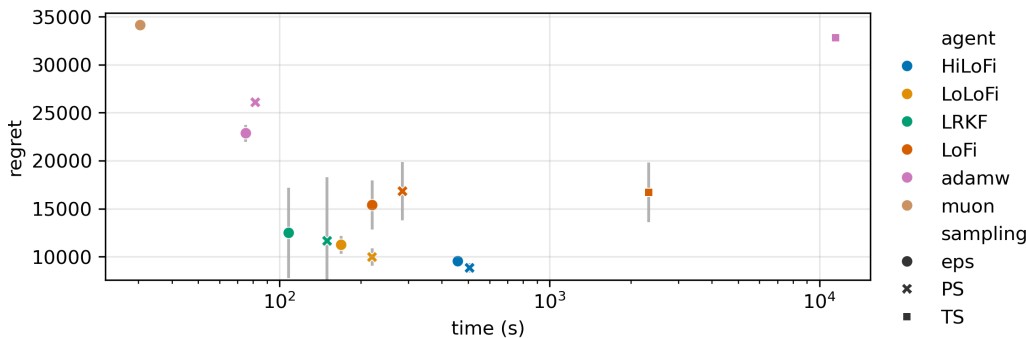

Figure 9: Total running time for experiments ($x$-axis) versus mean relative cumulative reward ($y$-axis).

is expected and is explained in Appendix D.2. We also observe that `LoFi` with $\epsilon$-greedy matches `LoLoFi` with PS in time; however, `LoLoFi` has lower regret. We conjecture that this result is due to the explicit modeling of the last layer done for `LoLoFi`.

Finally, we observe that for all methods, $\epsilon$-greedy is faster than TS. This is because $\epsilon$-greedy only requires the evaluation of the posterior predictive mean, whereas TS samples from the posterior predictive, which in turn requires building the posterior predictive covariance shown in (10).

### G.2 Online classification using `LRKF` on the CIFAR-10 dataset

In this Section, we study the ability of `LRKF` to scale to million-dimensional parameters for online classification on the CIFAR-10 dataset. We consider two VGG-style convolutional networks of different capacity. Both follow the pattern of stacked $3 \times 3$ convolutions, ELU activations. We summarize the architectures below We follow a similar setup as that of Appendix G.1.1. We take

| Model | Conv Blocks | Dense Layers | Params |
|---|---|---|---|
| VGG | $64 \times 2 \to 128 \times 2 \to 256$ | $256 \to 256$ | 1.7M |
| VGG+Block | $64 \times 2 \to 128 \times 2 \to 256 \to 512 \times 2$ | $256 \to 256$ | 4.7M |

Table 3: Summary of the two VGG-style architectures used in our experiments. "$C \times n$" denotes $n$ convolutional layers with $C$ channels.

the data $\mathcal{D}_{1:T}$ with $T = 10,000$ and $\mathcal{D}_t = (\boldsymbol{x}_t, \boldsymbol{y}_t)$, where $\boldsymbol{x}$ is a $(32 \times 32 \times 3)$-dimensional array and $\boldsymbol{y}_t \in \{0,1\}^{10}$ a one-hot encoded vector such that $\boldsymbol{y}_t$. At every timestep $t = 1, \ldots, T$ each agent is presented the image $\boldsymbol{x}_t$ which it has to classify. A predicted classification is made through the prediction $\boldsymbol{y}_{t|t-1} = f(\boldsymbol{\mu}_{t|t-1}, \boldsymbol{x}_t)$ with $\boldsymbol{\mu}_{t|t-1} = \mathbb{E}[\boldsymbol{\theta}_t \,|\, \mathcal{D}_{1:t-1}]$ and then updates its beliefs given the (true) reward $\boldsymbol{y}_t$. Updates are done as outlined in Appendix G.1.1.

Table 4 shows the one-step-ahead accuracy for LRKF for ranks 1, 5, and 10. These results show

| Model | LRKF-**1** | LRKF-**5** | LRKF-**10** |
|---|---|---|---|
| VGG (1.7M) | 0.3355 | 0.3384 | 0.3379 |
| VGG+Block (4.7M) | 0.3129 | 0.3141 | 0.3148 |
| **AdamW (baseline)** | VGG: 0.3442 | | VGG+Block: 0.3097 |

Table 4: Online classification results on CIFAR-10 using LRKF with varying ranks (1, 5, 10), compared against AdamW baselines. Lower values are better.

that LRKF scales reliably to multi-million parameter networks while maintaining stable performance across different low-rank configurations. This highlights the broader applicability of our approach (including HiLoFi and LoLoFi) to large-scale architectures in vision and related domains where sequential decision making is needed.

### G.3 Bandits as recommendation systems

Further results from Section 5.3

**Choice of hyperparameters.** In this experiment, all agents shared a *rank* of 20. The choice of hyperparameters for LLL, LoFi, VBLL, and OnVBLL were chosen following analysis of performance on the first $10,000$ observations of the datasets. In particular, we tried Bayesian optimization techniques, but found that manual tuning of hyperparameters on the first $10,000$ observations yielded higher overall performance.

For OnVBLL, we considered a buffer size of 10, a regularization weight of 1.0, and 50 inner iterations. For VBLL, we considered a buffer size of 1000, a regularization weight of $10^{-3}$, and 100 inner iterations.

Next, for HiLoFi, we take $q_{h,t} = q_{\ell,t} = 0.0$ and set the initial covariances to be the identity. Next, for LRKF, we take $q_t = 0$ and identity matrix as initial covariance.

**Further results.** Figure 10 shows the daily cumulative reward for all methods. To produce this plot, we sum the cumulative rewards for all users at the end of each day and then perform a cumulative sum over the end-of-day reward. We observe that HiLoFi obtains the highest daily cumulative reward.

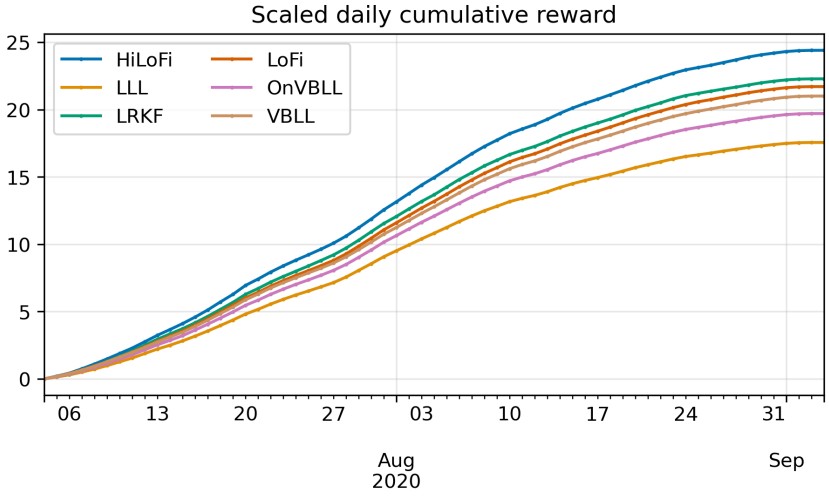

Figure 10: Daily cumulative reward.

The method with second highest daily cumulative reward is LRKF, closely followed by LoFi and VBLL. However, as shown in Figure 3. LRKF is more than an order of magnitude faster than VBLL.

From Figure 10 and Figure 10, we observe that the top performing methods in this experiment are `HiLoFi` and `LRKF`. However, `LRKF` is slightly faster than `HiLoFi` relative to all other methods.

## G.4   Bayesian optimization

Further results from Section 5.4

**Choice of hyperparameters.**   We fix hyperparameters across methods to ensure fairness and reproducibility. For rank-based methods, we use a rank of 50. Methods requiring buffers (e.g., LLL, `OnVBLL`) use a FIFO replay buffer of size 20, with 50 inner iterations per update. VBLL uses the full dataset and 100 iterations per step.

Where applicable, aleatoric uncertainty is discarded ($\mathbf{R}_t = 0$) to isolate epistemic effects. We consider equal learning rates across optimizer-based methods (e.g., LLL, `OnVBLL`) at $10^{-4}$. In filtering-based methods (e.g., `HiLoFi`, `LoFi`), this learning rate corresponds to the initial hidden-layer covariance; we set the final-layer prior variance to 1, emphasising epistemic modelling in the output layer. Hyperparameters for VBLL (Wishart scale, regularization weight) follow prior work [5].

**Per-step results.**   Figure 11 shows the median performance and the interquartile range of the best value found by each method for all test datasets. We observe that VBLL, `OnVBLL`, and `HiLoFi`

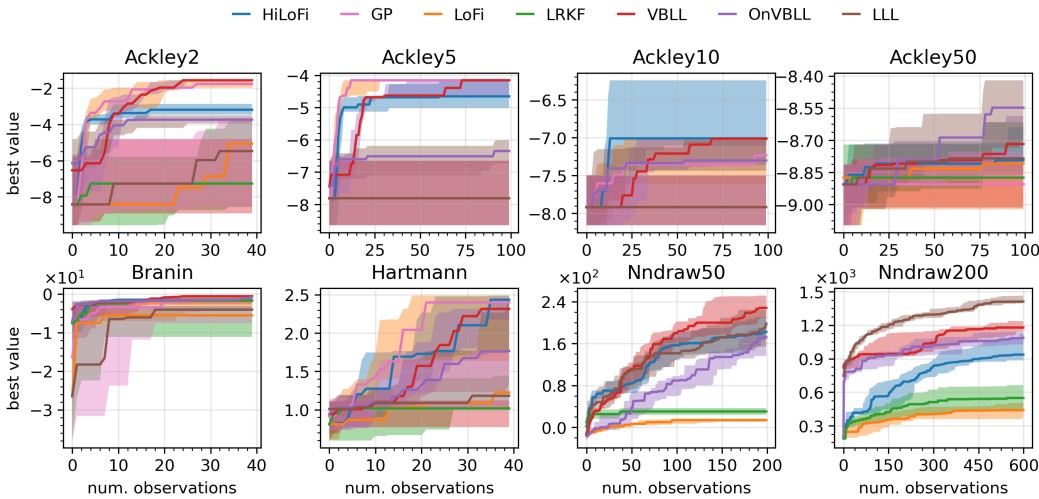

Figure 11:  Bayesian optimization benchmark on stationary environments.

are among the top performers for all datasets. Next, LLL is the least performant method for the low-dimensional datasets, but among the top performers for Nndraw50 and the top performer in Nndraw200.

Next, Figure 12 shows the median performance and the interquartile range of the best value found by each method for all test datasets using expected improvement.

### G.4.1   NN Draw

Here, we provide further results for the performance of `HiLoFi` and `LRKF` for the Bayesian optimization problem on the DrawNN dataset.

Figure 13 reports the best value found over the course of the optimization as a function of iteration, for different ranks. We observe that increasing the rank generally improves performance, with higher ranks yielding sharper improvements early in the search that compound into stronger final results. This comes with a trade-off between computational cost and the ability to capture uncertainty more effectively, as illustrated in Table 5, which reports the final performance and running time across ranks. We observe that both performance and running time generally increase with rank.

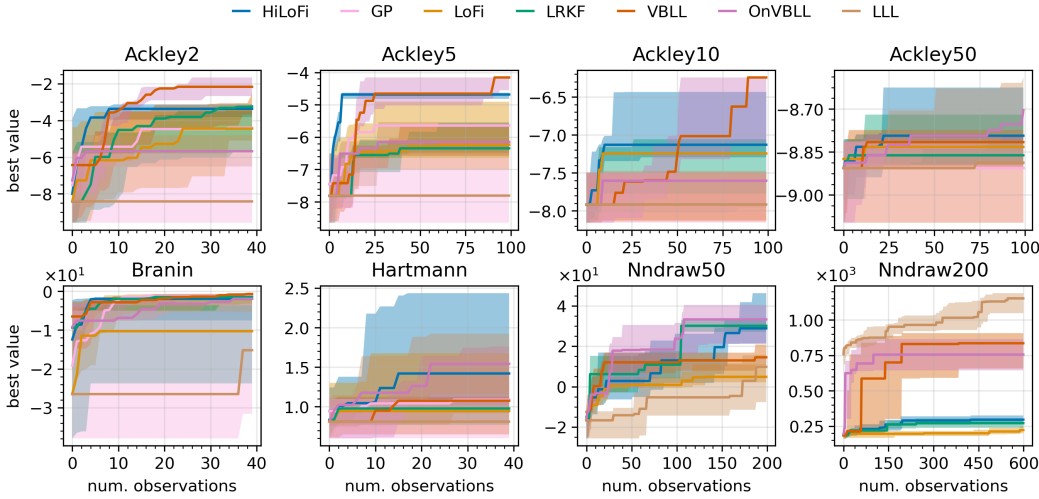

Figure 12: Bayesian optimization benchmark on stationary environments. Query points are chosen using expected improvement.

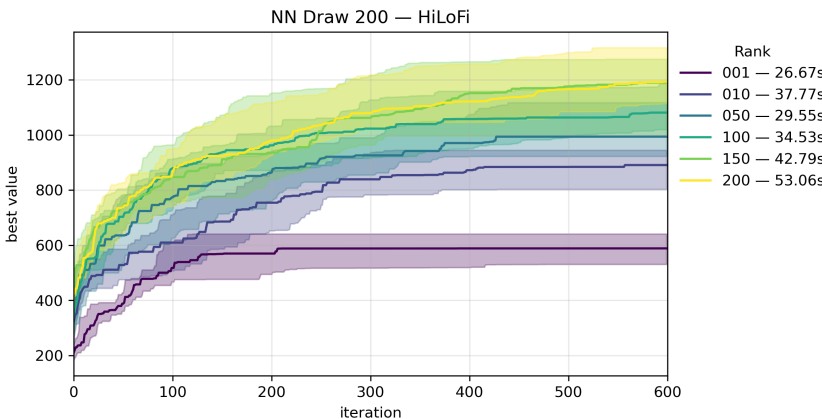

Figure 13: Solid lines show the median best value over 20 runs, and shaded regions denote the interquartile range.

Finally, Figure 14 shows the best value obtained during optimization as a function of the number of observations, for different ranks using LRKF. Compared to HiLoFi, LRKF requires substantially higher ranks to achieve similar median performance. For instance, performance is comparable at rank 300 for LRKF (73s) and rank 200 for HiLoFi (under 50s). This demonstrates that explicitly modeling the last layer in full rank, as done in HiLoFi, yields a more favorable accuracy–efficiency trade-off.

### G.5 In-between uncertainty

In this experiment, we test the ability of HiLoFi to capture the *in-between uncertainty* [21] after a single pass of the data. We consider the one-dimensional dataset introduced in [58] (Figure 1). For this experiment, we consider a four hidden-layer MLP with 128 units per layer and ELU activation function.

**Result for HiLoFi.** We take $q_{h,t} = q_{\ell,t} = 0$, initial covariances for last and hidden layers to be identity times $1/2$, and $\mathbf{R}_t = 0.0$, which corresponds to having no aleatoric uncertainty in the data generating process.

| Rank | Time (s) | Final $y_{\mathbf{best}}$ |
|---|---|---|
| 1 | 26.226 | 656.558 |
| 10 | 38.4492 | 630.556 |
| 20 | 28.8612 | 670.913 |
| 50 | 30.6409 | 684.009 |
| 100 | 34.5302 | 782.205 |
| 110 | 35.1504 | 774.416 |
| 120 | 39.2540 | 807.298 |
| 130 | 38.2288 | 846.218 |
| 140 | 39.1417 | 848.219 |
| 150 | 40.7939 | 864.806 |
| 160 | 42.1907 | 911.116 |
| 170 | 44.5485 | 900.566 |
| 180 | 45.5640 | 864.260 |
| 190 | 47.1553 | 911.262 |
| 200 | 48.2127 | 921.062 |

Table 5: Performance metrics across different low-rank configurations.

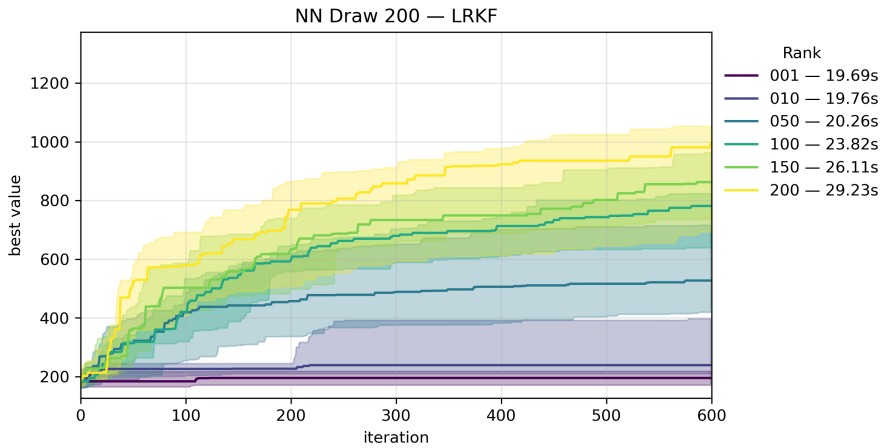

Figure 14: Bayesian optimization on the NNDraw dataset using LRKF. Solid lines show the median best value over 20 runs, and shaded regions denote the interquartile range.

Figure 15 shows the posterior predicted mean surrounded by the two-standard deviation posterior predictive for various ranks in the hidden layer.

We observe that a rank of 1 is not able to capture any uncertainty around the region with no observations; however, as we increase the rank, we observe that the posterior predictive becomes less and less confident about the *true* value of the mean on regions without data.

Next, Figure 16 shows the evolution of the posterior predictive mean and the posterior predictive variance as a function of the number of seen observations.

We observe that the uncertainty around the posterior predictive mean is wide (covering the limit from $-10$ to $10$) and starts to decrease after $10$ observations. By $30$ observations, the uncertainty is narrower over regions where data has been observed, and by $120$ observations, most of the posterior predictive uncertainty is on regions where no data has been observed.

**Result for LRKF** We repeat the experiment above for LRKF. Figure 17 shows the posterior predicted mean surrounded by the two-standard deviation posterior predictive for various ranks. In contrast to HiLoFi, we observe that LRKF requires a much higher rank to capture a reasonable level of *in-between* uncertainty. The panel for rank 10 is empty because the posterior predictive (with $\mathbf{R}_t = 0$) is psd, so it is not defined.

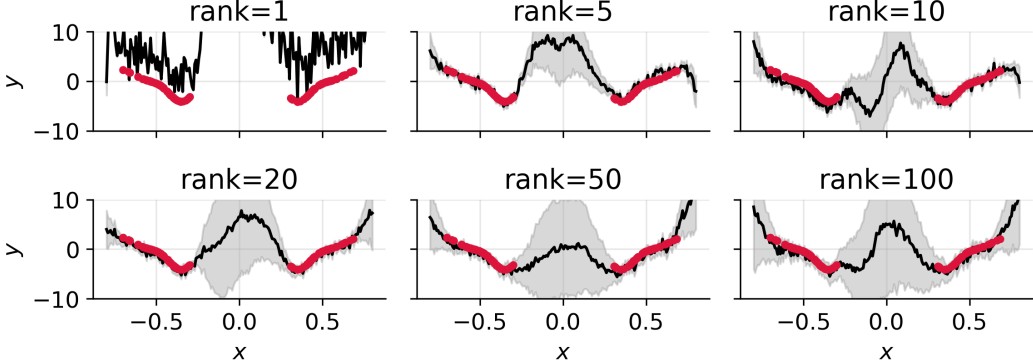

Figure 15: In-between uncertainty of the posterior predictive induced by `HiLoFi` as a function of rank.

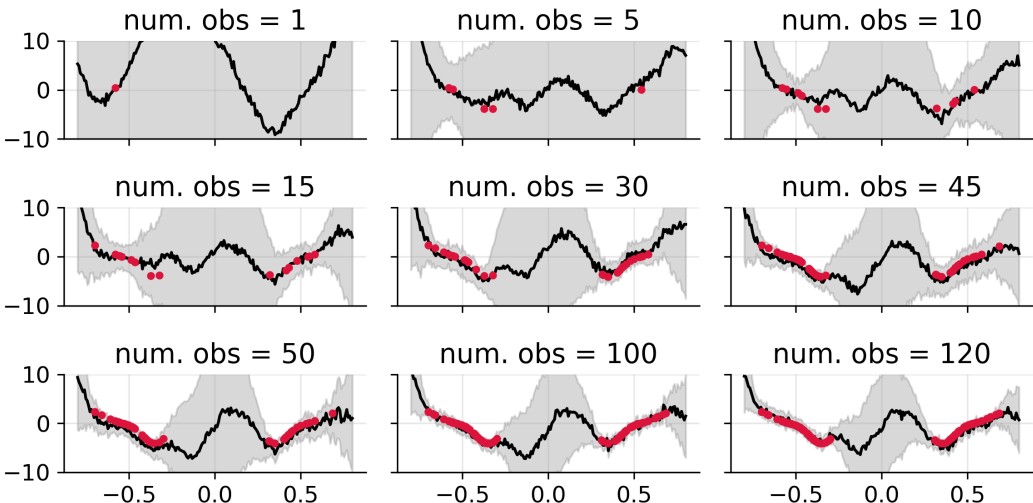

Figure 16: In-between uncertainty of the posterior predictive induced by `HiLoFi` as a function of seen observations.

An important distinction between `LRKF` and `HiLoFi` is the `HiLoFi` considers a full-rank covariance matrix in the last layer. Despite this, `LRKF` with rank 200 (which shows some notion of uncertainty) requires around 3 times more coefficients in the covariance than `HiLoFi` with rank 50 in the hidden layers.

We show the evolution of the posterior predictive mean and variance with rank 200 of `LRKF` as a function of seen observations in Figure 18.

We see that the evolution of the posterior predictive resembles that of `HiLoFi`, but at the cost of 3 times more number of coefficients in the covariance.

**Result for** `VBLL`.   We contrast the result of `HiLoFi` with the offline `VBLL` method. For `VBLL`, the data is presented all at once and we perform full-batch gradient descent using Adamw with learning rate $10^{-3}$. Figure 19 shows the posterior predictive mean and two standard deviations around the posterior mean, as well as the loss curve for various epochs.

We observe that `VBLL` takes around $10^4$ epochs to converge, whereas `HiLoFi` takes only one. We emphasize that each epoch in `VBLL` considers all datapoints at once, so the gradient update is more informative, whereas for `HiLoFi`, the method only gets to observe the data once and in a sequence.

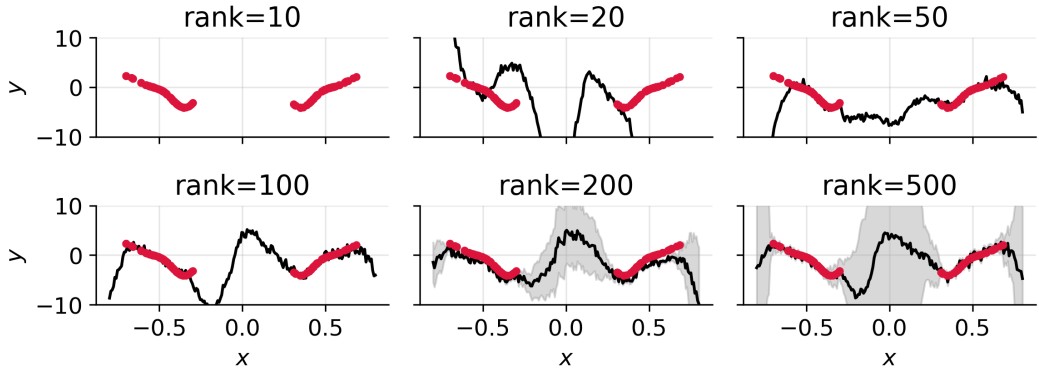

Figure 17: In-between uncertainty of the posterior predictive induced by LRKF as a function of rank.

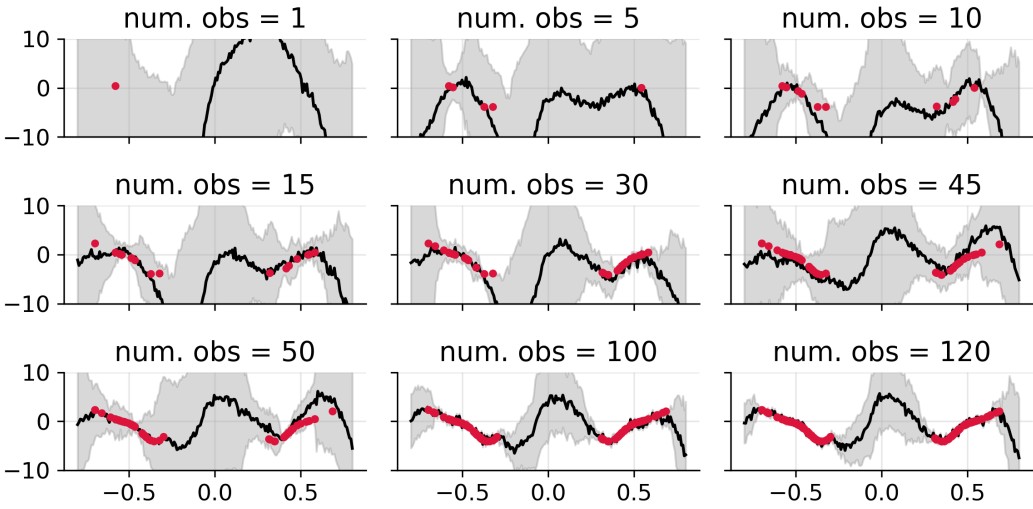

Figure 18: In-between uncertainty of the posterior predictive induced by LRKF as a function of seen observations.

This highlights the efficiency of our method to produce approximate posterior predictives that can be used online.

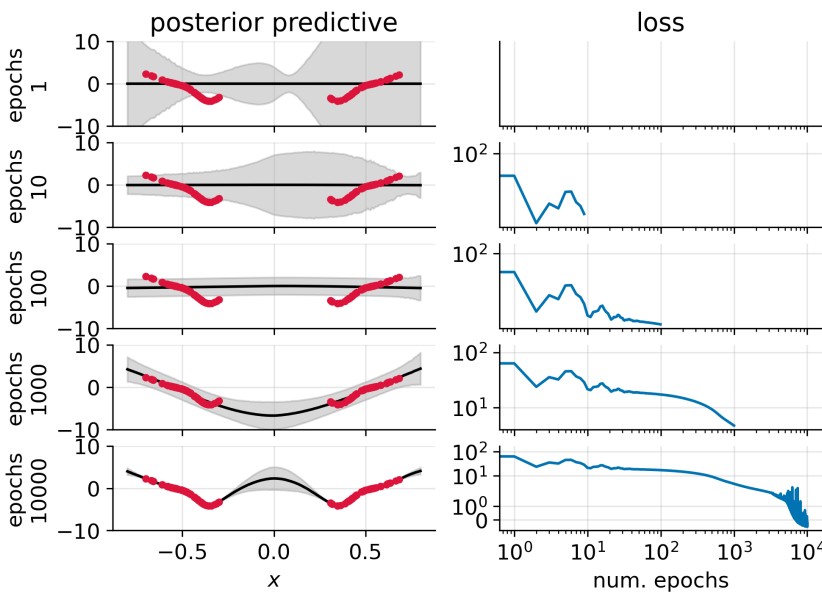

Figure 19: In-between uncertainty of the posterior predictive induced by VBLL as a function of number of epochs.

