# OpenReview forum: "Martingale Posterior Neural Networks for Fast Sequential Decision Making"
_NeurIPS.cc/2025/Conference — NeurIPS 2025 poster_

### Official Review · Reviewer_YaKM · 2025-07-01

**Clarity:** 3
**Significance:** 3
**Originality:** 3
**Rating:** 5
**Confidence:** 3

**Summary:**

This paper aims to improve the efficiency of online learning by applying the extended Kalman filtering to estimate the latent parameters. The proposed algorithm uses a recursive update of the best linear unbiased predictor and its associated error variance-covariance. Variants of the method that rely on full/low-rank approximation are considered and the tradeoff between accuracy and speed is demonstrated through a set of experiments.

**Questions:**

1. Why do you focus on uncertainty in predictions rather than uncertainty in parameters? How does this approach compare to the traditional TS?
2. Can you provide an intuitive explanation of Figure 1, the in-between uncertainty induced by HiLoFi?
3. It might be helpful to plot the cumulative regret as well, which is a more standard metric of online learning, for the bandit MNIST problem.

**Ethical Concerns:**

["NO or VERY MINOR ethics concerns only"]

**Limitations:**

yes

**Quality:**

3

**Strengths And Weaknesses:**

Strengths:
The paper introduces a novel approach for online learning based on linearization and low-rank approximations of error covariance matrices, with favorable computational efficiency.
It provides a comprehensive evaluation of the proposed methods, and the experimental results seem intuitive.
The proposed framework can be useful for solving contextual bandit problems, which have many real-world applications such as the recommender system.

Weakness:
While the paper establishes certain theoretical guarantees (Proposition 4.1), it is a little hard to gauge the tightness of the bound and the result does not seem to be very informative of the actual performance of the algorithm.
It looks like HiLoFi has a relatively high variance among the other methods in the recommender system task.

---

> ### Author Rebuttal · Authors · 2025-07-31
>
> Thank you for reviewing our paper and for your thoughtful questions.
>
> Regarding the theoretical bound in Proposition 4.1: while it is not tight, it is useful in practice for diagnosing model behavior. In particular, it helps assess whether the approximation error per step stabilizes over time. This behavior can be seen in Figure 7, where the surrogate error remains bounded throughout. Moreover, it shows that the two-step approximation incurs additive error, meaning that the errors from each approximation stage do not compound over time. We clarify this in the revised text.
>
> ## Questions
>
> **R4.1.1 Why do you focus on uncertainty in predictions rather than uncertainty in parameters?**
> A major drawback of many Bayesian neural network methods is that they make use of misspecified priors and likelihoods, often chosen for convenience [Knoblauch2022]. These choices can lead to degraded predictive performance [Wenzel2020] and make posterior sampling slower than non-Bayesian alternatives (e.g., deep ensembles). [Lakshminarayanan2017]
>
> By contrast, focusing directly on uncertainty in predictions avoids the need to define priors over "subjective probability distribution on parameters that may have no real-world interpretation", as noted by [Holmes2023]. Furthermore, sampling from the posterior predictive is typically much lower-dimensional and a more tractable object for decision making.
>
> We also observe improved performance over classical TS in deep neural networks, as demonstrated in a new experiment included in the revised version (See, R4.1.2 below for results).
>
> We clarify these points in the revised text.
>
>
> **R4.1.2 How does this approach compare to the traditional TS?**
>
> In traditional TS, one samples $\hat{\theta} \sim p(\theta | D_{1:t})$ and uses $\hat{y}_{t+1} = f(\hat{\theta}, x\_{t+1}, a)$ to make decisions. This requires the specification and sampling of a posterior over model parameters.
>
> In our predictive Bayes (pBayes) approach, one samples $\hat{y}\_{t+1} \sim p(y | x_{t+1}, \theta_{t|t})$ directly and use methods like HiLoFi/LoLoFi/LRKF to update the belief state $\theta\_{t|t}$ (and also $\Sigma\_{t|t}$ for the methods we consider) .This decoupling enables faster and more stable exploration-exploitation behavior.
>
> We clarify this distinction in the revised text, include pseudocode for the martingale-based predictive Bayes approach, and add a new experiment comparing against classical TS.
>
> The pseudocode for our proposed predictive TS in a multiarmed contextual bandit problem is given below.
>
> ```
> // Define posterior predictive parameterised by belief state b
> // and having inputs: the context ‘x’ and action ‘a’.
> def p(· | x, a, b):
>     ...
>
> For each a in actions:
>    // belief state:
>    // need not define a valid posterior or consider two moments only
>     b(t, a) = (mu(t, a), Sigma(t, a))
>
> For all a in actions:
>     // sample from Bayesian posterior predictive
>     ŷ(t, a) ~ p(· | x(t), a, b(t, a))
>
> a_next = argmax_a[ŷ(t, 1), ..., ŷ(t, A)]
>
> // frequentist update (e.g., HiLoFi)
> b(t+1, a_next) = update(b(t, a_next), y(t, a_next))
> ```
>
> In the table below, we show results for the MNIST contextual bandit task, comparing traditional TS against our pBayes approach. Note that HiLoFi does not admit a full posterior over parameters, so we report pBayes results only. For LoFi and Laplace methods, which support both TS and pBayes, we report both.
>
> | method         | cumulative reward | minutes | regret  |
> |----------------|-------------------|---------|---------|
> | HiLoFi-pBayes  | 31,167.7           | 8.43168 | 8,832.3  |
> | LoFi-ts        | 23,304.3           | 38.9005 | 16,695.7 |
> | LoFi-pBayes    | 23,184.5           | 4.76253 | 16,815.5 |
> | Laplace-ts     | 7,201.9            | 191.542 | 32,798.1 |
> | Laplace-pBayes | 13,922.3           | 1.36025 | 26,077.7 |
>
> In summary: Sampling from the posterior, as required by TS, is dramatically slower, while offering little or no performance gain. For example, Laplace-TS takes 190 minutes vs. 1.4 minutes for pBayes, with much worse reward. Even LoFi-TS is nearly 10X slower than LoFi-pBayes with similar results. pBayes offers a superior speed-performance trade-off.
>
> Note that the  significant increase in speedup is because sampling, at every step, from a 10-dimensional multivariate Gaussian is much more computationally tractable than sampling from a D-dimensional multivariate Gaussian (even if the covariance structure is diagonal-plus-low-rank, as is the case for LoFi).
>
> We add this new experiment with results in the revised text.
>
> **R4.2 Can you provide an intuitive explanation of Figure 1, the in-between uncertainty induced by HiLoFi?**
> Thank you for this suggestion.
>
> Figure 1 illustrates how explicitly modeling the last layer in full rank increases predictive uncertainty in regions without training data, where higher uncertainty is expected. Compared to fully low-rank methods (e.g., LRKF; see Figure 15 in the Appendix), HiLoFi forms more meaningful uncertainty estimates with fewer observations in these unseen regions.
>
> This provides a better balance between expressiveness of uncertainty and computational cost, which is important for managing the exploration-exploitation trade-off in sequential decision making.
>
> **R4.3 It might be helpful to plot the cumulative regret as well, which is a more standard metric of online learning, for the bandit MNIST problem.**
> Thank you for this suggestion. In the revised version of the paper, we include cumulative regret plots for the Bandit MNIST problem.
>
> ---
>
> [Knoblauch2022] Knoblauch, Jeremias, Jack Jewson, and Theodoros Damoulas. "An optimization-centric view on Bayes' rule: Reviewing and generalizing variational inference." Journal of Machine Learning Research 23.132 (2022): 1-109.
> [Wenzel2020] Wenzel, Florian, et al. "How good is the bayes posterior in deep neural networks really?." arXiv preprint arXiv:2002.02405 (2020).
> [Lakshminarayanan2017] Lakshminarayanan, Balaji, Alexander Pritzel, and Charles Blundell. "Simple and scalable predictive uncertainty estimation using deep ensembles." Advances in neural information processing systems 30 (2017)
> [Holmes2023] Holmes, Chris C., and Stephen G. Walker. "Statistical inference with exchangeability and martingales." Philosophical Transactions of the Royal Society A 381.2247 (2023): 20220143.

---

### Official Review · Reviewer_41Eo · 2025-07-01

**Clarity:** 3
**Significance:** 2
**Originality:** 3
**Rating:** 5
**Confidence:** 3

**Summary:**

- The paper proposes HiLoFi, which is a scalable method for maintaining uncertainty estimates of model parameters in online learning. The method uses a full-rank posterior over the final layer parameters with a low-rank approximation for hidden layers.
- The method can be broken down into using EKF for iterative updates and uncertainty estimation (which doesn’t necessarily yield a proper distribution on the parameter space), and using a Bayesian approach to construct a valid posterior distribution for the target space.
- Theoretical guarantees on low-rank approximation error are provided, and experiments are performed on several datasets.

**Questions:**

- In Figure 2, it seems that LoLoFi achieves better performance than HiLoFi. Do the authors have any hypothesis on why this happens?

- Intuition: why only separates the last layer (in a 3-layer MLP, for example, it may be insufficient to evaluate the method's performance when more or fewer layers use a full rank error matrix)? Are there ablation experiments for a larger number of layers?

- For proposition 4.1, are there empirical validations for this error bound? Can the rank d_h be dynamically tuned to trade off accuracy and compute?

**Ethical Concerns:**

["NO or VERY MINOR ethics concerns only"]

**Final Justification:**

The authors' newly conducted experiments addresses the majority of my concerns.

**Limitations:**

The ML model architecture choices are modest. This leaves questions about whether HiLoFi scales to models used in domains like protein design, large-scale NLP, or vision.

As mentioned in the weaknesses section, in many online learning scenarios, especially in scientific domains, samples are expensive but time is not the main constraint. It would be useful to see:

- Whether HiLoFi (and its underlying TS algorithm) offers superior sample efficiency over other acquisition approaches.
- How trade-offs between low-rank approximation fidelity and sample utility are managed.

**Quality:**

2

**Strengths And Weaknesses:**

Strengths

- The paper is clearly written: the methodology is sound, and step-by-step derivations are given to the methods to make them understandable.

- The algorithm lowers the computational burden of online learning—I can imagine it being used in situations where previous methods with GP would simply fail to fit, or BNNs that would take too long to train.

Weaknesses

- I would say that there is insufficient exploration of scalability to deeper or wider networks. Most tested architectures, 3-layer MLPs or shallow CNNs, do not reflect the scale encountered in harder tasks (e.g., protein engineering, molecular design).

- Also, in many real-world online tasks, the bottleneck is not computation time but sample efficiency. The paper does not discuss this tradeoff explicitly (it is quite crucial for many online applications like robotics or experimental design).

---

> ### Author Rebuttal · Authors · 2025-07-31
>
> Thank you for reviewing our paper.
>
> **R3.0**
> We agree with your assessment.
> In the new version of the paper, we discuss the tradeoff between sample efficiency and computation time and mention that our method is designed to tackle problems in which computation-time is the main bottleneck. This is often the case in domains such as finance [Cartea2023], recommender systems [Duran-Martin2022], and Bayesian optimization where the number of acquired datapoints is reasonably large [Brunzema2024].
>
> We also clarify in the revised version that our approach separates frequentist learning from Bayesian prediction. The model maintains a predictive distribution for decision making, updated via HiLoFi, without relying on full posterior inference. This is now reflected in the paper’s title and illustrated through added pseudocode and an additional comparison of "classical" TS and our "predictive TS" approach  (pBayes).
>
> The pseudocode for our proposed predictive TS in a multiarmed contextual bandit problem is given below.
>
> ```
> // Define posterior predictive parameterised by belief state b
> // and having inputs: the context ‘x’ and action ‘a’.
> def p(· | x, a, b):
>     ...
>
> For each a in actions:
>    // belief state:
>    // need not define a valid posterior or consider two moments only
>     b(t, a) = (mu(t, a), Sigma(t, a))
>
> For all a in actions:
>     // sample from Bayesian posterior predictive
>     ŷ(t, a) ~ p(· | x(t), a, b(t, a))
>
> a_next = argmax_a[ŷ(t, 1), ..., ŷ(t, A)]
>
> // frequentist update (e.g., HiLoFi)
> b(t+1, a_next) = update(b(t, a_next), y(t, a_next))
> ```
>
> Below we show results for the MNIST contextual bandit task, comparing traditional TS against our pBayes approach. Note that HiLoFi does not admit a full posterior over parameters, so we report pBayes results only. For LoFi and Laplace methods, which support both TS and pBayes, we report both.
>
> | method | reward  | time (min) |
> |:---------------|--------:|-----------:|
> | HiLoFi-pBayes  |  31,168 |    8.43 |
> | LoFi-TS  |  23,304 |   38.90 |
> | LoFi-pBayes |  23,185 |   4.76 |
> | Laplace-TS |  7,202 |  191.54 |
> | Laplace-pBayes |  13,922 |   1.36 |
>
> In summary: Sampling from the posterior, as required by TS, is dramatically slower, while offering little or no performance gain. For example, Laplace-TS takes 190 minutes vs. 1.4 minutes for pBayes, with much worse reward. Even LoFi-TS is nearly 10X slower than LoFi-pBayes with similar results. pBayes clearly offers a superior speed-performance trade-off.
>
> Note that the  significant increase in speedup is because sampling, at every step, from a 10-dimensional multivariate Gaussian is much more computationally tractable than sampling from a D-dimensional multivariate Gaussian (even if the covariance structure is diagonal-plus-low-rank, as is the case for LoFi).
>
> We have also introduced a new experiment with larger number of parameters ~4million (see R3.2 below) to address concerns about model scalability.
>
> ## Questions
>
> **R3.1 In Figure 2, it seems that LoLoFi achieves better performance than HiLoFi. Do the authors have any hypothesis on why this happens?**
>
> The purpose of Figure 2 is to show that when uncertainty modeling is less critical (e.g., a full-information setting), modelling the last layer explicitly performs on-par with other scalable approaches.
>
> In particular, LoLoFi outperforms HiLoFi in some cases because higher-rank models require more steps to stabilize the covariance update. With more degrees of freedom and full-information updates, convergence takes longer. In this case, lower-rank update procedures can outperform in full-information settings.
>
> We clarify this in the revised paper.
>
> **R3.2 Intuition: why only separates the last layer? Are there ablation experiments for a larger number of layers?**
> Applying full-rank updates only to the last layer is a common design choice in scalable uncertainty methods and has shown strong empirical results in bandit and Bayesian deep learning settings \[Harrison2024, Riquelme2018\].
>
> That said, we agree this warrants empirical support. We conducted a new experiment applying LRKF to VGG-style architectures with up to 4.7M parameters on CIFAR-10 after 10,000 observations. The results below show stable online classification accuracy across different low-rank approximation levels (ranks 1, 5, and 10):
>
> | model    | num. params     | accuracy (one-step-ahead) |
> |----------|-----------|----------|
> | VGG-1    | 1.7M      | 0.3355   |
> | VGG-5    | 1.7M      | 0.3384   |
> | VGG-10   | 1.7M      | 0.3379   |
> | \+VGG-1   | 4.7M      | 0.3129   |
> | \+VGG-5   | 4.7M      | 0.3141   |
> | \+VGG-10  | 4.7M      | 0.3148   |
>
> For comparison, a single-pass AdamW baseline achieves 0.3442 for VGG and 0.3097 for \+VGG.
>
> These results demonstrate that LRKF scales to multi-million parameter networks while maintaining stable performance across low-rank configurations. This supports the broader applicability of our approach (including HiLoFi and LoLoFi) to large-scale architectures in vision and related domains.
>
> In the revised paper, we also include an additional experiment of the "in-betwen uncertainty" plot as we increase the number of layers of the MLP used.
>
> **R3.3 For proposition 4.1, are there empirical validations for this error bound? Can the rank d\_h be dynamically tuned to trade off accuracy and compute?**
> Yes, please see Figure 7 in the Appendix where we show surrogate and low-rank error as well as rolling accuracy. These plots indicate that the error per step remains stable, and learning proceeds steadily across time.
>
> As a rule of thumb, increasing d\_h leads to better performance, with running time increasing at a roughly linear rate. This trend is illustrated in Figure 6 (MNIST bandit). Furthermore, Figure 13 shows that higher d\_h improves the quality of in-between uncertainty estimates.
>
> To make this point clearer,  we have also introduced a new ablation study on Bayesian optimization on the DrawNN dataset. Here,  we vary d\_h and report the relationship between performance and compute cost:
>
> | rank d\_h | time (s) | final best value |
> |----------|----------|------------------|
> | 1        | 27.5     | 656.6            |
> | 10       | 38.1     | 630.6            |
> | 20       | 27.6     | 670.9            |
> | 50       | 29.4     | 684.0            |
> | 100      | 32.6     | 782.2            |
> | 110      | 35.5     | 774.4            |
> | 120      | 39.5     | 807.3            |
> | 130      | 38.6     | 846.2            |
> | 140      | 39.6     | 848.2            |
> | 150      | 38.2     | 864.8            |
>
> The table above reports the performance of HiLoFi on the DrawNN dataset (Bayesian optimization), showing runtime in seconds and the final best value (averaged over 20 runs). As shown, both runtime and performance increase approximately linearly with d\_h, reinforcing the trade-off discussed above.
>
>
>
> We clarify these point in the revised paper (including this new experiment), emphasizing that d\_h offers a practical trade-off between accuracy and compute, depending on available resources.
>
> ---
>
> [Cartea2023] Cartea, Álvaro, Fayçal Drissi, and Pierre Osselin. "Bandits for algorithmic trading with signals." Available at SSRN 4484004 (2023).
> [Duran-Martin2022] Duran-Martin, Gerardo, Aleyna Kara, and Kevin Murphy. "Efficient online bayesian inference for neural bandits." International conference on artificial intelligence and statistics. PMLR, 2022.
> [Brunzema2024] Brunzema, Paul, et al. "Bayesian optimization via continual variational last layer training." arXiv preprint arXiv:2412.09477 (2024).
> [Harrison2024] Harrison, James, John Willes, and Jasper Snoek. "Variational Bayesian last layers." arXiv preprint arXiv:2404.11599 (2024).
> [Riquelme2018] Riquelme, Carlos, George Tucker, and Jasper Snoek. "Deep bayesian bandits showdown: An empirical comparison of bayesian deep networks for thompson sampling." arXiv preprint arXiv:1802.09127 (2018).

---

> ### Comment · Reviewer_41Eo · 2025-08-05
>
> I appreciate the authors' explanation of my questions and the addition of the experiments, especially experiments for validating scalability.
> I also like the idea of renaming the paper to incorporate the martingale posterior, as that does make a clever connection.
> For the new CIFAR-10 experiment, how is the performance evaluated? Is the one-step-ahead accuracy calculated on the posterior mean?

---

> > ### Author Response · Authors · 2025-08-06
> >
> > Thank you very much for the kind remarks.
> >
> > Regarding your question, you are completely right, similar to the MNIST experiment, the performance is the the one-step-ahead accuracy at time $t+1$ evaluated using the mean estimate $\theta_{t|t}$ found at the previous timestep. We will make this clear in the paper.

---

### Official Review · Reviewer_U5Um · 2025-07-02

**Clarity:** 2
**Significance:** 3
**Originality:** 3
**Rating:** 3
**Confidence:** 3

**Summary:**

This paper proposes a hybrid frequentist–Bayesian approach for online learning and Bayesian inference. It uses Kalman filtering to recursively update neural network parameters and their associated uncertainty, and employs Bayesian predictive sampling for decision-making. The key idea is to avoid expensive posterior inference over network parameters by instead focusing on modeling predictive uncertainty directly. The method is fully online and single-pass, requiring no replay buffers or offline retraining. The authors evaluate their approach on three sequential decision-making tasks, spanning contextual bandits and Bayesian optimization, and demonstrate that it achieves a competitive trade-off between speed and accuracy.

**Questions:**

What is the benefit of combining frequentist Kalman-style updates with Bayesian predictive sampling?

Why is the shift from uncertainty in parameters to uncertainty in predictions helpful for decision-making?

Why is Kalman filtering, especially the extended variant, a natural fit for online neural network training?

Why is replay buffer an issue in GP?

**Ethical Concerns:**

["NO or VERY MINOR ethics concerns only"]

**Final Justification:**

Most of my questions are addressed during rebuttal and discussions. But since there is a change of title, abstract, and introduction, I expect there would be a fair amount of updates.

**Limitations:**

Yes.

**Paper Formatting Concerns:**

No major formatting issue.

**Quality:**

2

**Strengths And Weaknesses:**

Strengths:

The combination of frequentist Kalman-style updates with Bayesian predictive sampling in a fully online, replay-buffer-free neural-network framework is novel and well-motivated. This method is also scalable to applications including bandits, Bayesian optimization, and classification. The experiment evaluations are comprehensive.


Weaknesses:

The paper has several issues that limit its clarity and broader accessibility.

It frequently assumes familiarity with key terminology (e.g., “replay buffer,” “in-different uncertainty”) without proper explanation. For instance, it mentions “replay-buffer approaches for Bayesian optimization” without specifying what they are (likely referring to GPs), nor does it clarify how replay buffers are costly in such methods.

The sentence “we focus on uncertainty in predictions rather than parameters” appears in the decision-making section without clearly explaining that this predictive uncertainty is used for downstream decisions (e.g., via Thompson Sampling), and why this shift of focus is beneficial in practice.

The connection between uncertainty quantification and decision-making needs to be made more explicit. Similarly, the purpose of Figure 2—contrasting full-feedback and partial-feedback settings—should be more clearly stated.

All experiment plots lack sufficient explanation. The term “performance” in Figure 5's caption is confusing as there is both "time performance" and "rank performance" in the right panel of Figure 5. The left panel of Figure 5 lacks error bars without justification or clarification of what uncertainty is being reported.

---

> ### Author Rebuttal · Authors · 2025-07-31
>
> Thank you for your review and thoughtful suggestions. Your comments have helped improve the accessibility of our paper. We summarize the revisions and respond to your questions below.
>
> We clarify that replay-buffer methods store data for repeated use in training (e.g., BNNs) or inference (e.g., GPs). While we don’t claim these methods are inherently costly, our approach avoids them entirely, which makes our method suitable for privacy-sensitive or streaming settings.
>
> Furthermore, in the revised version of the text, we introduce the notion of “in-between” uncertainty, originally discussed by \[Foong2019\]. This concept refers to how well a model captures uncertainty in regions of input space that lie between, or away from, observed data. Intuitively, we expect uncertainty to be higher in such unexplored areas. This behavior is important for Bayesian decision-making problems, where it supports effective balancing of exploration and exploitation.
>
> We have also made the connection between uncertainty quantification and sequential decision making more explicit (please see R2.1 and R2.2 below).
>
> Finally, we will improve figure explanations. For Figure 5, the left panel shows best final value vs. runtime; the right panel shows rank across tasks. Error bars are included in Figure 10 (Appendix) and are referenced in the main text (L28).
>
> **R2.0 Changes to paper**
> A key revision was to clarify the separation between the Bayesian predictive sampling and the frequentist update. To that end, we made the following changes
>
> * Renamed the paper to "Martingale Posterior Neural Networks for Fast Sequential Decision Making", which more succinctly expresses what we are doing: modelling uncertainty in predictions, rather than parameters. (Following \[Holmes2023\])
> * Revised the text to make the role of the posterior predictive in decision-making more explicit. In particular, we now mention that our approach allows us to make Bayesian decision making in a more efficient way than the prior-times-likelihood approach.
> * Introduced a new pseudoalgorithm to illustrate the key predict-act-update in contextual bandits and Bayesian optimization. For contextual bandits with one model per arm, the pseudoalgorithm has the form
>
>
> ```
> // Define posterior predictive parameterised by belief state b
> // and having inputs: the context ‘x’ and action ‘a’.
> def p(· | x, a, b):
>     ...
>
> For each a in actions:
>    // belief state:
>    // need not define a valid posterior or consider two moments only
>     b(t, a) = (mu(t, a), Sigma(t, a))
>
> For all a in actions:
>     // sample from Bayesian posterior predictive
>     ŷ(t, a) ~ p(· | x(t), a, b(t, a))
>
> a_next = argmax_a[ŷ(t, 1), ..., ŷ(t, A)]
>
> // frequentist update (e.g., HiLoFi)
> b(t+1, a_next) = update(b(t, a_next), y(t, a_next))
> ```
>
> This illustrates how Bayesian decision-making via the posterior predictive is combined with constant-cost frequentist updates.
>
> ## Questions
>
> **R2.1 What is the benefit of combining frequentist Kalman-style updates with Bayesian predictive sampling?**
> The Kalman-style frequentist update provides a lightweight alternative to a posterior parameter update. Instead of tracking the full posterior, we use the predictive distribution directly for sampling-based decision making (Thompson-sampling style). This achieves efficient online learning without sacrificing exploration-exploitation quality.
>
> This style of separating the update of model parameters from the model used to make predictions (the posterior predictive) has been called the martingale posterior [Holmes2023]. In the new version of the paper, we changed the title to "Martingale Posterior Neural Networks for Fast Sequential Decision Making", modified the introduction, and modified abstract to make this distinction more explicit.
>
> **R2.2 Why is the shift from uncertainty in parameters to uncertainty in predictions helpful for decision making?**
> Bayesian neural networks typically rely on priors and likelihoods that are misspecified or chosen for convenience [Knoblauch2022], which can degrade predictive performance [Wenzel2020] and make posterior sampling significantly slower than non-Bayesian alternatives like deep ensembles [Lakshminarayanan2017]. Moreover, these priors often involve subjective parameter distributions with no real-world interpretation [Holmes2023].
>
> By shifting focus to predictive uncertainty, we bypass the need for such priors and instead work directly with the posterior predictive, which is a lower-dimensional and more tractable object for decision-making.
>
> We introduced a new experiment on the MNIST bandit, where we observe that predictive sampling achieves comparable or better cumulative reward than traditional TS, while being up to 100x faster in high-dimensional models. We clarify this distinction and its practical benefits in the revised text.
>
> The results are shown in the table below
>
> | method | reward  | time (min) |
> |:---------------|--------:|-----------:|
> | HiLoFi-pBayes  |  31,168 |    8.43 |
> | LoFi-TS  |  23,304 |   38.90 |
> | LoFi-pBayes |  23,185 |   4.76 |
> | Laplace-TS  | 7,202 |  191.54 |
> | Laplace-pBayes |  13,922 |   1.36 |
>
> **R2.3 Why is Kalman filtering, especially the extended variant, a natural fit for online neural network training?**
> The (Extended) Kalman Filter is a form of second-order optimizer, equivalent to natural gradient descent. In practice, this means that each update is more sample-efficient than "rank-one" updates like, e.g., Adam
> [Yann2018] [Khan2023] [Jones2024].
>
> We clarify this in the main text.
>
> **R2.4 Why is replay buffer an issue in GP?**
> We do not claim replay buffers are inherently problematic. Instead, we emphasize that our method sidesteps this entirely. This is valuable in domains where data cannot be revisited (e.g., for privacy reasons [Tenenbaum2021]), where input distributions change over time (rendering stale data misleading [Chang2023]), or where the inputs are high-dimensional observations [Harrison2024].
>
> We will make sure to remark this observation in the revised text.
>
> ----
>
> [Knoblauch2022] Knoblauch, Jeremias, Jack Jewson, and Theodoros Damoulas. "An optimization-centric view on Bayes' rule: Reviewing and generalizing variational inference." Journal of Machine Learning Research 23.132 (2022): 1-109.
> [Wenzel2020] Wenzel, Florian, et al. "How good is the bayes posterior in deep neural networks really?." arXiv preprint arXiv:2002.02405 (2020).
> [Lakshminarayanan2017] Lakshminarayanan, Balaji, Alexander Pritzel, and Charles Blundell. "Simple and scalable predictive uncertainty estimation using deep ensembles." Advances in neural information processing systems 30 (2017)
> [Yann2018] Ollivier, Yann. "Online natural gradient as a Kalman filter." (2018): 2930-2961.
> [Khan2023] Khan, Mohammad Emtiyaz, and Håvard Rue. "The Bayesian learning rule." Journal of Machine Learning Research 24.281 (2023): 1-46.
> [Jones2024] Jones, Matt, Peter Chang, and Kevin Murphy. "Bayesian online natural gradient (BONG)." Advances in Neural Information Processing Systems 37 (2024): 131104-131153.
> [Tenenbaum2021] Tenenbaum, Jay, et al. "Differentially private multi-armed bandits in the shuffle model." Advances in Neural Information Processing Systems 34 (2021): 24956-24967.
> [Chang2023] Chang, Peter G., et al. "Low-rank extended Kalman filtering for online learning of neural networks from streaming data." Conference on Lifelong Learning Agents. PMLR, 2023.
> [Harrison2024] Harrison, James, John Willes, and Jasper Snoek. "Variational Bayesian last layers." arXiv preprint arXiv:2404.11599 (2024).
> [Holmes2023] Holmes, Chris C., and Stephen G. Walker. "Statistical inference with exchangeability and martingales." Philosophical Transactions of the Royal Society A 381.2247 (2023): 20220143.
> [Foong2019] Foong, Andrew YK, et al. "'In-Between'Uncertainty in Bayesian Neural Networks." arXiv preprint arXiv:1906.11537 (2019).

---

> > ### Comment · Reviewer_U5Um · 2025-08-04
> >
> > Thanks for your clarification. I have two follow-up questions:
> >
> > 1. Could you elaborate more on "martingale posterior" since you plan to switch to this title? Why it's called "martingale"?
> >
> > 2. Can any acquisition function in Bayesian optimization be adapted to your framework, or just TS?

---

> > > ### Author Response · Authors · 2025-08-04
> > >
> > > Thank you for your followup questions.
> > >
> > > > Could you elaborate more on "martingale posterior" since you plan to switch to this title? Why it's called "martingale"?
> > >
> > > The martingale posterior idea is from [Fong2023] (JRSS-B).
> > >
> > > Typically one thinks about Bayes in the context of neural networks in two ways: (i) for estimation of model parameters (the so-called posterior estimation,  typically through variational Bayes) and (ii) for usage in some downstream task such as Bayesian optimisation or bandits, which takes the estimated posterior to _induce_ uncertainty. Roughly speaking, the idea behind the martingale posterior (MP) [Fong2023] is to decouple the parameter estimation step (i) and the uncertainty estimation step (ii) by directly modelling a posterior predictive parameterized with some functional of the data. In our setting, this posterior predictive is given by Equation (4) and the functional of the data is $\theta_{t|t} = (h_{t|t}, \ell_{t|t})$.
> > >
> > > As shown in [Fong2023], a _valid_ functional of the data is required to be a martingale. In our setting, this means $E[\theta_{t+1|t+1} | y_{1:t}] = \theta_{t|t}$. Which is true since
> > >
> > > $$
> > > \begin{aligned}
> > > E[\theta_{t+1|t+1} | y_{1:t}]
> > > &= E[E[\theta_{t+1} | y_{1:t+1}] | y_{1:t}]\\\\
> > > &= E[\theta_{t+1} | y_{1:t}]\\\\
> > > &= E[\theta_t  + u_{t+1} | y_{1:t}]\\\\
> > > &= E[\theta_t | y_{1:t}] + E[u_{t+1} | y_{1:t}]\\\\
> > > &= E[\theta_t | y_{1:t}]\\\\
> > > &= \theta_{t|t}.
> > > \end{aligned}
> > > $$
> > > Here,
> > > the third equality and fifth equality follows follows from the state-space model assumption
> > > with random walk (L94).
> > >
> > > While the original works considers the estimation of the posterior predictive of all future observation $p(y_{1+1:\infty} | y_{1:t})$, in our sequential decision making setting we are only interested in the posterior predictive of the next observation since, once a decision is made, the outcome is observed and one can update the estimate $\theta_{t|t}$ using the “true observation”.
> > >
> > > We will include the above in the revised version of the paper.
> > >
> > >
> > > [Fong2023] Fong, Edwin, Chris Holmes, and Stephen G. Walker. "Martingale posterior distributions." Journal of the Royal Statistical Society Series B: Statistical Methodology 85.5 (2023): 1357-1391.
> > >
> > >
> > > > Can any acquisition function in Bayesian optimization be adapted to your framework, or just TS?
> > >
> > > Yes, any acquisition function can be adapted to fit our  framework. For example, we provide an experiment of Bayesian optimization using expected improvement (EI) in the Appendix. Please see Figure 11.

---

> > > > ### Comment · Reviewer_U5Um · 2025-08-04
> > > >
> > > > Thanks for your follow-up response. I suggest adding more discussion on alternative acquisition function. Currently it's buried in the appendix.
> > > >
> > > > I still have a confusion: what's your justification for the state-space model assumption with random walk in L94?
> > > >
> > > > I'd also like to see an updated abstract, as suggested by Reviewer `K1Y3`.

---

> > > > > ### Author Response · Authors · 2025-08-04
> > > > >
> > > > > > I suggest adding more discussion on alternative acquisition function
> > > > >
> > > > > Thank you for the suggestion. We have added discussion of alternative acquisition functions in the introduction and will further elaborate on this in Section 3.1 (Tools for Sequential Decision Making).
> > > > >
> > > > > > what's your justification for the state-space model assumption with random walk in L94?
> > > > >
> > > > > The justification is twofold: numerical stability and adaptivity: "In the classical filtering literature, the [state-space assumption] is typically used to model system dynamics. However, it has long been known that inflating [the dynamics covariance $q_t$] can compensate for unmodelled errors [Kelly2002] [Kuhl1990]" — [Duranmartin2025]. This is exactly the setting we are working with.
> > > > >
> > > > > In recent years, the use of a state-space assumption for online learning for neural networks has also been considered in, e.g., [Titsias2023] [Chang2023] [Gusakov2025].
> > > > > Please see related work for further information.
> > > > >
> > > > >
> > > > > ## Updated abstract
> > > > >
> > > > > We introduce scalable algorithms for online learning and sequential decision making through neural networks. Our methods are based on the martingale posterior idea: rather than working with the classical prior-times-likelihood formulation, we sequentially update a pre-defined posterior predictive density, which is parameterized by a neural network. Sequential decision making is performed through this posterior predictive, avoiding the need to sample from a posterior over model parameters. Our algorithms operate in a fully online, replay-free setting, and perform structured, curvature-aware updates by applying low-rank approximations to the hidden layers and full-rank updates to the final layer. We demonstrate empirically that our methods achieve competitive tradeoffs between speed and accuracy in non-stationary contextual bandits and Bayesian optimization, and offer 10 to 100 times faster inference than classical Thompson sampling while maintaining comparable or superior decision performance.
> > > > >
> > > > > ---
> > > > >
> > > > > [Kelly2002] Kelly, R. J. "Reducing geometric dilution of precision using ridge regression." IEEE transactions on aerospace and electronic systems 26.1 (2002): 154-168.
> > > > > [Kuhl1990] Kuhl, Mark R. "Ridge regression signal processing." NASA, Langley Research Center, Joint University Program for Air Transportation Research, 1989-1990 (1990).
> > > > > [Duranmartin2025] Duran-Martin, Gerardo. "Adaptive, Robust and Scalable Bayesian Filtering for Online Learning." arXiv preprint arXiv:2505.07267 (2025).
> > > > > [Titsias2023] Titsias, Michalis K., et al. "Kalman filter for online classification of non-stationary data." arXiv preprint arXiv:2306.08448 (2023).
> > > > > [Chang2023] Chang, Peter G., et al. "Low-rank extended Kalman filtering for online learning of neural networks from streaming data." arXiv preprint arXiv:2305.19535 (2023).
> > > > > [Gusakov2025] Gusakov, Yakov, et al. "Rapid Online Bayesian Learning for Deep Receivers." ICASSP 2025-2025 IEEE International Conference on Acoustics, Speech and Signal Processing (ICASSP). IEEE, 2025.

---

> ### Comment · Reviewer_U5Um · 2025-08-05
>
> Thanks for explaining Markovian posterior and sharing the updated abstract.
>
> However, I still have a few concerns.
>
> 1. From my perspective, the new title is less compelling than the previous one. The term Martingale posterior may not be widely recognized, and it’s unclear that it refers to a mechanism for uncertainty quantification and efficient belief updates.
>
> 2. I noticed that in the earlier abstract and introduction, you described your method as a combination of frequentist and Bayesian approaches. This framing seems to have been removed in the new version. Could you share your reasoning for the change?
>
> 3. In the first sentence of your new introduction, you state that "uncertainty quantification and efficient belief updates ... Bayesian approach is Thompson sampling". My understanding is that Thompson sampling is a decision-making strategy that relies on uncertainty quantification, rather than being a method for uncertainty quantification and efficient belief updates. Could you clarify why this change was made?

---

> > ### Author Response · Authors · 2025-08-06
> >
> > > From my perspective, the new title is less compelling than the previous one. The term Markovian posterior may not be widely recognized, and it’s unclear that it refers to a mechanism for uncertainty quantification and efficient belief updates.
> >
> > Thank you for your comment. Just to clarify, the term we use is **martingale** posterior. This concept was introduced in Fong, Holmes & Walker (2023), published in Journal of the Royal Statistical Society: Series B, and was listed among the initial keywords of our paper.
> >
> > While we agree the term may not yet be widely recognized in the Bayesian machine learning literature, we believe it is a valuable idea to highlight. The term directly captures our approach: specifying a predictive distribution and updating beliefs sequentially, without relying on the classical prior-times-likelihood formulation.
> >
> > ---
> >
> > > I noticed that in the earlier abstract and introduction, you described your method as a combination of frequentist and Bayesian approaches. This framing seems to have been removed in the new version. Could you share your reasoning for the change?
> >
> > Thank you for pointing this out. In the original abstract, we used the "frequentist + Bayesian" phrasing to convey the core design: frequentist-style parameter updates paired with Bayesian-style decision-making. In the revised abstract, we aimed to make this split more explicit by focusing on the mechanics: posterior predictive sampling and structured updates.
> >
> > However, we appreciate that the earlier wording may have helped orient readers more quickly to the hybrid nature of the approach. To address this, we will reintroduce this framing with minimal edits that preserve clarity and precision. Specifically, we will revise the abstract by introducing the following keywords (in [**bold**])
> >
> > > “[**Bayesian**] Sequential decision making is performed through this posterior predictive […]”
> > > “Our algorithms operate in a fully online, replay-free setting, and perform structured, curvature-aware updates by applying [**frequentist Kalman-filter-like**] low-rank updates to the hidden layers and full-rank updates to the final layer […]”
> >
> > ---
> >
> > >  In the first sentence of your new introduction, you state that "uncertainty quantification and efficient belief updates ... Bayesian approach is Thompson sampling". My understanding is that Thompson sampling is a decision-making strategy that relies on uncertainty quantification, rather than being a method for uncertainty quantification and efficient belief updates. Could you clarify why this change was made?
> >
> >
> > Thank you for this question.
> >
> > In the original introduction, we began with a similar framing:
> >
> > "In sequential decision-making problems, [...] uncertainty quantification and efficient belief updates are essential for balancing exploration and exploitation"
> > Later, in line 79, we introduced Thompson Sampling (TS) as an example of an approach that incorporates both elements.
> >
> > The revised version simply makes this connection more explicit. So while the wording has changed, the conceptual framing remains the same. Our goal was to highlight that TS is a strategy that couples both uncertainty quantification (via posterior sampling) and belief updates (via sequential inference).
> >
> > For example, suppose we have $A > 1$ arms and let $f(\theta, a)$ be the reward function modelled by parameters $\theta$ and action $a$. Then, in "classical" Bayesian TS for bandits, the algorithm is as follows
> > - At time $t+1$, sample $\hat{\theta} \sim p(\theta | y_{1:t})$
> > - Evaluate $\hat{y}_{t+1,a} = f(\hat{\theta}, a)$ for $a=1,\ldots A$
> > - Select action $a_{t+1}  = \max\{\hat{y}\_{t+1,a}\}$ and obtain reward $y_{t+1}$
> > - After observing $y_{t+1}$, we update the belief to  $p(\theta | y_{1:t+1}) \propto p(\theta | y_{1:t}) \, p(y_{t+1} | \theta)$
> >
> > This loop illustrates how TS naturally involves both components: decision-making based uncertainty (steps 1-3), and belief refinement through Bayesian updates (step 4).
> >
> > We will clarify this point further in the text.

---

> > > ### Comment · Reviewer_U5Um · 2025-08-06
> > >
> > > Sorry for the typo and thanks for your explanations. I'm still confused. Does the comma in step 4 mean times? How is this different from the classical prior-times-likelihood?

---

> > > > ### Author Response · Authors · 2025-08-06
> > > >
> > > > Apologies for the confusion.
> > > >
> > > > Step 4 is indeed the "classical" prior-times-likelihood formulation. We should have written $p(\theta | y\_{1:t+1}) \propto p(\theta | y\_{1:t}) p(y\_{t+1} | \theta)$ without the comma.
> > > >
> > > > Thank you for pointing this out.

---

> > > > > ### Comment · Reviewer_U5Um · 2025-08-07
> > > > >
> > > > > Thanks for your clarification. I thought only step 1-3 is called "Thompson sampling" while step 4 is called "Bayesian belief update" (Bayes' rule), as step 4 is not unique to TS, but a core part of any Bayesian approach. Could you also clarify how your method does not rely on step 4?

---

> ### Author Response · Authors · 2025-08-07
>
> Thank you for your comment.
>
> We believe your original question was about how Thompson Sampling (TS) is typically characterized in the literature, rather than about the specifics of our method:
>
> > "My understanding is that Thompson sampling is a decision-making strategy that relies on uncertainty quantification, rather than being a method for uncertainty quantification and efficient belief updates."
>
> Our earlier response presented the classical Bayesian TS algorithm for bandits (not our own approach), and our point was simply that TS inherently involves both uncertainty quantification and belief updating. We would like to point out that this perspective is consistent with the standard literature. For example, Algorithm 4.2 in:
>
> > Russo et al. (2018), *A Tutorial on Thompson Sampling*, *Foundations and Trends in Machine Learning*, 11(1):1–96
>
> explicitly includes all four steps: sampling, action selection, reward observation, and belief update of parameter distribution (belief update).
>
> With regard to our method, we would like to emphasize that it is not classical TS (i.e., steps 1–4 above), but rather the variant described in our response **R2.0**. Specifically, our approach does not sample from the posterior over model parameters, but instead from the posterior predictive. The belief update step does not necessarily follow a Bayesian (prior-times-likelihood) rule; as you correctly noted, we use frequentist-style updates, such as those implemented in HiLoFi, LoLoFi, or LRKF.  Finally, we show in a new experiment that our approach is significantly faster than classical Thompson Sampling, while achieving comparable or better performance. Please see the table in our response **R2.2**
>
>
>
> We hope this resolves your concerns.

---

> ### Comment · Reviewer_U5Um · 2025-08-07
>
> Right, my original question was about how Thompson Sampling is typically characterized in the literature and I also understand that your steps 1-4 is referring to the classical Bayesian TS. It's just I thought only steps 1-3 are usually considered part of TS while step 4 is not. That’s why I found the sentence in your revised introduction a bit confusing. I appreciate you sharing Daniel Russo’s tutorial to help clarify this.
>
> My follow-up question—about how your method does not rely on step 4—was meant to ask whether your method also has a similar step 1-3 but without $\theta$, and also without step 4. This is the key point I would like to confirm. Could you also write the high-level idea of your method in a similar clean way?

---

> > ### Author Response · Authors · 2025-08-07
> >
> > Thank you for your follow-up question.
> >
> > The table below gives a high-level comparison between our predictive strategy and the classical Thompson Sampling (TS) approach:
> >
> > | Step           | Our approach                             | Classical TS                                   |
> > |----------------|----------------------------------------|------------------------------------------------|
> > | 1. Sample       | Posterior predictive at given action   | Posterior over model parameters                |
> > | 2. Evaluate     | —                                      | Evaluate sampled parameter on reward function at given action        |
> > | 3. Select action| Argmax over sampled rewards            | Argmax over sampled rewards                    |
> > | 4. Update belief| Frequentist update (e.g., HiLoFi)      | Bayesian update over model parameters          |
> >
> > Key distinctions:
> >
> > 1. We sample rewards for all actions directly from the posterior predictive (as defined in Equation (4)), while TS samples from the posterior over model parameters.
> > 2. TS must then evaluate the sampled parameter on the reward function for all actions. Our method skips this step, since the reward is sampled directly.
> > 3. Both approaches use the argmax over sampled rewards to select the next action.
> > 4. After selecting an arm, TS performs a Bayesian update of the parameter posterior. In contrast, we apply a frequentist update to the parameters that define our posterior predictive (Equation (4)).
> >
> > We hope this clarifies the high-level structure of our approach in relation to classical TS.

---

> > > ### Comment · Reviewer_U5Um · 2025-08-07
> > >
> > > This is much clearer to me now. I suggest incorporating this kind of comparison into the revised version, as it really helps clarify the distinctions.
> > >
> > > One follow-up question: you mentioned that UCB and EI can be adapted to your framework. How would them map onto the steps you’ve outlined?

---

> > > > ### Author Response · Authors · 2025-08-07
> > > >
> > > > Thank you for the suggestion.
> > > >
> > > > We will include the table comparison in the revised text to improve clarity.
> > > >
> > > > Regarding EI and UCB: since these approaches do not involve sampling, they map to the same structure as follows:
> > > >
> > > > | Step                 | Our approach (sampling-based)               | Our approach (EI / UCB)                              |
> > > > |----------------------|----------------------------------------------|------------------------------------------------------|
> > > > | 1. Compute decision input | Sample from posterior predictive | Compute mean and variance of posterior predictive     |
> > > > | 2. Evaluate           | —                                            | —                                                    |
> > > > | 3. Select action      | Argmax over sampled rewards                  | Argmax over acquisition function (e.g., EI, UCB)     |
> > > > | 4. Update belief      | Frequentist update (e.g., HiLoFi)            | Frequentist update (e.g., HiLoFi)                    |
> > > >
> > > > In both cases, we rely on the same belief state. The key differences are that Step 1 becomes deterministic in EI/UCB (no sampling), and Step 3 selects actions based on a utility function rather than sampled rewards.
> > > >
> > > > We will also clarify this point in the revised text.

---

> > > > > ### Comment · Reviewer_U5Um · 2025-08-07
> > > > >
> > > > > Thanks for your clarification! I have no more questions.

---

> > > > > > ### Author Response · Authors · 2025-08-07
> > > > > >
> > > > > > You're welcome!
> > > > > >
> > > > > > We’ll make sure to clarify all of the above points in the revised text to improve both clarity and the presentation of our contributions.

---

### Official Review · Reviewer_K1Y3 · 2025-07-02

**Clarity:** 2
**Significance:** 3
**Originality:** 2
**Rating:** 4
**Confidence:** 4

**Summary:**

The paper proposes a class of scalable algorithms for sequential training of neural networks. The method sequentially computes best linear unbiased prediction (frequentist parameter estimate) and construct a pseudo posterior predictive using a Gaussian approximation with a block-diagonal error variance-covariance estimate and proceeds in an online fashion. The method is shown to achieve a good balance between speed and accuracy for a variety of online learning problems.

**Questions:**

1. If one is not interested in the uncertainty of the parameters, but rather of the predictions, why use a Bayesian model at all? For contextual bandits, example would be UCB-type algorithms. Since you are not constructing the full posterior, but approximating it as a normal by plug-in mean and variance frequentist estimates, what purpose does Bayesian inference serve in the work?
2. I am a bit worried about the setup. Do you assume the parameters at each time $\{\theta_t\}$ related in any way (as often done in hierarchical models for borrowing strength)? If not, what connects the observations at different time points? In your example of contextual bandit, the underlying $\theta$ is assumed to be fixed (not depending on time), only the estimate $\hat{\theta}_t$ depends on time - since all previous observations (say from same arm) are connected through that parameter. Can the authors clarify the setup here. In the case you assume a linear dynamic evolution (as in Section 3.2), how sensitive is the algorithm to misspecification of this dynamics?
3. For the bandit experiments, did you compare the results with a usual TS? Or maybe a high-dimensional version if the contexts are high-dimensional?
4. Using a point estimate (for example MAP) and using it for sequential learning in Bayesian models, through the posterior predictive, has been proposed in the literature - for example (i) Sequential Algorithm for fast fitting of DPM models [Zhang, Nott, Yau, Jasra], (ii) Fast Bayesian Inference in DPM models [Wang, Dunson], (iii) Scalable nonparametric Bayesian learning for dynamic velocity fields [Chakraborty, Guha, Lei, Nguyen] - how does your work differ in essence from these. It might be good to include some of these references.
5. Proposition 4.1 gives some idea - but can you elaborate on the role of the following $R_t, q_{\ell,t}, q_{h,t}$ on the overall algorithm? It seems these are hyper parameters - a discussion on the tuning might be good.
6. Can the method be used in probabilistic models where the observation noise is not additive? Or its variance unknown (related to point 5)
7. Proposition 4.1 - even when no low-rank structure is utilized, there is no consistent estimation of $\Sigma$ as $t\to\infty$ - can you comment on why this is true?
8. Can you say something about the estimation quality of the BLUP, characterize $\Vert \theta_t - \hat{\theta}_t\Vert$? Or at least the predictive accuracy?

**Ethical Concerns:**

["NO or VERY MINOR ethics concerns only"]

**Final Justification:**

score updated

**Limitations:**

See the questions.

**Paper Formatting Concerns:**

1. If possible, avoid using “latent parameter”; instead use only parameter. Latent potentially also refers to latent variables (in Bayesian inference), which is not exactly the case for the neural network parameters in Bayesian neural network.
2. For the notations, a table might be good since there are too many. Include all the notations, for example $\Sigma_t, \Sigma_{t|k}, \hat{\Sigma}_{\ell,t}, \tilde{\Sigma}_t$.
3. For algorithm 1, do not place numbering on the comment lines (e.g. //predict step)

**Quality:**

2

**Strengths And Weaknesses:**

The paper is well written (although the notations are a bit too much) and the extensive experiments demonstrate promise to the proposed methodology. The only concern is novelty (since both sequential estimation by approximate posterior)

---

> ### Author Rebuttal · Authors · 2025-07-31
>
> Thank you for your review.
>
> We believe the statement in your Strengths/Weaknesses section was left incomplete. That said, we hope our responses address your concerns.
>
> To clarify novelty: we introduce a constant‑cost, Kalman‑style update that keeps a single theta estimate for a neural network and enables Bayesian decisions via the posterior predictive. Unlike prior cited DPM/iHMM work (i-iii above), we target tractable online neural‑learning and sequential decision‑making.
>
> We appreciate the formatting suggestions. We will add a table of notation, remove numbering on comments in Algorithm 1, and use "parameter" only.
>
> **R1.1: Why Use a Bayesian Model If You Don’t Model Parameter Uncertainty?**
>
> We adopt the martingale posterior view [Holmes2023], in which the posterior predictive drives sequential decisions and the belief state is updated separately. Although classical TS samples parameters, it is ultimately outcome uncertainty that drives exploration. Our method captures this directly via the posterior predictive.
>
> This new approach has several advantages:
>
> * It enables Thompson sampling (TS)-style exploration without sampling from a high-dimensional posterior (which comes at a 10-100x speedup with comparable or increased performance. See R1.4)
> * It supports plug-and-play non-Bayesian learning updates while retaining Bayesian decision logic, e.g., using HiLoFi.
>
> As you mention, UCB and also Expected improvement are valid alternatives; we include an EI example in Figure 11 (Appendix).
>
> We will emphasize this framing more clearly. Specifically:
> * Rename the paper to "Martingale Posterior Neural Networks for Fast Sequential Decision Making",
> * Update the abstract and introduction,
> * Add a new pseudoalgorithm to illustrate the sampling + frequentist update loop.
>
> For contextual bandits with one model per arm, the pseudoalgorithm is
> ```
> // Define posterior predictive parameterised by belief state b
> // and having inputs: the context ‘x’ and action ‘a’.
> def p(· | x, a, b):
>     ...
>
> For each a in actions:
>    // belief state:
>    // need not define a valid posterior or consider two moments only
>     b(t, a) = (mu(t, a), Sigma(t, a))
>
> For all a in actions:
>     // sample from Bayesian posterior predictive
>     ŷ(t, a) ~ p(· | x(t), a, b(t, a))
>
> a_next = argmax_a[ŷ(t, 1), ..., ŷ(t, A)]
>
> // frequentist update (e.g., HiLoFi)
> b(t+1, a_next) = update(b(t, a_next), y(t, a_next))
> ```
>
> **R1.2 Do you assume the parameters at each time related in any way?**
>
> We do not use a hierarchical model for "borrowing strength". Instead, we assume that the parameters evolve according to a random walk, (Line 94), i.e., $\theta_t = \theta_{t-1} + u_t$, with $var(u_t) = q I_D$.
>
> This assumption is common in adaptive filtering and online learning (see e.g. [Ismail1996] and [Chang2023]) and provides a simple way to ensure adaptivity and numerical stability.
>
> In the special case where $q=0$, and without a low-rank assumption, we recover a static parameter model and our estimate $\theta_{t|t}$ is simply the Bayesian posterior mean given data up to time t, i.e., $\theta_{t|t} = E[\theta | y_{1:t}]$.
>
> **R1.3 How sensitive is the algorithm to misspecification of this dynamics?**
>
> The dynamics $\theta_t = \theta_{t-1} +  u_t$ are misspecified by design. They are intended to serve as a regularisation mechanism for adaptivity and numerical stability, rather than reflecting the true data generating process. This dates back to, e.g., [Mehra1970].
>
> To empirically assess sensitivity, we conducted a new experiment on the DrawNN Bayesian Optimization benchmark mentioned in the paper (using 600 steps). We varied both the hidden layer ($q_h$) and last-layer ($q_\ell$) dynamics for HiLoFi across several values and report the final maximum value found (averaged over 10 runs).
>
> |qh/qh|0.0|1e-06|1e-05|0.0001|0.001|
> |---|---|---|---|---|---|
> |0|911.023|925.541|903.86|856.529|850.668|
> |1e-06|908.951|877.041|902.752|906.037|915.497|
> |1e-05|885.912|858.062|902.754|883.262|882.13|
> |0.0001|880.655|905.342|877.899|861.019|895.526|
> |0.001|834.449|791.294|823.188|809.964|809.104|
>
> The results are robust. Some tuning helps but no catastrophic failure was observed. We’ll include these results in the revision.
>
> **R1.4 For the bandit experiments, did you compare the results with a usual TS?**
> Thank you for the suggestion.
>
> Given an arm $a$,  standard TS samples $\hat{\theta} \sim p(\theta | D_{1:t})$ and uses $\hat{y}\_{t+1} = f(\hat{\theta}, x\_{t+1}, a)$ to make decision. In predictive TS (pBayes), we sample directly from $\hat{y}_{t+1} \sim p(y |x\_{t+1}, \theta\_{t|t}, a)$.
> In the revised paper, we expand our MNIST bandit experiment to compare standard TS with pBayes approach.
>
> We find that pBayes is 10-100x faster, than classical TS, while also giving comparative or higher cumulative reward. The speedup stems from sampling a low-dimensional predictive distribution instead of a high-dimensional parameter posterior, even when that posterior has a structured form (e.g., diagonal-plus-low-rank, as in LoFi).
>
> The table below shows the results of comparing standard TS (LoFi, Laplace) and predictive TS (HiLoFi, LoFi, Laplace) on the 	MSNIST bandit task.
>
> |method|reward|time(min)|
> |---|---|---|
> |HiLoFi-pBayes|31,168|8.43|
> |LoFi-TS|23,304|38.90|
> |LoFi-pBayes|23,185|4.76|
> |Laplace-TS|7,202|191.54|
> |Laplace-pBayes|13,922|1.36|
>
>
> **R1.5 How does your use of posterior predictive with point estimates differ from prior MAP-based approaches in the literature**
>
> The works suggested above (i-iii) use MAP‐plug‑in updates for nonparametric mixture models (iHMMs/DPMs) to accelerate posterior inference. Our approach departs in three key ways:
>
> 1. Those papers maintain a growing set of mixture components, whereas we keep and update only one global parameter vector $\theta_{t|t}$ that (possibly) evolves over time.
> 2. We target neural networks trained sequentially, with constant-time per update. These settings are not addressed by the cited DPM literature.
> 3. We tackle the problem of sequential decision making using so-called Martingale posteriors, which is not considered in cited works.
>
> In the revised section, we reference each of the papers you mention and emphasize that our approach is a single-estimate parametric approach for online learning and sequential decision making.
>
> **R1.6 Can you elaborate on the roles of R, q(ell), and q(h), and how they should be tuned?**
>
> $R_t$ is the heteroskedastic noise process. Its purpose is to downweight the update information arising from natural variation of the observations y. This parameter can be estimated using an exponentially-weighted moving variance or in a burn-in period, as we do in the experiments.
>
> The q-values, $q_\ell$ and $q_h$,  are parameters for adaptation and stability:
>
> * $q_\ell$  is the speed of adaptation for the last layer and
> * $q_h$ is the speed of adaptation for the hidden layers.
>
> As a rule of thumb: $q_h \leq q_\ell$ because, for a neural network, the capacity of the hidden layers is much greater than that of the last layer. In our experiments, these parameters are estimated during an initial phase or using alternative methods such as those in, e.g., [Titsias2023]
>
> **R1.7 Can the method be used in probabilistic models where the observation noise is not additive? Or its variance unknown (related to point 5)**
> Our method could be employed in parametric probabilistic models where the observation noise is not additive, however, these are better suited for sampling-based methods, similarly, for unknown variance, one can use methods such as [Agamennoni2012].
>
> **R1.8 Consistency of Proposition 4.1 when no low-rank structure is used**
> If no low-rank structure is used, then we would couple all layers and there is no need to use the "surrogate matrix" (shown in Equation (94),  L787 in the Appendix) required for the low-rank projection. Hence, the approximation error would be zero (by definition).
>
> **R1.9 Can you say something about the estimation quality of the BLUP?**
> While little can be said about the true $\theta$, we can bound the estimation error by comparing a low-rank approximation (e.g., LRKF) against a full-rank update. In the linear regression case $y\_t = \theta^\intercal x\_t + e\_t$ with fixed $\theta$ and known $var(e\_t) = r$, let $\Sigma\_{t-1}$ be the full-rank covariance  The one-step parameter difference between the BLUP using full-rank and using a rank-d update is upper bounded by:
>
> $$
> \\|\theta\_{t|t}-\hat{\theta}_{t|t}\\|\_2 \leq |\epsilon\_t| \sigma\_{d+1}(\Sigma\_{t-1})\\|x\_t\\|_2 \left( \frac{1}{\gamma\_t} \+ \frac{\sigma\_{1}(\Sigma\_{t-1})\\|x\_t\\|_2^{2}}{\gamma\_t^{2}} \right), $$
>
> Here, $\sigma\_{d+1}(\Sigma_{t-1})$ is the first discarded singular value and $\sigma\_1(\Sigma_{t-1})1$ is the largest singular value, $\theta_{t|t}$ is the BLUP computed using the full rank and $\hat{\theta}_{t|t}$ is the BLUP with rank-d LRKF update, $\gamma\_t  =\max\\{x\_t^{\intercal}\Sigma\_{t-1}x\_t, x\_t^{\intercal}\widehat\Sigma\_{t-1}x\_t\\} \+ r$, and
> $\epsilon_t = y\_t - \theta\_{t-1|t-1}^\intercal x\_t$.
>
> We can see that if $d=D$, then the error is zero. In general, having a fixed  innovation and features, the error grows linearly with the first discarded singular value. A derivation will be provided in the revised text.
>
> [Holmes2023] Holmes & Walker (2023). Statistical inference with exchangeability and martingales, Phil. Trans. R. Soc. A.
> [Ismail1996] Ismail & Principe (1996). Equivalence between RLS algorithms and the ridge regression technique, IEEE Conf.
> [Chang2023] Chang et al. (2023). Low-rank EKF for NN learning, Conf. on Lifelong Learning Agents.
> [Mehra1970] Mehra (1970). Adaptive filtering approaches, IEEE CDC.
> [Titsias2023] Titsias et at. (2023). Kalman filter for online classification of non-stationary data arXiv:2306.08448
> [Agamennoni2012] Agamennoni et al. (2012) Approximate inference in state-space models with heavy-tailed noise, IEEE Transactions on Signal Processing

---

> > ### Comment · Reviewer_K1Y3 · 2025-08-04
> >
> > Thank you for the responses. My concerns have been fairly addressed. However, since changing the title, abstract and introduction is typically a big change - can you share the updated abstract and introduction?

---

> > > ### Author Response · Authors · 2025-08-04
> > >
> > > We are glad to know that your concerns have been addressed. Please see the updated abstract and introduction below.
> > >
> > >
> > > # Abstract
> > > We introduce scalable algorithms for online learning and sequential decision making through neural networks. Our methods are based on the martingale posterior idea: rather than working with the classical prior-times-likelihood formulation, we sequentially update a pre-defined posterior predictive density, which is parameterized by a neural network. Sequential decision making is performed through this posterior predictive, avoiding the need to sample from a posterior over model parameters. Our algorithms operate in a fully online, replay-free setting, and perform structured, curvature-aware updates by applying low-rank approximations to the hidden layers and full-rank updates to the final layer. We demonstrate empirically that our methods achieve competitive tradeoffs between speed and accuracy in non-stationary contextual bandits and Bayesian optimization, and offer 10 to 100 times faster inference than classical Thompson sampling while maintaining comparable or superior decision performance.
> > >
> > > # Introduction
> > >
> > > In various sequential decision-making problems, such as Bayesian optimization and contextual bandits, uncertainty quantification and efficient belief updates are essential for balancing exploration and exploitation. A prominent Bayesian approach is Thompson sampling [russo2020], which samples from the posterior over model parameters and evaluates the corresponding predictive function (e.g., reward or surrogate) for decision-making.
> > >
> > > Recent work aims to combine Bayesian decision-making with the expressiveness of neural networks for sequential decision making [brunzema2024][riquelme2018]. These methods, often called Bayesian neural networks (BNNs), typically approximate the posterior over model parameters (e.g., via variational inference) and use it for downstream tasks. However, BNNs often rely on misspecified priors and likelihoods [knoblauch2022], which can degrade predictive performance [wenzel2020] and result in slower inference than non-Bayesian alternatives [lakshminarayanan2017].
> > >
> > > These limitations are especially problematic in online settings where updates and decisions must be made quickly, such as in recommender systems, adaptive control, financial forecasting, and large-scale Bayesian optimization [shen2015][liu2022][mcdonald2023][cartea2023][waldon2024].
> > >
> > > To address these problems, we present a class of algorithms for Bayesian sequential decision making based on the martingale posterior framework [fong2023]. This approach directly models a posterior predictive density parameterized by a functional of the data, avoiding the need for subjective priors or costly posterior inference.
> > >
> > > The parameters in the predictive density are updated using Kalman filter–style methods, which resemble online natural gradient steps [ollivier2018] and require no replay buffer. We explore three structured covariance strategies:
> > >  LRKF applies low-rank updates to all layers;
> > >  HiLoFi applies a full-rank update to the last layer and a low-rank update to the hidden layers;
> > >  LoLoFi applies low-rank updates to both.
> > >  HiLoFi and LoLoFi are inspired by last-layer Bayesian methods [harrison2024variational], and offer different trade-offs between computational efficiency and expressiveness.
> > >
> > > Decision making is performed by sampling directly from the posterior predictive, resulting in 10 to 100 times faster inference than classical TS, even when compared to structured posteriors that use diagonal plus low-rank covariances [chang23]. Our algorithms scale to million-parameter networks, maintain constant-time updates, and operate in fully online, streaming-compatible settings. In addition, our methodology supports any acquisition strategy, such as classical TS or expected improvement.
> > >
> > > In summary, our probabilistic method for training neural networks enjoys the following properties:
> > >  (i) efficient and closed-form updates (no Monte Carlo sampling required);
> > >  (ii) closed-form and uncertainty-aware predictions via the posterior predictive.
> > >
> > > We evaluate the performance of our methods across a range of sequential decision-making tasks. In non-stationary neural contextual bandits, our methods achieve the highest performance at the lowest computational cost compared to baselines. In stationary settings such as Bayesian optimization, our approach matches the performance of replay-buffer methods while achieving Pareto-efficient trade-offs between runtime and performance.

---

> > > > ### Author Response · Authors · 2025-08-04
> > > >
> > > > Please see below for the references
> > > >
> > > > * [russo2020]: Russo, D. J., Van Roy, B., Kazerouni, A., Osband, I., Wen, Z.,. (2018). A tutorial on Thompson sampling. Foundations and Trends® in Machine Learning, 11(1), 1–96.
> > > > * [brunzema2024]: Brunzema, P., Jordahn, M., Willes, J., Trimpe, S., Snoek, J., Harrison, J.. (2024). Bayesian Optimization via Continual Variational Last Layer Training. arXiv preprint arXiv:2412.09477.
> > > > * [riquelme2018]: Riquelme, C., Tucker, G., & Snoek, J. (2018). Deep Bayesian bandits showdown: An empirical comparison of Bayesian deep networks for Thompson sampling. arXiv preprint arXiv:1802.09127.
> > > > * [knoblauch2022]: Knoblauch, J., Jewson, J., & Damoulas, T. (2022). An optimization-centric view on Bayes’ rule: Reviewing and generalizing variational inference. Journal of Machine Learning Research, 23(132), 1–109.
> > > > * [wenzel2020]: Wenzel, F., Roth, K., Veeling, B., Swiatkowski, J., Tran, L., Mandt, S., Snoek, J., Salimans, T., Jenatton, R., & Nowozin, S. (2020). How Good is the Bayes Posterior in Deep Neural Networks Really? In International Conference on Machine Learning (pp. 10248–10259). PMLR.
> > > > * [lakshminarayanan2017]: Lakshminarayanan, B., Pritzel, A., & Blundell, C. (2017). Simple and scalable predictive uncertainty estimation using deep ensembles. Advances in Neural Information Processing Systems, 30.
> > > > * [shen2015]: Shen, W., Wang, J., Jiang, Y.-G., & Zha, H. (2015). Portfolio Choices with Orthogonal Bandit Learning. IJCAI, 15, 974–980.
> > > > * [liu2022]: Liu, Z., Zou, L., Zou, X., Wang, C., Zhang, B., Tang, D., Zhu, B., Zhu, Y., Wu, P., Wang, K.,. (2022). Monolith: real time recommendation system with collisionless embedding table. arXiv preprint arXiv:2209.07663.
> > > > * [mcdonald2023]: McDonald, T. M., Maystre, L., Lalmas, M., Russo, D., & Ciosek, K. (2023). Impatient bandits: Optimizing recommendations for the long-term without delay. Proceedings of the 29th ACM SIGKDD Conference on Knowledge Discovery and Data Mining, 1687–1697.
> > > > [cartea2023]: Cartea, Á., Drissi, F., & Osselin, P. (2023). Bandits for algorithmic trading with signals. Available at SSRN 4484004.
> > > > * [waldon2024]: Waldon, H., Drissi, F., Limmer, Y., Berdica, U., Foerster, J. N., & Cartea, Á. (2024). DARE: The deep adaptive regulator for control of uncertain continuous-time systems. ICML 2024 Workshop: Foundations of Reinforcement Learning and Control–Connections and Perspectives.
> > > > * [fong2023]: Fong, E., Holmes, C., & Walker, S. G. (2023). Martingale posterior distributions. Journal of the Royal Statistical Society Series B: Statistical Methodology, 85(5), 1357–1391.
> > > > * [ollivier2018]: Ollivier, Y. (2018). Online natural gradient as a Kalman filter. Electronic Journal of Statistics, 12(2), 2930–2961.
> > > > * [chang23]: Chang, P. G., Durán-Martín, G., Shestopaloff, A., Jones, M., & Murphy, K. P. (2023). Low-rank extended Kalman filtering for online learning of neural networks from streaming data. Proceedings of The 2nd Conference on Lifelong Learning Agents, 232, 1025–1071.

---

### Note · Authors · 2025-08-11

We thank all reviewers for their constructive feedback. Below, we summarize the main points raised during the discussion phase and how they will be addressed in the final version.

**Positive comments from reviewers**
* Novelty of our method (YaKM, U5Um).
* The clarity of the paper (K1Y3, 41Eo).
* The breadth and number of experiments (K1Y3, YaKM, U5Um).
* The favorable computational efficiency of the method (41Eo, YaKM, U5Um).
* Scalability in fully online settings (41Eo, YaKM, U5Um).
* The conceptual framing combining frequentist Kalman-style updates with Bayesian predictive sampling (U5Um).

**Main constructive feedback received**
* Explain and motivate the frequentist versus Bayesian part of the proposed methodology.
* Include additional ablation experiments and expand on the explanation of some of the material.

We have addressed all reviewer concerns and completed all additional experiments. The following changes will be made to improve the clarity and presentation of our contributions:

**Revisions**
* New title, abstract, and introduction reflecting the martingale posterior framing and hybrid frequentist–Bayesian nature of our approach (**K1Y3**). These updates clarify the original contributions based on reviewer feedback, without altering the underlying content.
* Table of notations (**K1Y3**).
* Clarification of the distinction between classical Thompson Sampling (TS) and our predictive sampling approach (**U5Um, R2.0**).
* High-level comparison tables: our predictive sampling approach vs. (i)  classical TS, and (ii) vs. EI/UCB acquisition functions (**U5Um**).
* Pseudocode for contextual bandits using our predictive sampling approach (**R3.0**).
* New experiment between our predictive sampling and classical TS (**R1.4**).
* New ablation study on sensitivity to dynamics parameters (**R1.3**).
* Experiments with deeper architectures (~4.7M params) on CIFAR-10 (**R3.2**).
* Clarification that our method targets settings where compute time is the primary bottleneck (**R3.0**).
* Regret plots for contextual bandit tasks (**R4.3**).
* Expanded background and explanation of in-between uncertainty (**R4.2, U5Um**).
* Clarification of the state-space model motivation and dynamics parameters (**K1Y3, U5Um**).
* Clarification and new theoretical results (**YaKM, R1.9, U5Um**).

We thank the reviewers again for their engagement. We believe the revised version of the paper will present our contributions more clearly and accessibly.

---

### Decision · Program_Chairs · 2025-09-17

**Decision:**

Accept (poster)

**Comment:**

This paper primarily makes an algorithmic contribution towards the very relevant and practical problem of learning neural network models along with a notion of uncertainty. The performance evaluation of the proposed approach, based on extended Kalman filtering for a creatively set up model of a dynamically evolving neural network together with computationally friendly low-rank approximations, is chiefly empirical. The approach is geared especially towards enabling efficient randomization for posterior-sampling style algorithms for sequential decision making.

The initial reviews saw the referees raising several questions and requesting clarifications on various aspects. These included the necessity for Bayesian inference for learning predictive output distributions as oppposed to conventionally learning the posterior over the network parameters, comparison to other related work that uses point estimates in the context of sequential Bayesian learning, the motivation for using a novel random-walk model for the evolving network parameters, sensitivity of the algorithm to misspecification, explanations for terms used without adequate definition (e.g., 'replay buffer approaches'), more insight about 'martingale posteriors', and the comparison of the suggested approach with the vanilla version of Thompson sampling involving maintenance of uncertainty in parameter space.

The authors and reviewers engaged in lengthy and patient discussions to clarify many of these points, which was noted positively by several reviewers, and for which I thank all the reviewers and authors. The discussion period also saw the authors include the results of additional experiments and ablations related to benchmarking prediction errors for architectures with more neural layers, and measuring the computational cost incurred by the proposed scheme.

Overall, most of the reviewers lean positive on this paper to varying degrees, with one reviewer being (borderline) negative; however, many of the questions raised by the reviewers have been adequately addressed by the authors. After going through the paper and the discussions post review, I am inclined to recommend acceptance in view of the fact that the paper proposes a conceptually new approach, based on a linear dynamical systems view, towards prediction with uncertainty quantification for neural pipelines, and hope that these ideas can spur more innovation both on the theoretical front (refined studies of error bounds and their improvement) and on the practical front (more extensive empirical evaluation on real-world datasets) to perhaps result in a method competitive with the current state of the art.